# VISUALPUZZLES: Decoupling Multimodal Reasoning Evaluation from Domain Knowledge

Yueqi Song [* 1]  Tianyue Ou [* 1]  Yibo Kong [† 1]  Zecheng Li [† 1]  Graham Neubig [1]  Xiang Yue [1]

## Abstract

Current multimodal benchmarks often conflate reasoning with domain knowledge, making it difficult to isolate and evaluate reasoning abilities in non-expert settings. To address this, we introduce VISUALPUZZLES, a benchmark that targets visual reasoning while deliberately minimizing reliance on specialized knowledge. VISUALPUZZLES consists of diverse puzzles spanning 5 categories: algorithmic, analogical, deductive, inductive, and spatial reasoning. One major source of our questions is manually translated logical reasoning questions from the Chinese Civil Service Examination. Experiments show that VISUALPUZZLES requires significantly less intensive domain-specific knowledge and more complex reasoning compared to benchmarks like MMMU, enabling us to better evaluate genuine multimodal reasoning. Evaluations show that state-of-the-art multimodal large language models consistently lag behind human performance on VISUALPUZZLES, and that strong performance on knowledge-intensive benchmarks does not necessarily translate to success on reasoning-focused, knowledge-light tasks. Additionally, reasoning enhancements such as scaling up inference compute (with "thinking" modes) yield inconsistent gains across models and task types, and we observe no clear correlation between model size and performance. We also found that models exhibit different reasoning and answering patterns on VISUALPUZZLES compared to benchmarks with heavier emphasis on knowledge. VISUALPUZZLES offers a clearer lens through which to evaluate reasoning capabilities beyond factual recall and domain knowledge.

[*]Equal Contributions. [†]Equal Contributions. [1]Carnegie Mellon University, Pittsburgh, PA, USA. Correspondence to: Yueqi Song <yueqis@cs.cmu.edu>, Tianyue Ou <tianyueo@cs.cmu.edu>, Graham Neubig <gneubig@cs.cmu.edu>, Xiang Yue <xyue2@cs.cmu.edu>.

*Proceedings of the $43^{rd}$ International Conference on Machine Learning*, Seoul, South Korea. PMLR 306, 2026. Copyright 2026 by the author(s).

## 1. Introduction

*Reasoning* is a cornerstone of both human and artificial intelligence, enabling systems to solve problems, draw inferences, and make decisions from information. Recent advances in multimodal large language models (MLLMs) (Anthropic, 2023; 2025; Grattafiori et al., 2024; Gemini Team, 2024; Comanici et al., 2025; Jaech et al., 2024; Li et al., 2025; Liu et al., 2023a; Hurst et al., 2024; OpenAI, 2025; Bai et al., 2025; Yue et al., 2025a) exhibit early signs of reasoning in tackling complex tasks such as answering expert-level visual questions (Winata et al., 2025; Yue et al., 2024; 2025b), interpreting scientific figures (Roberts et al., 2024), and solving challenging math problems (Lu et al., 2024).

Many of the tasks mentioned above are inherently *knowledge-intensive*; large amounts of knowledge in domains such as science or math are necessary to answer questions correctly (Yue et al., 2024). However, in reality, reasoning does not necessitate knowledge. Even non-expert humans can successfully solve logic puzzles, spatial reasoning problems, and analogical tasks using general inferential skills, without requiring deep domain expertise. This raises an important question: *Can we measure MLLMs' reasoning ability independently of measuring their acquisition of domain-specific knowledge?* This question is particularly important with the recent rapid development of reasoning models in the textual domain, and emerging application to the visual domain (Anthropic, 2025; DeepSeek-AI, 2025; Gemini Team, 2025; OpenAI, 2025; Qwen Team, 2024).

To address this question, we introduce VISUALPUZZLES, a multimodal benchmark explicitly crafted to assess reasoning capabilities independent of specialized knowledge. VISUALPUZZLES comprises 1,168 carefully curated puzzle-like questions that span five distinct categories of reasoning: algorithmic, analogical, deductive, inductive, and spatial, each annotated with varying difficulty levels. VISUALPUZZLES only requires basic common knowledge and information presented in the puzzles to solve problems, disentangling reasoning from domain-specific knowledge. Our experiments show that VISUALPUZZLES requires significantly fewer domain-specific knowledge concepts compared to benchmarks like MMMU (Yue et al., 2024; 2025b), and models have sufficient knowledge to solve VISUALPUZZLES ques-

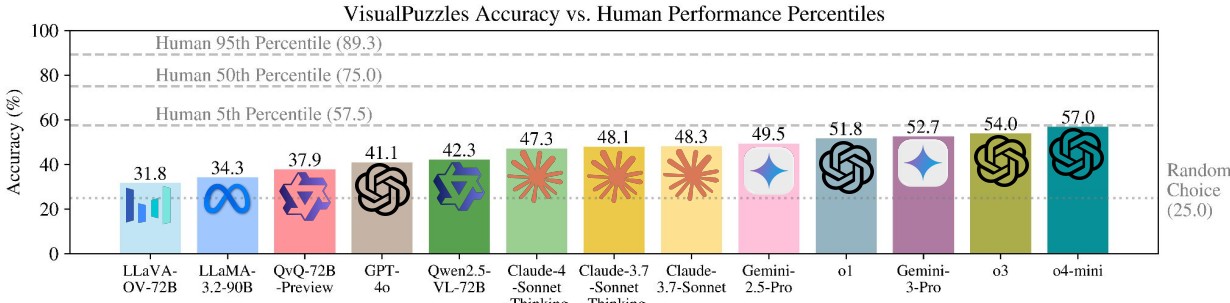

*Figure 1.* Model accuracy on VISUALPUZZLES compared to human performance percentiles. All evaluated models fall below the human 5th percentile (57.5%), highlighting the difficulty of VISUALPUZZLES. Interestingly, models with explicit "thinking" modes do not consistently outperform their base versions, suggesting that current reasoning strategies do not yet generalize well to VISUALPUZZLES, even though these strategies have proven effective in existing reasoning tasks that often rely heavily on domain-specific knowledge.

tions, enabling us to better assess multimodal reasoning versus pretrained factual knowledge. While VISUALPUZZLES minimizes reliance on domain expertise, its reasoning complexity exceeds that of existing benchmarks: in VISUALPUZZLES, 82.1% of models' solution steps are logical reasoning steps, compared to 71.5% in MMMU. Additionally, no current MLLM surpasses even the 5th-percentile human performance, highlighting VISUALPUZZLES's difficulty. Our experiments reveal critical limitations in current MLLMs' visual reasoning ability by factoring out domain knowledge and only focusing on reasoning. Specifically, we uncover four key findings:

- **Strong performance on knowledge-heavy benchmarks does not transfer well.** Models that rank highly on MMMU often see substantial performance drops on VISUALPUZZLES, highlighting a gap between knowledge-rich and knowledge-light visual reasoning tasks.

- **Humans outperform models on easy and medium tasks, while both degrade on harder ones.** In contrast, models struggle even on simpler tasks.

- **Scaling model size does not ensure stronger reasoning.** We observe no clear trend indicating that larger models outperform smaller ones on VISUALPUZZLES, suggesting that scaling up parameters alone is insufficient to improve domain-agnostic multimodal reasoning.

- **Reasoning enhancements (e.g., long CoT and "thinking" mode) yield inconsistent gains.** While explicit reasoning strategies help certain models tackle complex reasoning tasks, these techniques do not consistently improve performance across all models and tasks.

## 2. VISUALPUZZLES

### 2.1. Motivation and Design Principles

Existing benchmarks often conflate multimodal reasoning with domain-specific knowledge, making it difficult to isolate and measure models' pure reasoning capabilities.

VISUALPUZZLES is designed to explicitly address this issue by providing a testbed focused on evaluating multimodal reasoning in isolation from specialized knowledge. Specifically, VISUALPUZZLES centers on puzzle-like questions that rely solely on the provided image, question text, and basic common-sense knowledge. The core design principle behind VISUALPUZZLES is to limit the need for external or pretrained domain knowledge. Figure 2 shows various examples of VISUALPUZZLES.

### 2.2. Data Collection and Curation

We curated VISUALPUZZLES using a multi-stage pipeline. The process involved sourcing, adapting, and validating questions with an emphasis on reasoning quality and minimal reliance on knowledge.

**Question Sourcing**. We collected questions from three primary sources: (1) online resources and textbooks focused on logical, visual, and spatial puzzles, (2) synthesized items using images from large-scale vision datasets paired with text prompts, and (3) carefully repurposed items from existing multimodal reasoning benchmarks. Each source was selected to ensure a wide variety of reasoning challenges while avoiding trivial or fact-heavy questions. One major source of our questions is manually translated logical reasoning questions from the Chinese Civil Service Examination[1].

**Format Adaptation**. All collected items were adapted into a consistent multiple-choice format with four options, balancing between text-based and image-based answer choices. This modality balance allows us to better test models' abilities to perform reasoning across diverse formats.

**Data Validation**. During curation, we applied strict filtering criteria to eliminate questions requiring advanced mathematical knowledge, specialized domain knowledge and facts. Questions were retained only if they could be solved using information present in the image, the question prompt,

---

[1] Chinese Civil Service Examination (Logic Test), 中国国家公务员考试行测（逻辑推理）. Other sources listed in § B.2.

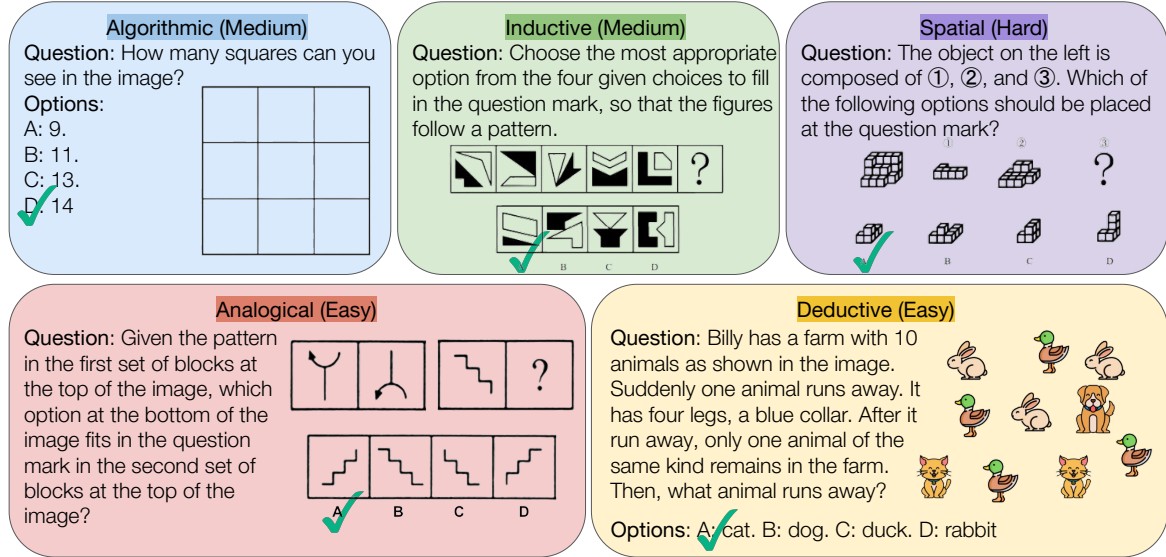

*Figure 2.* Example VISUALPUZZLES instances within each reasoning category.

and basic common sense. A multi-round validation process was conducted by human annotators, focusing on question clarity, solvability, and reasoning type classification.

**Attribute Annotation**. Finally, each question was annotated with two key attributes:

- Reasoning Category: Each item was categorized as *algorithmic*, *analogical*, *deductive*, *inductive*, or *spatial* reasoning. These five categories were selected as they represent fundamental forms of reasoning widely discussed in literature (Gao et al., 2023; Liu et al., 2020; Lu et al., 2024; Yue et al., 2024). At the same time, we aimed to balance comprehensiveness with conciseness, avoiding an overly fine-grained taxonomy that could dilute the benchmark's clarity and usability. This categorization ensures that VISUALPUZZLES covers a broad yet manageable set of reasoning skills relevant to multimodal LLMs.

  - Algorithmic Reasoning involves reasoning over algorithmic rules.
  - Analogical Reasoning requires analyzing the relationships between a pair of entities.
  - Deductive Reasoning involves logically drawing conclusions from known premises.
  - Inductive Reasoning focuses on generalizing rules from observed patterns.
  - Spatial Reasoning requires interpreting and manipulating spatial relationships.

- Difficulty Level: Labeled as easy, medium, or hard, based on annotators' estimated cognitive load and time-to-solve metrics.

This pipeline ensures that VISUALPUZZLES presents a diverse set of high-quality questions designed to challenge MLLMs on their reasoning abilities without involving pretrained domain knowledge.

### 2.3. Dataset Statistics

| Category | Statistics |
|---|---|
| Total Questions | 1168 |
| - Algorithmic Reasoning | 262 |
| - Analogical Reasoning | 211 |
| - Deductive Reasoning | 200 |
| - Inductive Reasoning | 209 |
| - Spatial Reasoning | 286 |
| Easy/Medium/Hard | 46%/39%/15% |
| Option Type (Image/Text) | 57%/43% |
| AVG. Question Length | 154.9 |
| % Easy Words | 54% |

*Table 1.* Statistics of VISUALPUZZLES.

VISUALPUZZLES comprises 1,168 multimodal reasoning puzzles. It is designed to provide a balanced distribution across reasoning categories, difficulty levels, and option formats for comprehensive evaluation. Table 1 shows statistics of VISUALPUZZLES. Across the five reasoning types, we maintain a roughly even distribution, ensuring that no single reasoning style dominates the benchmark. Similarly, we balanced the dataset across the three difficulty levels (easy, medium, hard) to capture a wide spectrum of cognitive demands. Approximately half of the answer choices in the dataset are image-based and the other half are text-based, enabling evaluation of models' abilities to reason across diverse query formats. Additionally, VISUALPUZZLES was constructed with an emphasis on accessibility,

with most of the question text uses Basic English vocabulary[2] to minimize the impact of linguistic complexity on reasoning performance, focusing the evaluation strictly on multimodal reasoning.

Compared to prior benchmarks, VISUALPUZZLES is unique in that it explicitly minimizes domain-specific knowledge requirements while maintaining high reasoning complexity. We demonstrate these traits of VISUALPUZZLES in § 4.

## 3. Experiments and Results

### 3.1. Experimental Setup

We comprehensively evaluated a variety of MLLMs on VISUALPUZZLES[3]. Additionally, we performed human evaluations to better understand the gap between human and models' reasoning capabilities. We selected a diverse set of proprietary and open MLLMs to ensure broad coverage of models. This diversity allows us to capture a wide spectrum of current approaches and capabilities in the field. We applied both direct multiple-choice and Chain-of-Thought (CoT) prompting to each model, following recent findings that CoT can significantly enhance model reasoning (Wei et al., 2022; Zhang et al., 2023).

**Human Performance.** To establish a strong baseline for comparison, we conducted human evaluations with 70 college-level volunteers. While human performance provides a valuable upper-bound reference for assessing the current capabilities and limitations of multimodal reasoning model, it is possible that future models could surpass human performance. Each participant was randomly assigned a subset of the puzzles and completed them under the same resource-constrained conditions as the models (without access to external tools or the internet). On average, participants completed each puzzle in 78 seconds, reflecting VISUALPUZZLES's cognitive load and time demands.

### 3.2. Overall Results

Table 2 and Figure 1 compare the performance of humans and a selected set of models.[4] For each model we report the best performance, whether achieved by direct multiple-choice or CoT prompting. All evaluated models, even the proprietary ones, perform below the 5th percentile of human accuracy, underscoring the significant gap in multimodal reasoning abilities. These results reinforce our finding that, although models have made progress in multimodal understanding, there remains a substantial margin for improve-

---

[2]https://en.wiktionary.org/wiki/Appendix: Basic_English_word_list.

[3]A full list of these models can be found in Table 11.

[4]Full results for every model discussed in § 3 are provided in Appendix E, including separate performance outcomes for both direct multiple-choice and CoT prompting.

ment before they can match or surpass human performance on multimodal reasoning.

This pattern holds across categories as well. In Table 2, top human participants (95th percentile) exhibit near-perfect accuracy on multiple reasoning categories, while model performance remains substantially lower, even lower than the worst human performance (5th percentile). These results emphasize the need for continued innovation in model architectures and training paradigms if we aim to close the gap between model and human intelligence on complex multimodal reasoning.

## 4. Disentangling Reasoning from Domain Knowledge

### 4.1. Knowledge Intensity of VISUALPUZZLES

**Is VISUALPUZZLES less knowledge-intensive than existing reasoning benchmarks?** This question is central to our goal of disentangling reasoning ability from domain-specific knowledge. Many benchmarks blur this line, making it difficult to assess reasoning in non-expert settings. VISUALPUZZLES was designed to target visual reasoning while minimizing reliance on specialized knowledge.

To test whether VISUALPUZZLES achieves this goal, we prompted GPT-4o to generate "knowledge concept checklists" for 50 randomly selected questions from a widely-used knowledge-intensive reasoning dataset MMMU (Yue et al., 2024) and 50 from VISUALPUZZLES, and we manually verified each as discussed in § F.5. Each checklist comprises knowledge-specific questions intended to assess whether models possess the background information needed to solve the original task. For example, if a question requires understanding two physics laws, its checklist would include a question to explain each. The number of checklist items per instance serves as a proxy for knowledge intensity.

We found that MMMU problems resulted in significantly more checklist items on average (3.9) compared to VISUALPUZZLES (1.1), as shown in Table 3. This supports the hypothesis that VISUALPUZZLES is substantially less reliant on domain knowledge. As a result, performance on VISUALPUZZLES more directly reflects a model's ability to reason over visual and textual content, offering a clearer signal of progress in multimodal reasoning. Full prompt examples and further discussion are provided in Appendix F.

**Do models already possess the knowledge required to solve VISUALPUZZLES?** To explore this, we measured models' knowledge accuracy—their ability to answer the knowledge checklist questions correctly—on both benchmarks. This metric reflects how much of the required knowledge is already known by the model, independent of reasoning. We found a stark contrast: while many models exceed

| Model | Algorithmic | Analogical | Deductive | Inductive | Spatial | Overall |
|---|---|---|---|---|---|---|
| Random Choice | 25.0 | 25.0 | 25.0 | 25.0 | 25.0 | 25.0 |
| Human (95th Percentile) | 100.0 | 100.0 | 100.0 | 81.6 | 100.0 | 89.3 |
| Human (50th Percentile) | 88.0 | 66.0 | 80.0 | 50.0 | 90.0 | 75.0 |
| Human (5th Percentile) | 68.1 | 25.0 | 37.0 | 0.0 | 59.1 | 57.5 |
| *Proprietary Models* | | | | | | |
| GPT-4o | 49.2 | 58.3 | 49.0 | 27.3 | 26.2 | 41.3 |
| o1 | 63.7 | 68.3 | 67.5 | 29.2 | 34.3 | 51.8 |
| o3 | 64.5 | 68.3 | 69.5 | 27.3 | 42.7 | 54.0 |
| o4-mini | 65.3 | 68.7 | 75.5 | 33.0 | 45.5 | 57.0 |
| Gemini-2.0-flash | 55.3 | 58.8 | 57.0 | 24.4 | 31.8 | 45.0 |
| Gemini-2.0-flash-thinking | 46.6 | 70.1 | 49.0 | 24.9 | 25.5 | 42.2 |
| Gemini-2.5-pro | 60.0 | 64.0 | 60.0 | 29.7 | 36.4 | 49.5 |
| Gemini-3-pro | 67.6 | 60.7 | 49.0 | 34.0 | 49.7 | 52.7 |
| Claude-3.7-Sonnet | 64.5 | 48.3 | 65.0 | 26.8 | 37.4 | 48.3 |
| Claude-3.7-Sonnet-Thinking | 67.2 | 44.1 | 61.5 | 31.1 | 37.1 | 48.2 |
| Claude-4-Sonnet-Thinking | 64.9 | 46.5 | 60.5 | 25.8 | 38.1 | 47.3 |
| Claude-4.5-Sonnet-Thinking | 67.2 | 51.2 | 69.0 | 27.8 | 33.9 | 49.4 |
| Claude-4.5-Opus-Thinking | 67.6 | 59.7 | 71.0 | 28.2 | 37.8 | 52.4 |
| Claude-4.6-Opus-Thinking | 72.9 | 58.3 | 76.5 | 23.9 | 39.2 | 53.9 |
| Kimi-K2.5 | 62.2 | 56.9 | 61.0 | 26.8 | 48.6 | 51.4 |
| *Open Models (Qwen-Based)* | | | | | | |
| LLaVA-OV-7B | 27.5 | 28.0 | 40.5 | 24.4 | 28.0 | 29.4 |
| Pangea-7B | 32.4 | 23.7 | 38.5 | 28.7 | 32.5 | 31.3 |
| Qwen2.5-VL-7B-Instruct | 38.2 | 23.7 | 51.5 | 24.9 | 31.1 | 33.7 |
| LLaVA-OV-72B | 34.7 | 26.5 | 37.0 | 27.3 | 28.7 | 30.8 |
| QvQ-72B-Preview | 44.8 | 43.6 | 44.0 | 26.8 | 30.8 | 37.8 |
| Qwen2.5-VL-72B-Instruct | 53.4 | 46.9 | 58.0 | 25.8 | 29.5 | 42.3 |
| Qwen3-VL-30B-A3B | 64.1 | 56.4 | 72.0 | 28.7 | 35.3 | 50.7 |
| *Open Models (Llama-Based)* | | | | | | |
| Cambrian-8B | 31.3 | 24.2 | 36.0 | 24.0 | 29.0 | 28.9 |
| Llama-3.2-11B-Vision-Instruct | 31.0 | 30.8 | 39.0 | 21.1 | 26.2 | 29.4 |
| Llama-3.2-90B-Vision-Instruct | 45.0 | 23.2 | 43.0 | 26.3 | 31.5 | 34.1 |

*Table 2.* Performance (%) comparison of humans and selected models on VISUALPUZZLES. We report the best performance resulting from direct multiple-choice prompting and CoT prompting for each method. We highlighted all the reasoning models .

| Benchmark | # Knowledge Qs. |
|---|---|
| MMMU | 3.9 |
| VISUALPUZZLES | 1.1 |

*Table 3.* AVG. number of knowledge concept questions generated per instance on MMMU vs. VISUALPUZZLES.

90% knowledge accuracy on VISUALPUZZLES, most score below 60% on MMMU. Only the largest models approach 80% accuracy on MMMU, underscoring its heavier reliance on domain-specific knowledge.

**Does scaling up model size improve performance?** We plot reasoning accuracy (i.e., overall performance on the benchmark) in Figure 3, revealing some interesting trends:

- **MMMU.** Larger models tend to have higher knowledge accuracy, and this often translates into higher overall benchmark performance. This aligns with MMMU's re-

liance on domain-specific understanding; models with more parameters and training data are better at recalling relevant knowledge, thus improving their performance.

- **VISUALPUZZLES.** Although many models achieve near-100% knowledge accuracy on VISUALPUZZLES, we observe no clear increase in both knowledge and reasoning accuracy as model size grows. In contrast to MMMU, simply scaling number of parameters does not guarantee better performance on VISUALPUZZLES, implying that further gains on VISUALPUZZLES must stem from improvements in models' reasoning abilities rather than reliance on extensive knowledge.

**What is the relationship between knowledge and reasoning?** Figure 3 shows two scatter plots with trend lines that measure how knowledge accuracy correlates with rea-

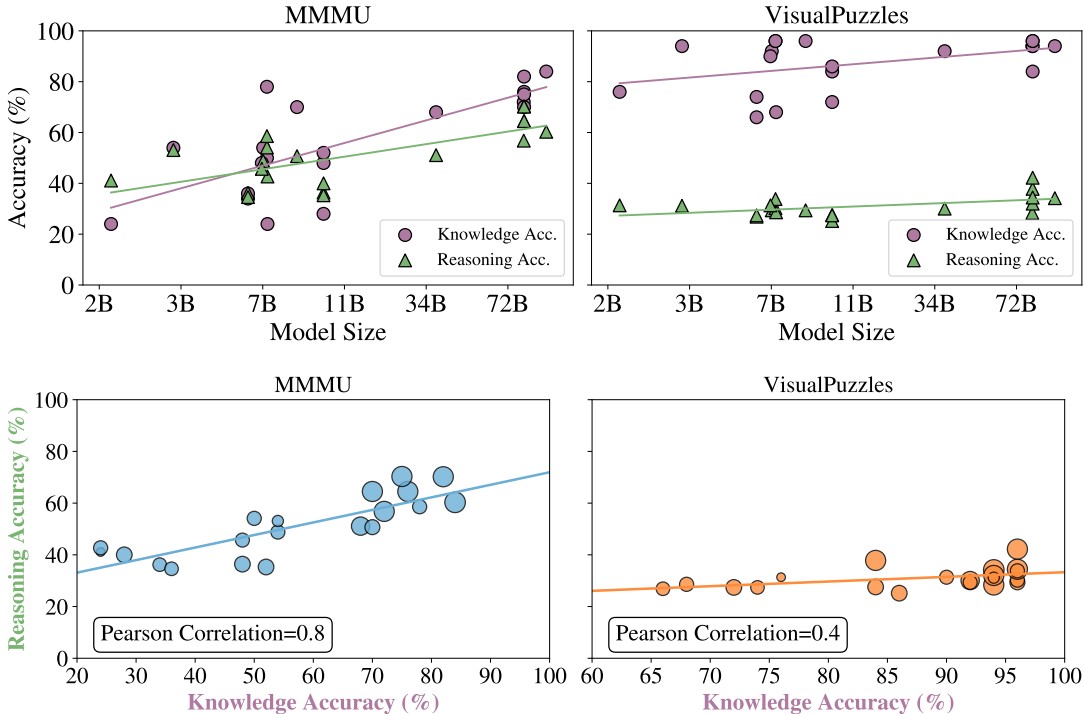

*Figure 3.* Scatter plots with fit lines of the trend between accuracy and model size (top) and that between reasoning and knowledge accuracy (bottom) on MMMU and VISUALPUZZLES. Dot sizes represent relative model sizes. The correlation between reasoning accuracy and knowledge accuracy is higher on MMMU (0.8) than on VISUALPUZZLES (0.4).

soning accuracy across different open models, where the relative sizes of the dots represent the sizes of the models. On MMMU (left), there is a strong positive correlation (0.8), suggesting that a model possessing more knowledge strongly correlates better reasoning performance. In contrast, VISUALPUZZLES (right) exhibits a more modest correlation (0.4). Although there is still an upward trend, gains in knowledge accuracy lead to smaller improvements in reasoning accuracy. This discrepancy implies that while overcoming knowledge gaps is central to reasoning success on MMMU, VISUALPUZZLES tasks demand more nuanced inference steps that depends less on domain knowledge.

Overall, these findings reinforce that VISUALPUZZLES's comparatively lower knowledge requirements are readily met by models. By contrast, MMMU poses a greater challenge to smaller models in terms of knowledge, for which scaling in size clearly benefits knowledge-intensive tasks. However, on VISUALPUZZLES, larger model size alone is not a decisive factor, which might imply that genuine multimodal reasoning depends on more than just number of parameters or pre-trained knowledge.

### 4.2. Reasoning Complexity of VISUALPUZZLES

**Does VISUALPUZZLES require more complex reasoning than existing benchmarks like MMMU?**

| Model | MMMU | VISUALPUZZLES |
|---|---|---|
| GPT-4o | 75.1% | 87.0% |
| Gemini-2.0-Flash | 67.9% | 77.3% |

*Table 4.* Percentage of logical reasoning steps in solving VISUALPUZZLES questions.

Besides observing that models generally achieve lower accuracy on VISUALPUZZLES compared to MMMU, we further investigated whether this gap stems from increased reasoning complexity. To do so, we measured the proportion of reasoning steps required to solve each question. We began by gathering detailed, step-by-step solutions from the models for each question, which are manually verified for completeness. Then we classified if each step is a logical reasoning step with the help of LLM. As shown in Table 4, on average logical reasoning steps take up 14.8% more total steps in solving VISUALPUZZLES questions compared to those of MMMU (82.1% v.s. 71.5%). Results suggest that VISUALPUZZLES demand more extensive reasoning, aligning with its goal of evaluating deeper multimodal reasoning beyond factual recall (prompt examples in Appendix G).

### 4.3. Do Reasoning Models Perform Better than Their Baselines?

Recent reasoning models often scale up inference compute by generating longer CoTs to enhance reasoning ability. To

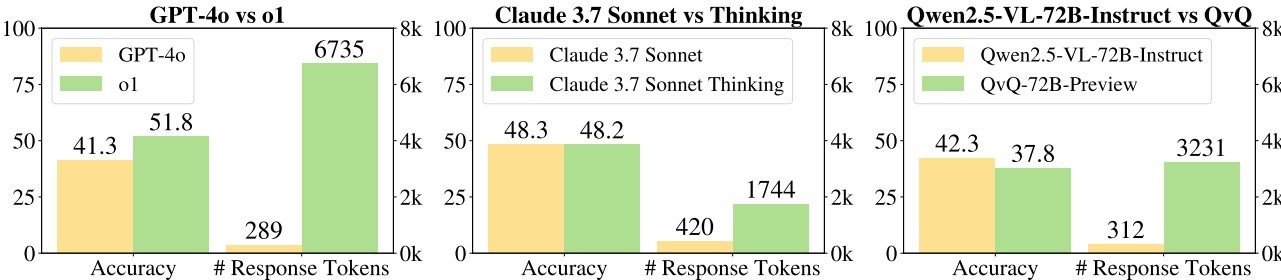

*Figure 4.* Comparison of accuracy and average number of total completion tokens of reasoning models and their general counterparts on VISUALPUZZLES. We didn't include Gemini-2.0-Flash models here because Gemini-2.0-Flash-Thinking does not reveal the number of reasoning tokens of responses. The accuracies of Gemini-2.0-Flash and Gemini-2.0-Flash-Thinking is 45.0% and 42.2% respectively. Despite much higher number of completion tokens, reasoning models do not often achieve better performance on VISUALPUZZLES.

assess the effectiveness of this strategy on VISUALPUZZLES, we compare several reasoning models with their non-reasoning counterparts in Figure 4. The reasoning model o1 outperforms GPT-4o overall. However, structured "thinking" modes, despite much higher number of completion tokens, show no consistent gain. This pattern persists even for the most recent models: on the latest Claude-4.5-Sonnet, enabling extended thinking improves overall accuracy by only 1% (49.4% with thinking vs. 48.4% without), consistent with our observation that scaling inference-time compute via extended thinking does not reliably yield substantial gains on VISUALPUZZLES. Similarity of output in Figure 13 further reveals that thinking modes primarily increase verbosity without meaningfully altering reasoning processes.

### 4.4. Are Branching and Revalidation Reasoning Patterns Effective on VISUALPUZZLES?

As discussed in § 4.3, reasoning-enabled models do not consistently outperform their non-reasoning counterparts on VISUALPUZZLES. To better understand this discrepancy, we examine Claude-3.7-Sonnet-Thinking's reasoning behaviors present in long CoTs, specifically, branching and re-validation, which are known to play important roles in enhancing reasoning performance[5]. In our analysis, *branching* refers to systematically exploring multiple reasoning paths, which is conceptually aligned with the notion of *Exploration* (Chen et al., 2025). Similarly, *revalidation* refers to re-assessing and correcting conclusions when errors are detected, corresponding closely to the notion of *Reflection* (Chen et al., 2025; Shinn et al., 2023). By adopting these standard definitions, we aim to strengthen the connection between our terminology and prior work on reasoning strategies in language models. As shown in Figure 5, our analysis reveals a striking contrast between benchmarks. On MMMU, both branching and re-validation correlate positively with model accuracy. These strategies help mod-

---

[5]We examined Claude-3.7-Sonnet as a case study, since it is the best model with explicitly thinking output.

| Benchmark | Answer-First | Option-First |
|---|---|---|
| MMMU | 29.3% | 70.7% |
| VISUALPUZZLES (Image Options) | 72.5% | 27.5% |
| VISUALPUZZLES (Text Options) | 98.3% | 1.7% |

*Table 5.* Answer Strategy of Claude-3.7-Sonnet-Thinking

els explore alternative reasoning paths and revisit earlier steps, aiding in the retrieval of relevant factual knowledge, an essential component for solving MMMU's knowledge-intensive questions. An example is provided in Appendix F.

Surprisingly, these reasoning behaviors are more frequent yet less predictive of success on VISUALPUZZLES. Despite increased presence in long responses, we observe no significant correlation between these strategies and task accuracy. This suggests that models may be using branching and re-validation in ways that do not contribute to problem solving. Figure 14 highlights this with an example from Claude-3.7-Sonnet-Thinking, where it applies branching on a VISUALPUZZLES question. However, the branched reasoning paths remain shallow and fail to engage with the core challenge—understanding the spatial arrangement of chairs (Full response in Appendix F).

## 5. Analysis

### 5.1. Do Models Approach VISUALPUZZLES Differently?

Table 5 shows statistics of Claude-3.7-Sonnet-Thinking's answering strategy. We observe clear divergence in answering strategies between MMMU and VISUALPUZZLES. On MMMU, the model tend to follow an option-driven approach—using the options early to eliminate unlikely answers and select the most relevant one, without explicitly solving the task. In contrast, the model more frequently adopt an answer-first strategy on VISUALPUZZLES, attempting to solve the question independently before comparing the result to the answer choices. This pattern holds across both textual and image-based options, though the option-first approach appears more often (around 30%) for image-

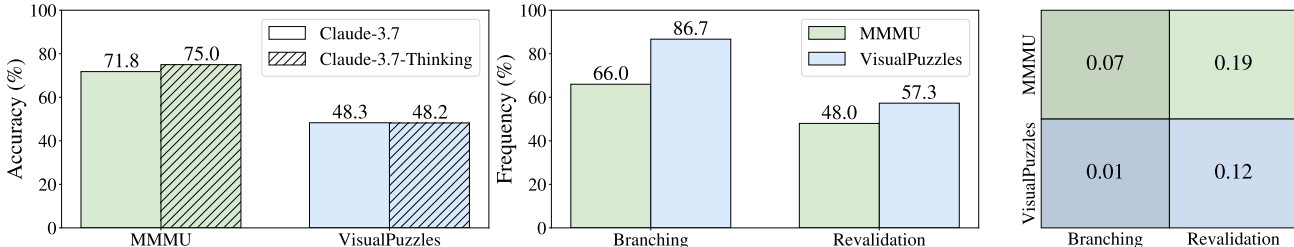

*Figure 5.* Comparison of Reasoning Pattern of Claude-3.7-Sonnet-Thinking on MMMU and VISUALPUZZLES. Left figure compares the accuracy of Claude-3.7-Sonnet and Claude-3.7-Sonnet-Thinking on MMMU and VISUALPUZZLES. Middle figure shows frequency of each pattern. Right figure shows correlation of the patterns with accuracy on the benchmarks.

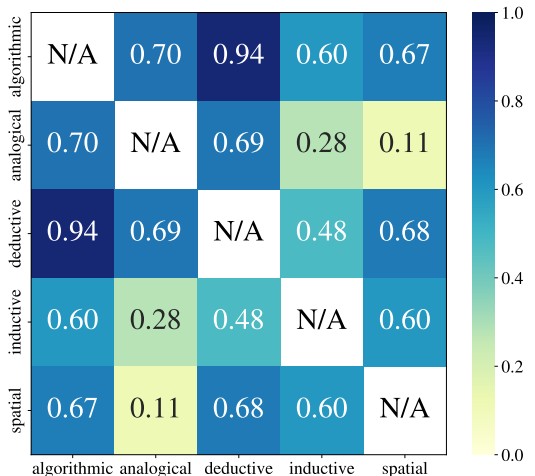

*Figure 6.* Model correlation heatmap among reasoning categories.

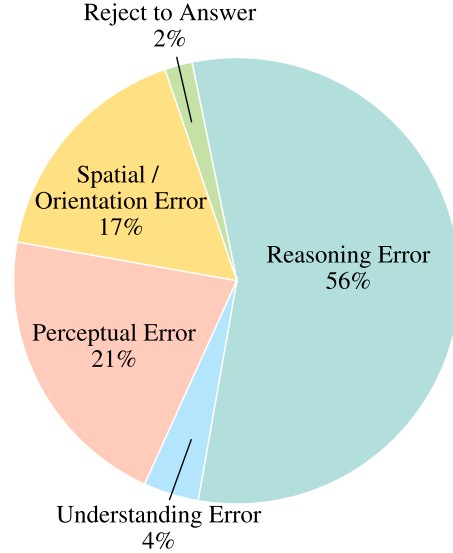

*Figure 7.* Error Distribution of Claude-3.7-Sonnet-Thinking

based tasks, likely due to the complexity of visual comparison (Liu et al., 2021; Song et al., 2025; Suhr et al., 2019). To test whether these spontaneous strategy preferences correspond to the strategy that is actually more effective on each benchmark, we further conduct a controlled experiment that explicitly enforces each strategy via prompting on Kimi-K2.5. The enforced results shift in the same direction as the observed preferences, i.e. Answer-First performs better on VISUALPUZZLES, whereas Option-First performs better on MMMU, though the gaps are modest (see § J.13 for more details).

### 5.2. Is there performance co-occurrence among reasoning categories?

Figure 6 presents a heatmap showing the correlation among the five reasoning categories in VISUALPUZZLES. We report correlations averaged across all models in Table 2. For humans, each category likely engages different cognitive processes (Babcock & Vallesi, 2015; Bright & Feeney, 2014; Goel & Dolan, 2004; Green et al., 2010), so performance in one category may not co-occur with performance in another (See Appendix J.6 for more discussion). However,

the correlation heatmap of the models tells a different story. We observe notably strong correlations across reasoning categories, with values ranging from 0.11 to as high as 0.94. In particular, algorithmic and deductive reasoning show high correlation (0.94), and other pairs such as algorithmic-analogical and deductive-analogical also exhibit strong associations. This suggests that model performance tends to generalize across categories. However, this generalization may not reflect true reasoning abilities. Instead, the high correlations could indicate that models are leveraging shared surface-level patterns or shortcut strategies that happen to work across multiple structurally different categories, unlike humans, who may rely on distinct cognitive processes.

### 5.3. Error Analysis

Figure 7 is a pie chart showing the error category distribution of Claude-3.7-Sonnet-Thinking on 100 randomly selected instances from VISUALPUZZLES. Reasoning errors dominate at 56%, reinforcing the fact that reasoning is the greatest challenge in VISUALPUZZLES. Perceptual

errors (21%) and spatial / orientation errors (17%) also constitute substantial portions of failures, reflecting difficulties in interpreting visual elements and understanding spatial relationships. These three categories together account for 94% of mistakes, emphasizing a need for multimodal models with stronger reasoning capabilities with more robust perception and spatial understanding. Textual and visual understanding errors (4%) and reject-to-answer cases (2%) are relatively rare. Appendix M shows samples of error and correct cases of each reasoning and difficulty category.

## 6. Conclusion

We presented VISUALPUZZLES, a novel and complex multimodal reasoning benchmark carefully designed to minimize requirement of domain-specific knowledge. Our results show that while proprietary and large-scale open models achieve relatively higher performance, they still fall short of human-level reasoning—especially on more complex tasks such as analogical and inductive reasoning. Moreover, we observe that strong performance on knowledge-intensive benchmarks like MMMU does not necessarily translate into high accuracy on VISUALPUZZLES, underscoring the distinct challenge of knowledge-light reasoning tasks. Our findings also suggest that purely scaling inference compute, model size and knowledge resources may not suffice for robust multimodal reasoning skills.

By disentangling domain knowledge from multimodal reasoning, we hope VISUALPUZZLES will support the development and evaluation of next-generation MLLMs that excel at genuinely understanding and reasoning about the world without depending heavily on specialized knowledge.

## Impact Statement

**Potential Broader Impact.** In this work, we introduce VISUALPUZZLES, a benchmark designed to evaluate multimodal reasoning capabilities while minimizing reliance on domain-specific knowledge. By decoupling multimodal reasoning and factual recall of domain knowledge, we aim to evaluate multimodal reasoning more accurately, instead of measuring how much knowledge models memorize. With this motivation, VISUALPUZZLES could potentially benefit the broader research community by offering a clear and accurate evaluation of progress in multimodal reasoning of the research community, particularly for models that are designed or trained to be used under a knowledge-light but reasoning-heavy scenario. Additionally, with VISUALPUZZLES as an evaluation suite, we hope that VISUALPUZZLES could be able to guide the training strategies or architecture designs targeting multimodal reasoning rather than scale alone. On the other hand, although VISUALPUZZLES minimizes reliance on domain knowledge, it still assumes that

models possess basic commonsense knowledge, and thus does not represent completely knowledge-free evaluation.

**Future Societal Consequences.** Beyond research in multimodal reasoning abilities of models, diagnostic evaluation suites like VISUALPUZZLES could potentially contribute to the development of safer and more reliable multimodal systems in a knowledge-light but reasoning-heavy settings.

**Ethical Aspects.** This paper uses samples extracted from existing quiz sources for scholarly analysis and testing purposes, in accordance with US fair use law and standard practice. These data are neither intended for, nor capable of, substituting for the original works; thus, we believe their inclusion does not diminish the market value or utility of the source materials. A complete list of references for the data sources is attached in Appendix B.

We took several steps to enable independent verification of our results. The dataset design, curation pipeline, validation procedures, and attribute annotation are described in § 2. Our experimental setup, including model families evaluated and human study protocol, is summarized in § 3. The appendices contain the materials needed to replicate analyses.

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

# A. Use of LLMs

We employed Large Language Models (LLMs) to assist in polishing the style, clarity, and presentation of text and tables throughout the paper and appendix. This included refining phrasing, improving readability, standardizing terminology, and ensuring consistency in table formatting and captions. All content, analysis, and results were generated and verified by the authors; LLMs were used solely as a writing aid and did not influence the underlying data or experimental findings.

# B. VISUALPUZZLES Statistics

## B.1. Breakdown of Statistics of VISUALPUZZLES

Table 6 shows a breakdown of statistics of VISUALPUZZLES questions.

| Reasoning Category | Image Options | | | Text Options | | | Total |
|---|---|---|---|---|---|---|---|
| | Easy | Medium | Hard | Easy | Medium | Hard | |
| Algorithmic | 21 | 8 | 0 | 124 | 100 | 9 | 262 |
| Analogical | 120 | 81 | 10 | 0 | 0 | 0 | 211 |
| Deductive | 29 | 24 | 2 | 45 | 79 | 21 | 200 |
| Inductive | 7 | 70 | 127 | 3 | 2 | 0 | 209 |
| Spatial | 123 | 41 | 6 | 61 | 52 | 3 | 286 |
| Total | 300 | 224 | 145 | 233 | 233 | 33 | 1168 |

*Table 6.* Number of questions in each reasoning category, option types, and difficulty levels.

## B.2. Data Sources

- Chinese Civil Service Examination (中国国家公务员考试) [6] (224 puzzles): we manually translated questions from this exam to English from Chinese.

- Textbooks (210 puzzles): we carefully collected and re-purposed questions from online resources and textbooks.

- Smart-101 (Cherian et al., 2023) (247 puzzles): we carefully selected images from this benchmark and synthesized new questions.

- MATH-Vision (Wang et al., 2024a) (293 puzzles): we carefully selected and re-purposed questions from this benchmark.

- VASR (Bitton et al., 2023) (194 puzzles): we carefully selected questions from this benchmark.

# C. Model Evaluation Setup

> **Model Evaluation Prompt with Chain-of-Thought**
>
> Solve the multiple-choice question and then answer with the option letter from the given choices. The last line of your response should be of the following format: 'Answer: $LETTER' (without quotes) where LETTER is one of options. Think step by step before answering.

> **Model Evaluation Prompt w/n Chain-of-Thought**
>
> Answer the question with the option's letter from the given choices directly.

**Experiments** We integrated VISUALPUZZLES into Lmms-eval (Li* et al., 2024; Zhang et al., 2024). We used 8 H100 (80GB) GPUs for experiments. However, one should be able to use less number of GPUs to reproduce all the experiments we did, depending on the size of the model and GPU memories.. We set all hyper-parameters to default in Lmms-eval, with the maxmium number of completion tokens be 16,000.

---

[6] https://en.wikipedia.org/wiki/Civil_service_of_the_People%27s_Republic_of_China#Examinations.

## C.1. Correlation Analysis

Earlier models generally exhibited weaker reasoning abilities compared to more recent ones. To better understand the relationship between model size and performance, we conducted correlation analyses across two benchmarks: MMMU and VISUALPUZZLES, the results of which are shown in Table 7.

Restricting our analysis to the Qwen model family, we find a strong correlation between model size and accuracy on MMMU ($r = 0.93$), whereas the correlation with VISUALPUZZLES is notably lower ($r = 0.64$).

To further control for the potential confounding effect of release date, we divided all models into two cohorts: those released prior to August 1, 2024, and those released afterwards. The correlations between model size and accuracy remain consistently higher on MMMU ($r = 0.75$ pre-t, $r = 0.89$ post-t) compared to VISUALPUZZLES ($r = 0.49$ pre-t, $r = 0.58$ post-t).

These results suggest that, even within a single release cohort, VISUALPUZZLES is less sensitive to model size than MMMU. This highlights the distinctive evaluation focus of VISUALPUZZLES, which emphasizes reasoning over sheer parameter scale.

*Table 7.* Correlation between model size and benchmark accuracy.

| Model Family / Cohort | MMMU (Correlation with Size) | VISUALPUZZLES (Correlation with Size) |
|---|---|---|
| Qwen Models | 0.93 | 0.64 |
| Models Released Prior to 2024-08-01 | 0.75 | 0.49 |
| Models Released After 2024-08-01 | 0.89 | 0.58 |

# D. Human Annotation Setup

## D.1. Difficulty Labeling

Each question was also carefully assigned a difficulty label from easy, medium, or hard, based on the cognitive load required for reasoning.

- **Easy Level** questions could be solved by the annotator in less than one minute.

- **Medium Level** questions could be solved by the annotator in one to three minutes.

- **Hard Level** questions require the annotator more than five minutes to solve or quit solving.

---

**Annotation Guideline for Puzzle Difficulty**

Try to solve the puzzle first. You need to measure the time you attempted to solve each puzzle. Then, select from Easy, Medium, or Hard based on the time required.
- Easy Level: You can solve or answer the question within 1 minute. This level of puzzles should require minimal reasoning.
- Medium Level: You can solve or answer the question within 1-3 minutes. This level of puzzles should demand moderate reasoning.
- Hard Level: You can / cannot solve this question with more than 5 minutes. This level of puzzles should involve significant / multi-step reasoning.

---

## D.2. Reasoning Category Labeling

---

**Annotation Guideline for Puzzle Reasoning Category**

Assign the category that *best* describes the primary type of reasoning or logic required for each puzzle:
- Algorithmic Reasoning: Involves following or devising a step-by-step procedure or rule-based process.
- Analogical Reasoning: Requires identifying relationships by comparison between pairs of entities.
- Deductive Reasoning: Involves deriving specific conclusions from general or given premises.
- Inductive Reasoning: Focuses on generalizing a rule or pattern from specific instances.
- Spatial Reasoning: Involves visualizing and manipulating shapes, distances, or orientations.

You should first decide all potential labels for each puzzle, then select a single **primary** label using a fixed rubric:

---

(i) choose the category whose absence makes the puzzle unsolvable;
(ii) if two or more remain, pick the more specific one (priority order: Spatial = Algorithmic = Analogical > Inductive = Deductive);
(iii) resolve any tie by majority vote.

Each question is labeled by three annotators for the reasoning category. Around 4% of the questions, annotators disagree on the primary reasoning category, and either the annotators reach consensus through discussion or the puzzles are removed.

### D.3. Reasoning Categories

Reasoning methods are commonly categorized into *analogical reasoning*, *deductive reasoning*, and *inductive reasoning*. At an abstract level, we adopt this categorization. However, during annotation, we observed that mistakes regarding *spatial reasoning* were particularly prevalent. For this reason, we determined that spatial reasoning merited its own category.

### D.4. Annotation Protocol

Annotators followed a two-step process for labeling each puzzle:

1. **Candidate Labels:** Annotators first identified all potential reasoning categories applicable to a given puzzle.

2. **Primary Label Selection:** Annotators then selected a single *primary* label using the fixed decision rubric described below.

Decision rubric for selecting the primary reasoning label:

1. Choose the category whose absence makes the puzzle unsolvable.

2. If two or more remain, pick the more specific one.
   *Priority order:* Spatial = Algorithmic = Analogical > Inductive = Deductive.

3. If a tie still remains, resolve by majority vote among annotators.

This hierarchical rubric ensured consistency across annotators and reduced ambiguity when multiple reasoning strategies appeared relevant.

### D.5. Error Analysis

In Figure 7, two annotators conducted an error analysis on 100 randomly sampled model outputs, following the guidelines below. 30 samples were co-annotated by both annotators, yielding an inter-annotator agreement of 86.7%. Remaining disagreements were resolved through discussion to reach consensus.

---

**Annotation Guideline for Identifying Model Errors**

Error Annotation Guideline
For each incorrect model answer, assign one primary label from:
- Reasoning Error
- Perceptual Error
- Spatial / Orientation Error
- Understanding Error
- Reject to Answer
Annotators first read the question and model reasoning, then choose the label that best explains the mistake.
1. Reasoning Error (RE)
Definition: Model perceives the scene correctly but applies a wrong rule, makes an invalid inference, or fails to combine relations correctly.
2. Perceptual Error (PE)
Definition: Model miss or wrongly perceive key visual details, so later reasoning is based on an incorrect representation.
3. Spatial / Orientation Error (SE)
Definition: Model recognizes the correct objects but fails on their positions, rotations, or relative layout.
Borderline rule:
- If the object itself is mis-identified -¿ Perceptual Error.
- If the object is identified correctly but its rotation/placement is wrong -¿ Spatial / Orientation Error.

---

4. Understanding Error (UE)
Definition: Model misinterprets the instructions or answer format, not the image content or logical rule.
5. Reject to Answer (RA)
Definition: Model explicitly declines to solve the problem or states it cannot answer.

# E. Full Results

## E.1. Full Results w/ CoT

## E.2. Full Results w/n CoT

| Model | Algorithmic | Analogical | Deductive | Inductive | Spatial | Overall |
|---|---|---|---|---|---|---|
| Random Choice | 25.0 | 25.0 | 25.0 | 25.0 | 25.0 | 25.0 |
| Human (95th Percentile) | 100.0 | 100.0 | 100.0 | 81.6 | 100.0 | 89.3 |
| Human (50th Percentile) | 88.0 | 66.0 | 80.0 | 50.0 | 90.0 | 75.0 |
| Human (5th Percentile) | 68.1 | 25.0 | 37.0 | 0.0 | 59.1 | 57.5 |
| *Proprietary Models* | | | | | | |
| o4-mini | 65.3 | 68.7 | 75.5 | 33.0 | 45.5 | 57.0 |
| o3 | 64.5 | 68.3 | 69.5 | 27.3 | 42.7 | 54.0 |
| o1 | 63.7 | 68.3 | 67.5 | 29.2 | 34.3 | 51.8 |
| GPT-4o | 49.2 | 58.3 | 49.0 | 27.3 | 26.2 | 41.3 |
| Gemini-3-pro | 67.6 | 60.7 | 49.0 | 34.0 | 49.7 | 52.7 |
| Gemini-2.5-pro | 60.0 | 64.0 | 60.0 | 29.7 | 36.4 | 49.5 |
| Gemini-2.0-flash | 55.3 | 58.8 | 57.0 | 24.4 | 31.8 | 45.0 |
| Gemini-2.0-flash-thinking | 46.6 | 70.1 | 49.0 | 24.9 | 25.5 | 42.2 |
| Gemini-1.5-Pro | 53.4 | 57.4 | 58.5 | 26.3 | 32.5 | 45.0 |
| Claude-4.6-Opus-Thinking | 72.9 | 58.3 | 76.5 | 23.9 | 39.2 | 53.9 |
| Claude-4.5-Opus-Thinking | 67.6 | 59.7 | 71.0 | 28.2 | 37.8 | 52.4 |
| Claude-4.5-Sonnet-Thinking | 67.2 | 51.2 | 69.0 | 27.8 | 33.9 | 49.4 |
| Claude-4-Sonnet-Thinking | 64.9 | 46.5 | 60.5 | 25.8 | 38.1 | 47.3 |
| Claude-3.7-Sonnet | 64.5 | 48.3 | 65.0 | 26.8 | 37.4 | 48.3 |
| Claude-3.7-Sonnet-thinking | 67.2 | 44.1 | 61.5 | 31.1 | 37.1 | 48.2 |
| Claude-3.5-Sonnet | 53.4 | 47.9 | 51.5 | 25.4 | 34.3 | 42.4 |
| Kimi-K2.5 | 62.2 | 56.9 | 61.0 | 26.8 | 48.6 | 51.4 |
| *Open Models* | | | | | | |
| LLaVA-1.5-7B | 23.3 | 21.8 | 36.0 | 20.6 | 19.2 | 23.7 |
| LLaVA-1.5-13B | 24.8 | 21.8 | 23.0 | 25.4 | 25.5 | 24.2 |
| LLaVA-1.6-7B | 27.5 | 23.7 | 30.0 | 22.5 | 21.3 | 24.8 |
| LLaVA-1.6-13B | 25.2 | 25.6 | 27.0 | 27.3 | 23.4 | 25.5 |
| LLaVA-1.6-34B | 29.4 | 28.0 | 43.0 | 24.9 | 25.5 | 29.7 |
| LLaVA-OV-0.5B | 21.0 | 26.1 | 30.5 | 22.5 | 25.2 | 24.8 |
| LLaVA-OV-7B | 27.9 | 26.1 | 36.5 | 23.4 | 25.5 | 27.7 |
| LLaVA-OV-72B | 34.7 | 26.5 | 37.0 | 27.3 | 28.7 | 30.8 |
| Llama-3.2-11B-Vision-Instruct | 31.0 | 30.8 | 39.0 | 21.1 | 26.2 | 29.4 |
| Llama-3.2-90B-Vision-Instruct | 45.0 | 23.2 | 43.0 | 26.3 | 31.5 | 34.1 |
| Qwen-VL | 21.4 | 31.3 | 25.0 | 26.3 | 24.1 | 25.3 |
| Qwen2-VL-72B | 41.6 | 28.4 | 39.5 | 22.5 | 29.0 | 32.4 |
| QvQ-72B-Preview | 43.1 | 45.5 | 48.0 | 27.3 | 27.6 | 37.8 |
| Qwen2-VL-2B-Instruct | 26.0 | 26.1 | 24.5 | 27.8 | 25.5 | 26.0 |
| Qwen2-VL-7B-Instruct | 36.3 | 21.8 | 38.5 | 20.6 | 22.7 | 27.9 |
| Qwen2-VL-72B-Instruct | 39.9 | 33.5 | 45.2 | 23.5 | 32.4 | 34.9 |
| Qwen2.5-VL-3B-Instruct | 35.1 | 27.5 | 44.5 | 25.8 | 24.8 | 31.2 |
| Qwen2.5-VL-7B-Instruct | 40.5 | 26.6 | 39.0 | 24.0 | 29.7 | 32.1 |
| Qwen2.5-VL-72B-Instruct | 53.4 | 46.9 | 58.0 | 25.8 | 29.5 | 42.3 |
| Qwen3-VL-30B-A3B | 64.1 | 56.4 | 72.0 | 28.7 | 35.3 | 50.7 |
| Cambrian-8B | 31.3 | 24.2 | 36.0 | 24.0 | 29.0 | 28.9 |
| Cambrian-13B | 24.8 | 25.6 | 39.5 | 24.4 | 21.0 | 26.5 |
| Pangea-7B | 30.5 | 28.9 | 35.0 | 24.4 | 25.2 | 28.6 |

*Table 8.* Performance (%) of various models with Chain of Thoughts (CoT) on VISUALPUZZLES.

| Model | Algorithmic | Analogical | Deductive | Inductive | Spatial | Overall |
|---|---|---|---|---|---|---|
| Random Choice | 25.0 | 25.0 | 25.0 | 25.0 | 25.0 | 25.0 |
| Human (95th Percentile) | 100.0 | 100.0 | 100.0 | 81.6 | 100.0 | 89.3 |
| Human (50th Percentile) | 88.0 | 66.0 | 80.0 | 50.0 | 90.0 | 75.0 |
| Human (5th Percentile) | 68.1 | 25.0 | 37.0 | 0.0 | 59.1 | 57.5 |
| *Proprietary Models* | | | | | | |
| GPT-4o | 40.8 | 34.1 | 40.5 | 24.9 | 29.7 | 34.0 |
| Gemini-2.0-flash | 57.6 | 41.7 | 58.0 | 23.0 | 35.7 | 43.2 |
| Gemini-1.5-Pro | 51.2 | 46.5 | 54.0 | 24.9 | 29.4 | 40.8 |
| *Open Models* | | | | | | |
| LLaVA-1.5-7B | 24.4 | 24.7 | 34.5 | 26.8 | 25.5 | 26.9 |
| LLaVA-1.5-13B | 24.4 | 26.1 | 33.5 | 26.3 | 28.3 | 27.6 |
| LLaVA-1.6-7B | 27.5 | 25.1 | 32.5 | 24.9 | 27.3 | 27.4 |
| LLaVA-1.6-13B | 21.4 | 24.7 | 29.5 | 28.2 | 23.1 | 25.0 |
| LLaVA-1.6-34B | 31.3 | 27.3 | 43.0 | 24.4 | 27.6 | 29.8 |
| LLaVA-OV-0.5B | 24.4 | 25.6 | 37.5 | 24.9 | 25.5 | 27.2 |
| LLaVA-OV-7B | 27.5 | 28.0 | 40.5 | 24.4 | 28.0 | 29.4 |
| LLaVA-OV-72B | 31.7 | 23.6 | 45.0 | 21.3 | 24.6 | 28.8 |
| Llama-3.2-11B-Vision-Instruct | 27.5 | 24.2 | 31.0 | 26.3 | 27.6 | 27.3 |
| Llama-3.2-90B-Vision-Instruct | 38.2 | 22.3 | 44.5 | 25.8 | 33.6 | 33.1 |
| Qwen-VL | 23.7 | 26.5 | 29.5 | 27.8 | 26.6 | 26.6 |
| Qwen2-VL-72B | 38.9 | 28.4 | 43.0 | 20.6 | 29.0 | 32.0 |
| QvQ-72B-Preview | 44.8 | 43.6 | 44.0 | 26.8 | 30.8 | 37.8 |
| Qwen2-VL-2B-Instruct | 31.7 | 29.4 | 40.5 | 23.9 | 31.5 | 31.3 |
| Qwen2-VL-7B-Instruct | 33.6 | 24.2 | 46.0 | 22.5 | 26.2 | 30.2 |
| Qwen2-VL-72B-Instruct | 40.5 | 30.3 | 46.0 | 25.4 | 29.4 | 34.2 |
| Qwen2.5-VL-3B-Instruct | 36.3 | 26.1 | 47.0 | 25.8 | 22.4 | 31.0 |
| Qwen2.5-VL-7B-Instruct | 38.2 | 23.7 | 51.5 | 24.9 | 31.1 | 33.7 |
| Qwen2.5-VL-72B-Instruct | 43.1 | 40.3 | 51.5 | 25.4 | 33.7 | 38.6 |
| Cambrian-8B | 25.2 | 20.4 | 35.0 | 23.0 | 20.6 | 24.5 |
| Cambrian-13B | 23.3 | 28.0 | 36.5 | 24.9 | 26.2 | 27.4 |
| Pangea-7B | 32.4 | 23.7 | 38.5 | 28.7 | 32.5 | 31.3 |

*Table 9.* Performance (%) of various models with Multiple Choice Direct prompting on VISUALPUZZLES.

# F. Knowledge Checklist

## F.1. Knowledge Checklist Generation

---

**Prompt to Generate Knowledge Checklist Questions**

You are an exam writer. You are now writing a knowledge test. You are given a question (Question) regarding an image and its standard solution (Solution), your task is to write free response questions that test on individual knowledge required in answering the question correctly.

You should follow these steps to complete the task:
1. explicitly analyze the given image, Question, and Solution
2. explicitly list out the individual knowledge concepts required to reach Solution.
3. write free response questions to test on the definition of each concept listed. Your generated questions should not include details of the given Question. Note that you need to provide answer keys to these questions too.
4. format the free response questions in json format.

Question: question
Solution: answer

---

## F.2. Knowledge Checklist Construction

We adopt a structured process for constructing a *knowledge checklist*, which enumerates the atomic facts that a human or model must know before engaging in reasoning on a given benchmark instance.

- **LLM-based list generation and manual verification:** Large Language Models (LLMs) are first used to compile a candidate list of atomic facts. Each fact is expressed as a QA pair. For example:

  *Q: Explain the Arbitrage Pricing Theory (APT) model and its purpose in finance. A: The Arbitrage Pricing Theory (APT) model is a financial theory that estimates the expected return . . .*

- **Human verification:** Two annotators independently review each checklist to ensure correctness, self-containment, and comprehensiveness of the QA pairs.

## F.3. Evaluation Protocol

- **Model evaluation:** Models are evaluated on the knowledge checklist questions, with correctness judged by an *LLM-as-a-judge* approach.

- **Knowledge accuracy calculation:** We define knowledge accuracy as the percentage of benchmark instances for which a model answers *all* checklist questions correctly.

## F.4. Example Knowledge Checklist Question

---

**Example Knowledge Checklist Question (MMMU)**

- Question: Explain the Arbitrage Pricing Theory (APT) model and its purpose in finance.
- Answer: The Arbitrage Pricing Theory (APT) model is a financial theory that estimates the expected return on an asset based on the asset's sensitivity to various macroeconomic factors. It is used to determine the fair price of an asset by considering multiple factors that could affect its return, as opposed to relying on a single market index as in the Capital Asset Pricing Model (CAPM).

---

**Example Knowledge Checklist Question (VISUALPUZZLES)**

- Question: What is the definition of distance in a geometric context?
- Answer: Distance in a geometric context refers to the measurement of space between two points.

---

## F.5. Knowledge Checklist Human Annotation

We asked two human annotators to manually verify and correct the knowledge checklist questions and gave them the following instructions. The inter-annotator agreement rate is 87.8%.

---

**Human Annotation Instructions**

You are given a json file, where each item contains the following elements:
- Question: a multiple-choice question.
- Answer: the answer to the question with an optional explanation.
- Knowledge Concept Checklist: a list of question-answer pairs, where each question in the list is intended to represent a distinct knowledge concept necessary for solving the Question.

You task is to annotate the knowledge concept checklists generated by a model. You should carefully evaluate each question-answer pair based on the following criteria:
1. Necessity: Is the question genuinely necessary for solving the problem? If not, then remove the question.
2. Repetition: Check if any questions are repetitive or duplicate existing questions within the list. If the question is repetitive or duplicate, then remove the question.
3. Completeness: Ensure no critical knowledge concepts required to solve the problem are missing, and identify if any additional important questions should have been included.
4. Correctness: Verify whether the provided answers are accurate. Revise the checklist in case of incorrect checklist QA pairs.
5. Knowledge v.s. Skills: Ensure each question explicitly evaluates a knowledge concept rather than testing skills or problem-solving techniques. Remove any questions that primarily evaluate skills instead of knowledge.

---

# G. Reasoning Complexity

**Instruction Prompt to Solve Questions in Detailed Steps**

$< Question >< Image >$
Solve this question with First Order Logic. Write out each thinking step explicitly, do not skip steps.
In your response, begin each step with $\_\_\_STEP\_START\_\_\_$
step $< step\_num >$

---

# H. Related Work

**Multimodal Language Models (MLLMs)**, particularly vision language models have experienced significant improvements recently. Large scale vision language models, including open weight ones are capable of utilizing both image and text inputs to solve challenging questions (Anthropic, 2023; Grattafiori et al., 2024; Gemini Team et al., 2023; Khanuja et al., 2024; Li et al., 2025; Liu et al., 2025; Hurst et al., 2024; Tong et al., 2024; Yue et al., 2025a). Multimodal reasoning models, models that specialize in complex reasoning, further push the boundary of MLLMs' capabilities. Large scale multimodal reasoning models such as QVQ (Qwen Team, 2024), Claude-3.7-Sonnet-thinking (Anthropic, 2023), o1 (Jaech et al., 2024), Gemini-2.0-flash-thinking (Gemini Team et al., 2023) excel in reasoning heavy tasks such as coding and solving math problems.

**Multimodal Reasoning Benchmarks**. There exists a number of multimodal benchmarks that test on both the models' world knowledge and reasoning abilities. These benchmarks emphasize on the multimodal ability of models as a whole, without further separation of knowledge and reasoning (Phan et al., 2025; Liu et al., 2023b; Marino et al., 2019; Yue et al., 2024; 2025b; Jiang et al., 2025). Recently, more multimodal benchmarks have placed emphasis on multimodal logical reasoning abilities. Many of them focus primarily on mathematic problems, testing on both mathematical knowledge and reasoning (Liu et al., 2021; Lu et al., 2024; Wang et al., 2024b; Suhr et al., 2019). Some others cover on more general logical reasoning problems, testing on both models' knowledge and reasoning in different domains (Cherian et al., 2023; Gao et al., 2023; Huang et al., 2025).

# I. Comparison with Other Benchmarks

## I.1. Comparison with Non Puzzle-Type Benchmarks

Figure 8 provides a comparative analysis between VISUALPUZZLES and several widely-used benchmarks for multimodal reasoning, visualizing the knowledge requirement and reasoning complexity of each benchmark. VISUALPUZZLES has high reasoning complexity and low knowledge requirement, with an aim to disentangle multimodal reasoning from domain-specific knowledge to evaluate general reasoning abilities in non-expert settings.

Table 11 compare the performance of various model families across MathVista, MMMU, and VISUALPUZZLES. Both MathVista and MMMU are benchmarks that have a heavy emphasis on both knowledge and reasoning, whereas VISUALPUZ-

| Dataset | Size | Reasoning Load | Knowledge Requirement | % Easy Words | Question Type | Answer Type |
|---|---|---|---|---|---|---|
| LogiQA | 0.7K | Heavy | Light | 52.0 | Text | Text |
| GSM8K | 8.5K | Heavy | Heavy | 54.0 | Text | Text |
| WikiDiverse | 0.8K | Light | Heavy | 35.8 | Image+Text | Text |
| MathVista | 6.1K | Heavy | Heavy | 51.9 | Image+Text | Text |
| MMMU | 11.5K | Heavy | Heavy | 46.4 | Image+Text | Text |
| MATH-Vision | 3.0K | Heavy | Heavy | 53.8 | Image+Text | Image+Text |
| MathVerse | 2.6K | Heavy | Heavy | 38.2 | Image+Text | Text |
| LogicBench | 1.5K | Heavy | Light | 53.6 | Text | Text |
| LogicVista | 0.4K | Heavy | Heavy | 41.2 | Image+Text | Image |
| NaturalBench | 10K | Light | Light | 52.5 | Image+Text | Text |
| VISUALPUZZLES | 1.2K | Heavy | Light | **54.1** | Image+Text | Image+Text |

*Table 10.* Comparison of other existing benchmarks with VISUALPUZZLES

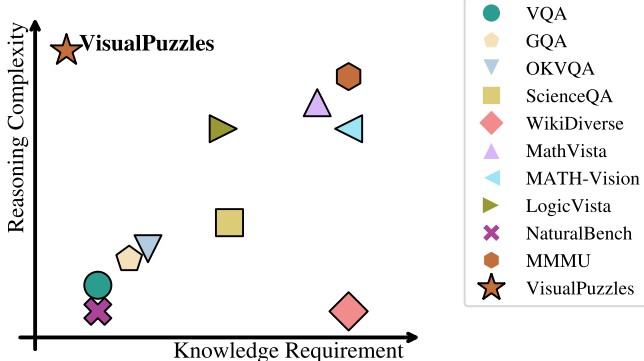

*Figure 8.* Comparison between VISUALPUZZLES and several widely-used benchmarks.

ZLES assess models on domain-disentangled multimodal reasoning alone. We found that success on knowledge-intensive multimodal reasoning benchmarks as MathVista and MMMU does not always carry over to VISUALPUZZLES that emphasize reasoning rather than extensive pre-trained knowledge.

### I.2. Related Puzzle-Type Benchmarks

There exists a large body of work on puzzle-type reasoning (Ji et al., 2022; Chen et al., 2024; Tang et al., 2025; **?**; Cheng et al., 2025). Our purpose in this paper is to assess MLLM's multimodal reasoning abilities *disentangled from domain knowledge*, a gap not fully addressed by existing benchmarks. VISUALPUZZLES is designed to fill this gap.

Specifically:

- **KiloGram** (Ji et al., 2022) focuses on tangram-based visual reasoning. In contrast, VISUALPUZZLES evaluates a broad variety of reasoning types with minimized domain knowledge.

- **M3CoT** (Chen et al., 2024) targets domain-specific visual reasoning and requires substantial external knowledge, whereas VISUALPUZZLES remains knowledge-light.

- **LEGO-Puzzles** (Tang et al., 2025) emphasize spatial reasoning, while VISUALPUZZLES evaluates five reasoning categories, including but not limited to spatial reasoning.

- **PuzzleVQA** (**?**) emphasizes abstract pattern recognition with limited reasoning complexity, while VISUALPUZZLES covers diverse and complex logical reasoning.

| Model | MathVista | MMMU | VISUALPUZZLES |
|---|---|---|---|
| Human | 60.3 | 88.6 | 80.1 |
| o1 | 73.9 | 78.2 | 51.8 |
| GPT-4o | 63.8 | 69.1 | 41.1 |
| Gemini-2.0-Flash | - | 71.7 | 45.0 |
| Gemini-1.5-Pro | 63.9 | 62.2 | 45.4 |
| Claude-3.5-Sonnet | 67.7 | 68.3 | 42.4 |
| Claude-3.7-Sonnet | - | 71.8 | 48.3 |
| Claude-3.7-Sonnet (Thinking) | - | 75.0 | 48.3 |
| LLaVA-1.5-7B | - | 36.2 | 26.9 |
| LLaVA-1.5-13B | 27.6 | 36.4 | 27.6 |
| LLaVA-NeXT-7B | 35.8 | 34.6 | 27.4 |
| LLaVA-NeXT-13B | 36.2 | 35.3 | 25.3 |
| LLaVA-NeXT-34B | 46.5 | 51.1 | 29.8 |
| LLaVA-OV-0.5B | 34.8 | 31.4 | 27.2 |
| LLaVA-OV-7B | 63.2 | 48.8 | 29.4 |
| LLaVA-OV-72B | 67.5 | 56.8 | 31.8 |
| Llama-3.2-11B-Vision-Instruct | 51.5 | 50.7 | 29.4 |
| Llama-3.2-90B-Vision-Instruct | 57.3 | 60.3 | 34.3 |
| Qwen2-VL-72B | 70.5 | 64.5 | 32.1 |
| QvQ-72B-Preview | 71.4 | 70.3 | 37.9 |
| Qwen2-VL-2B-Instruct | 43.0 | 41.1 | 31.3 |
| Qwen2-VL-7B-Instruct | 58.2 | 54.1 | 30.2 |
| Qwen2-VL-72B-Instruct | 70.5 | 64.5 | 34.9 |
| Qwen2.5-VL-3B-Instruct | 62.3 | 53.1 | 31.2 |
| Qwen2.5-VL-7B-Instruct | 68.2 | 58.6 | 33.7 |
| Qwen2.5-VL-72B-Instruct | 74.8 | 70.2 | 42.3 |
| Cambrian-8B | 49.0 | 42.7 | 28.5 |
| Cambrian-13B | 48.0 | 40.0 | 27.4 |

*Table 11.* Comparison of other MathVista and MMMU with VISUALPUZZLES on human and SOTA models

- **CoMT** (Cheng et al., 2025) examines failures in maintaining coherent, image-grounded reasoning steps through explicit CoT documentation, whereas VISUALPUZZLES evaluates whether models can correctly reason about a wide range of visual patterns.

A tabular summary of the differences between VISUALPUZZLES and related benchmarks is shown in Table 12.

## J. Additional Analysis

### J.1. Proprietary V.S. Open Models

From Table 2, proprietary models (e.g., o4-mini and Claude-3.7-Sonnet) consistently achieve higher overall accuracy than most open-source models on VISUALPUZZLES. However, some open models also show competitive or even higher performance in both the overall accuracy and specific reasoning categories. For instance, Qwen2.5-VL-72B-Instruct demonstrates higher performance than GPT-4o on algorithmic reasoning, deductive reasoning, spatial reasoning, and overall accuracy. This indicates that while proprietary models currently have leading performance, open models are also rapidly improving on multimodal reasoning capabilities.

### J.2. Reasoning Category and Difficulty Levels

Figure 10 and Figure 9 present complementary views of human accuracy against three representative models: o1 (one of the best-performing proprietary models), Qwen2.5-VL-72B-Instruct (the strongest Qwen-based open model), and Llama-3.2-90B-Vision-Instruct (the strongest Llama-based open model). Specifically, Figure 9 compares performance across difficulty

*Table 12.* Comparison of puzzle-type reasoning benchmarks.

| Dataset | Reasoning load | Knowledge requirement | Dataset focus |
|---|---|---|---|
| KiloGram (Ji et al., 2022) | Heavy | Light | Tangram reasoning |
| M3CoT (Chen et al., 2024) | Heavy | Heavy | Domain-specific reasoning |
| LEGO-Puzzles (Tang et al., 2025) | Heavy | Light | Spatial reasoning |
| PuzzleVQA (?) | Moderate | Light | Abstract pattern recognition |
| CoMT (Cheng et al., 2025) | Heavy | Light | Image-grounded CoT and visual operation tracking |
| **VISUALPUZZLES (ours)** | Heavy | Light | Complex reasoning disentangled from domain knowledge |

levels for each reasoning category, while Figure 10 compares performance across categories within each difficulty level.

Humans consistently outperform all models across categories and difficulty levels, often by large margins. Notably, human performance remains high and relatively stable in the algorithmic, deductive, and spatial categories, even on hard questions. While accuracy does decline in analogical and inductive reasoning as difficulty increases, humans still maintain a clear advantage over models.

In contrast, model performance declines sharply as difficulty increases, especially for open-source models. Accuracy of Llama-3.2-90B-Vision-Instruct on hard analogical tasks drops to just 10%. Even one of the strongest proprietary models, o1, while more robust, still lags significantly behind humans, particularly on analogical, inductive, and spatial tasks. On easy tasks, some models perform competitively in certain categories, but this advantage largely disappears on medium and hard questions.

Interestingly, these models maintain a generally stable performance on algorithmic and deductive reasoning. For o1 and Qwen2.5-VL-72B-Instruct, their performances on algorithmic reasoning even go up for more difficult tasks, whereas human performance degraded as the difficulty level increases. However, all models, including o1, perform the worse at analogical, inductive and spatial reasoning in general, especially as the difficulty level increases. This suggests that models are relatively better at tasks requiring structured, rule-based algorithmic processing, while their performance degrades more steeply in tasks requiring relational abstraction (analogical), pattern induction (inductive), and visual understanding (spatial), particularly as the difficulty level increases. In summary, these results indicate that while some models exhibit promising performance on structured and easier reasoning tasks, multimodal models still struggle with abstract and complex reasoning, particularly when difficulty increases. Bridging the gap between model and human reasoning remains a critical challenge.

### J.3. Option Types and Difficulty Levels

Figure 11 compares human accuracy against three representative models, o1 (one of the best-performing proprietary models), Qwen2.5-VL-72B-Instruct (the strongest Qwen-based open model), and Llama-3.2-90B-Vision-Instruct (the strongest Llama-based open model), across different difficulty levels, separately for textual and visual answer options.

Across all participants and models, we observe a consistent pattern: text-based options result in higher accuracy than image-based options, with the performance gap widening as task difficulty increases. This trend holds even for human participants, whose accuracy drops from 92% to 40% on visual options when moving from easy to hard tasks, compared to a much smaller drop on text-based ones (93% to 73%).

For models, the gap is even more pronounced. For instance, Qwen2.5-VL-72B-Instruct achieves 58% accuracy on hard questions with text options, but only 20% when image options are used. o1 and Llama-3.2-90B-Vision-Instruct exhibit similar drops, suggesting a broad weakness in multi-image reasoning and visual option discrimination. These findings suggest that image-based answer options introduce significant additional complexity, requiring models not just to understand the question but to reason over multiple visual cues. This capability is essential for real-world tasks such as product selection, recommendation, and visual planning, where their decision-making process often depends on comparing visual content.

However, most pretraining datasets and benchmarks have traditionally emphasized textual QA formats, with far fewer examples involving visual options or structured visual comparisons. As a result, models may lack the inductive bias or learned attention mechanisms to handle visual alternatives effectively. These results highlight an important direction for

future work: expanding and diversifying training corpora to include multi-choice visual reasoning tasks, and developing architectures that are explicitly designed to process and compare visual candidates, especially under challenging conditions.

### J.4. Impact of CoT

Table 13 compares model performance under two prompting strategies: direct multiple-choice prompt vs. Chain-of-Thought (CoT) prompt. We observe that proprietary models and larger open models ($\geq$72B) benefit from CoT, while others show little to no improvement or even a decline in performance with CoT. For instance, both GPT-4o and Qwen2.5-VL-72B-Instruct show more than 20% increases in performance when using CoT. In contrast, several smaller models, such as Qwen2-VL-2B-Instruct and Cambrian-13B, exhibit decreased accuracy with CoT prompting. These results suggest that CoT can indeed enhance the reasoning capability of larger models whereas it may introduce unnecessary complexity or confusion for smaller models and thus decreasing performance.

| Model | Direct | CoT |
|---|---|---|
| GPT-4o | 34.0 | 41.6 |
| Gemini-1.5-Pro | 41.0 | 45.1 |
| Claude-3.5-Sonnet | 40.0 | 42.5 |
| Qwen2-VL-2B-Instruct | 31.3 | 26.1 |
| Qwen2.5-VL-7B-Instruct | 33.7 | 32.0 |
| Cambrian-13B | 27.4 | 26.5 |
| LLaVA-NeXT-34B | 29.8 | 29.6 |
| Qwen2.5-VL-72B-Instruct | 38.6 | 42.3 |
| LLama-3.2-90B-Vision-Instruct | 33.3 | 33.9 |

*Table 13.* Comparison of models with Direct Multiple Choice and CoT prompting.

### J.5. Comparison of Reasoning Paths

We compared the step-by-step traces of Claude-3.7-Sonnet on 50 VISUALPUZZLES instances where both the thinking and non-thinking modes failed. In **96%** of these cases, the thinking mode followed essentially the same core reasoning path as the non-thinking mode, differing only in verbosity rather than substance. In rare cases, the thinking mode pursued a more advanced reasoning path, while in a similarly small fraction, the non-thinking mode was actually more direct.

These results suggest that on VISUALPUZZLES, the addition of explicit "thinking" often does not lead to genuine reasoning improvements. The distribution of observed differences is presented in Table 14.

*Table 14.* Differences in reasoning paths between thinking and non-thinking modes (Claude-3.7-Sonnet).

| Difference in Reasoning Paths | Percentage |
|---|---|
| Same Core Logic but Thinking More Verbose | 96.0% |
| Thinking Mode More Advanced | 2.0% |
| Non-Thinking Mode More Direct | 2.0% |

### J.6. Correlation among reasoning categories for humans on VISUALPUZZLES

Figure 12 presents a heatmap showing the correlation among the five reasoning categories of human performance on VISUALPUZZLES. The correlation numbers are uniformly low, indicating that performance in one category is only weakly correlated to performance in other categories. This pattern is consistent with prior cognitive science work suggesting that different types of reasoning rely on partially dissociable cognitive and neural mechanisms rather than a single ability (Babcock & Vallesi, 2015; Bright & Feeney, 2014; Goel & Dolan, 2004; Green et al., 2010).

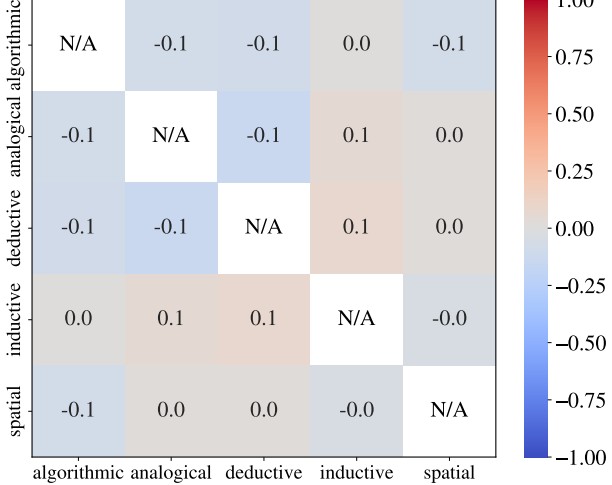

*Figure 12.* Correlation Heatmap among reasoning categories for humans on VISUALPUZZLES.

### J.7. Error Analysis of Thinking Mode

We further performed a manual error analysis on 50 instances where the thinking mode failed but the non-thinking mode succeeded. The majority of errors (**60%**) were caused by overthinking, in which the model performed redundant or repetitive reasoning steps. Another 32% of errors were due to getting lost in unnecessary details, which often obscured the correct reasoning path. Smaller

fractions of errors were due to refusal to answer (**6%**) and expressions of self-doubt (**2%**). These findings are summarized in Table 15.

Table 15. Error types in thinking mode where non-thinking mode succeeded (Claude-3.7-Sonnet).

| Error Type | Percentage |
| --- | --- |
| Overthinking | 60.0% |
| Excessive Detail | 32.0% |
| Refused to Answer | 6.0% |
| Self-Doubt | 2.0% |

### J.8. Distribution of Strategies in Correct and Incorrect Answers

We manually analyzed 40 algorithmic reasoning tasks (20 solved correctly and 20 solved incorrectly) to investigate the strategies used by current models. Annotation was performed with the assistance of a volunteer who had previously passed the Chinese Civil Service Exam.

Each response was categorized into one of four mutually exclusive strategies:

- **Surface-Pattern Copy:** Matching the output format or arithmetic pattern without following the underlying rule.

- **Early Halt:** Stopping the reasoning process prematurely once a plausible answer appears.

- **Genuine Reasoning:** Faithfully following the intended multi-step reasoning procedure.

- **Other:** Irrelevant speculation or hallucination.

Our analysis shows that models often reach correct answers through shallow heuristics rather than genuine deductive reasoning, with shortcut strategies more common in incorrect answers. The distributions for incorrect and correct answers are shown in Tables 16 and 17.

Table 16. Shortcut strategies for *incorrect* algorithmic reasoning answers.

| Strategy | Percentage |
| --- | --- |
| Surface-Pattern Copy | 40.0% |
| Early Halt | 30.0% |
| Genuine Reasoning | 25.0% |
| Other | 5.0% |

Table 17. Shortcut strategies for *correct* algorithmic reasoning answers.

| Strategy | Percentage |
| --- | --- |
| Surface-Pattern Copy | 30.0% |
| Early Halt | 15.0% |
| Genuine Reasoning | 45.0% |
| Other | 10.0% |

While genuine reasoning accounts for a larger share of correct answers, shortcut strategies such as surface-pattern matching and premature halting remain widespread. This suggests that models often rely on shallow heuristics rather than robust, human-like deductive reasoning.

### J.9. Visualizer

To facilitate deeper inspection and reproducibility, we have also developed a visualizer containing model responses for all evaluated models. The tool supports interactive comparison across models and exploration of reasoning traces, and will be released upon acceptance.

## J.10. Case Study of Reasoning

Figure 13 shows a case study demonstrating the similarity in structure and reasoning strategy between Claude-3.7-Sonnet and Claude-3.7-Sonnet-Thinking. Average textual similarity between model responses of these two models on VISUALPUZZLES is 0.9. Figure 14 shows an example of Claude-3.7-Sonnet-Thinking attempting to use branching to solve a puzzle but failed.

## J.11. Perception versus Reasoning

Perception, knowledge, and reasoning are three core pillars of multimodal reasoning for both humans and models. Because VISUALPUZZLES explicitly targets multimodal reasoning, it is impossible to completely isolate and control the perceptual component. Nevertheless, error analysis (see Figure 8) reveals that only **21%** of errors are attributable to perceptual mistakes, while **56%** of errors are reasoning-related. This indicates that the perceptual burden of VISUALPUZZLES is moderate, while reasoning emerges as the primary bottleneck.

## J.12. Analysis with o4-mini on Reasoning Strategies

We re-ran the reasoning strategy analysis using *o4-mini*, the best-performing model on VISUALPUZZLES. The results, shown in Table 18, are consistent with those obtained using Claude-3.7.

On average, o4-mini employs a higher proportion of reasoning strategies on VISUALPUZZLES than on MMMU. Furthermore, the correlation between accuracy and the occurrence of reasoning strategies is consistently lower on VISUALPUZZLES than on MMMU. This suggests that while VISUALPUZZLES elicits frequent use of branching and re-validation, their direct relationship to accuracy is weaker compared to MMMU.

*Table 18.* Usage of reasoning strategies and their correlation with accuracy for o4-mini.

| Benchmark | % Branching | % Re-validation | Correlation with Branching | Correlation with Re-validation |
|---|---|---|---|---|
| MMMU | 58.6% | 75.3% | 0.02 | 0.20 |
| VISUALPUZZLES | 91.3% | 64.0% | -0.11 | 0.04 |

We additionally analyzed the reasoning strategies of *o4-mini* in terms of *Answer-First* versus *Option-First* approaches. The results are shown in Table 19. Although the differences are less pronounced compared to those observed in Claude-3.7, the same trend emerges: o4-mini employs the Option-First approach more frequently on MMMU, while relying more on the Answer-First approach on VISUALPUZZLES.

*Table 19.* Distribution of Answer-First vs. Option-First approaches for o4-mini.

| Benchmark | % Answer-First | % Option-First |
|---|---|---|
| MMMU | 46.0% | 54.0% |
| VISUALPUZZLES | 56.7% | 43.3% |

## J.13. Enforcing Answer-First vs. Option-First Strategies

The analyses above measure the strategies that models *spontaneously* adopt. A natural follow-up question is whether these spontaneous preferences correspond to the strategy that is actually more effective on each benchmark, or whether they merely reflect differing model reactions to surface patterns in the questions. To probe this, we used explicit prompts to *force* Kimi-K2.5 to follow either the Answer-First or the Option-First strategy, and measured the resulting accuracy on both benchmarks (Table 20).

When the two strategies are explicitly enforced, the relative performance shifts in the *same* direction as the spontaneously observed preferences: on VISUALPUZZLES, Answer-First slightly outperforms Option-First (51.1% vs. 50.3%), whereas on MMMU, Option-First outperforms Answer-First (38.1% vs. 36.1%). The magnitude of these gaps is modest (0.8–2.0 points), consistent with the fact that strategy choice is only one of several factors determining whether a model answers correctly. Nonetheless, the consistency is suggestive that models tend to favor the locally more effective strategy for each

*Table 20.* Accuracy of Kimi-K2.5 when explicitly prompted to follow an Answer-First vs. Option-First strategy.

| Benchmark | Answer-First | Option-First |
|---|---|---|
| VISUALPUZZLES | 51.1% | 50.3% |
| MMMU | 36.1% | 38.1% |

benchmark, i.e. solving the puzzle independently before consulting the options on VISUALPUZZLES, versus using the options to guide knowledge retrieval on MMMU.

## K. Future Work

Our findings suggest that purely scaling inference compute, model size, and knowledge resources may not suffice for robust multimodal reasoning skills. Future research can explore new training strategies, specialized architectures, or model interpretations tailored to reduce reliance on memorized facts and enhance logical reasoning.

Extending VISUALPUZZLES to include additional types of multi-image reasoning or temporally dynamic visual information may further stress-test models' core reasoning abilities. By disentangling domain knowledge from multimodal reasoning, we hope VISUALPUZZLES will serve as a valuable tool for developing and evaluating next-generation MLLMs that excel at genuinely understanding and reasoning about the world without depending heavily on specialized factual knowledge.

## L. Limitations

**Disentangling Knowledge** Despite our best efforts to isolate domain-specific knowledge from the evaluation of multimodal reasoning, VISUALPUZZLES is still not entirely free of knowledge dependencies. Basic familiarity with everyday objects or common scenarios is still required; complete knowledge free evaluation remains an ideal rather than a practical reality.

**Real World Application** VISUALPUZZLES emphasizes puzzle-like questions that may not reflect the full diversity of real-world scenarios, limiting generalizability to more specialized domains.

**Question Format** VISUALPUZZLES focuses on multiple-choice questions, which may not capture the breadth of open-ended reasoning tasks where models must generate complex textual or visual outputs.

Future work can address these limitations by including more varied question formats, broader domains, and more granular analyses of a model's knowledge versus its multimodal reasoning abilities.

# M. Case Study

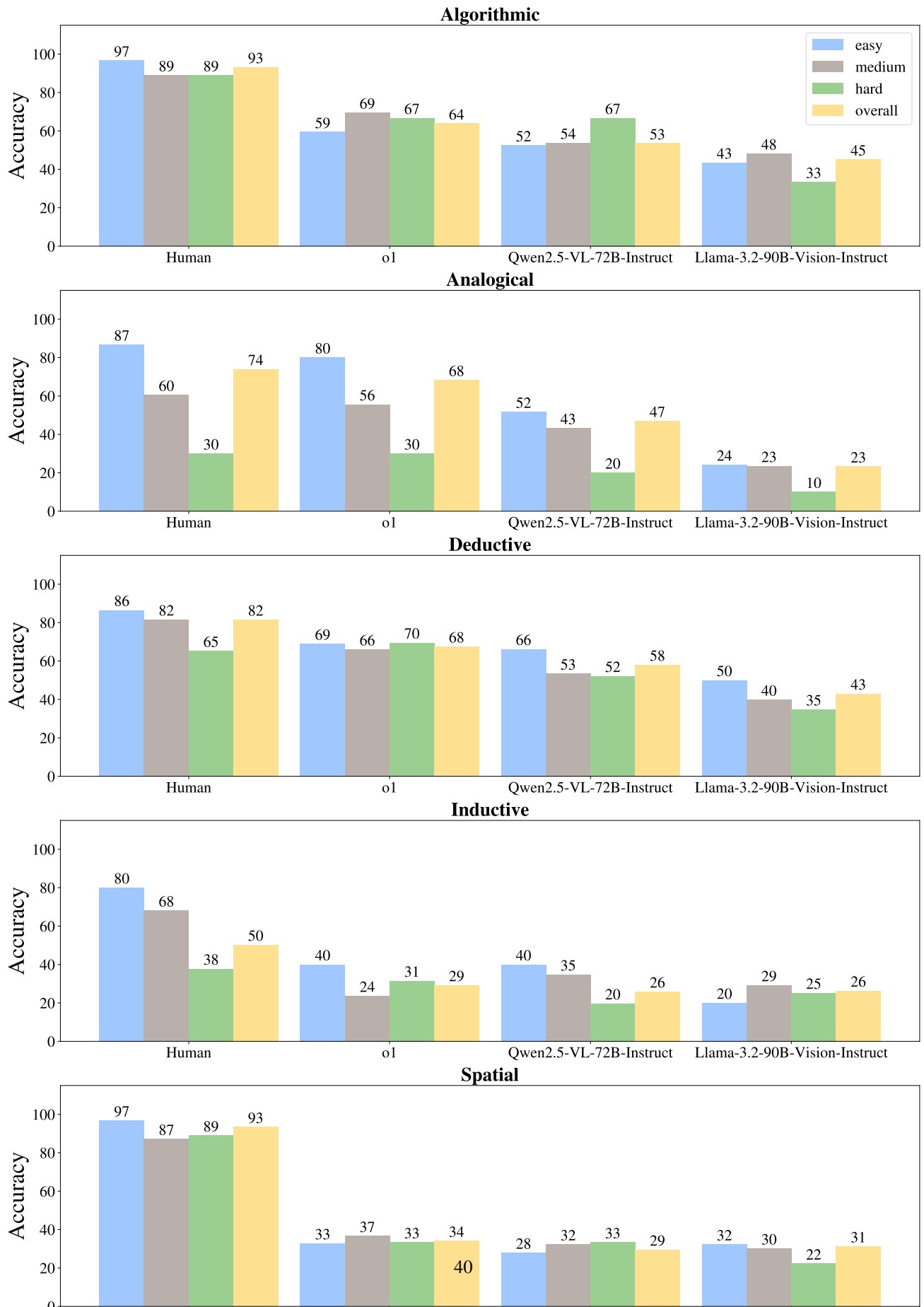

*Figure 9.* Comparison of accuracy across different reasoning categories for human participants, one of the best performing proprietary models o1, the best performing Qwen-based open model Qwen2.5-VL-72B-Instruct, and the best performing Llama-based open model

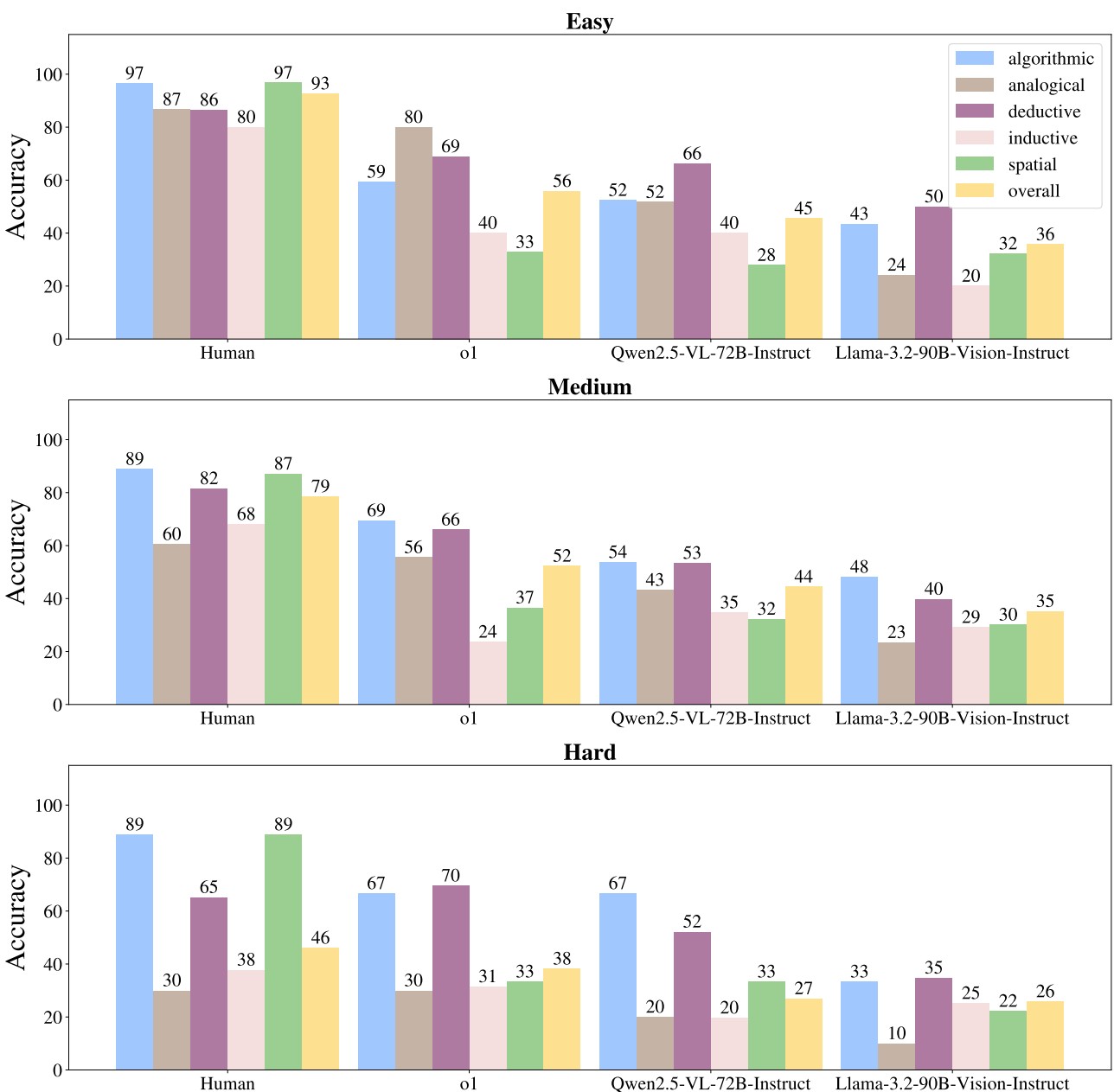

*Figure 10.* Comparison of accuracy across different difficulty levels for human participants, one of the best performing proprietary models o1, the best performing Qwen-based open model Qwen2.5-VL-72B-Instruct, and the best performing Llama-based open model Llama-3.2-90B-Vision-Instruct, measured across reasoning categories.

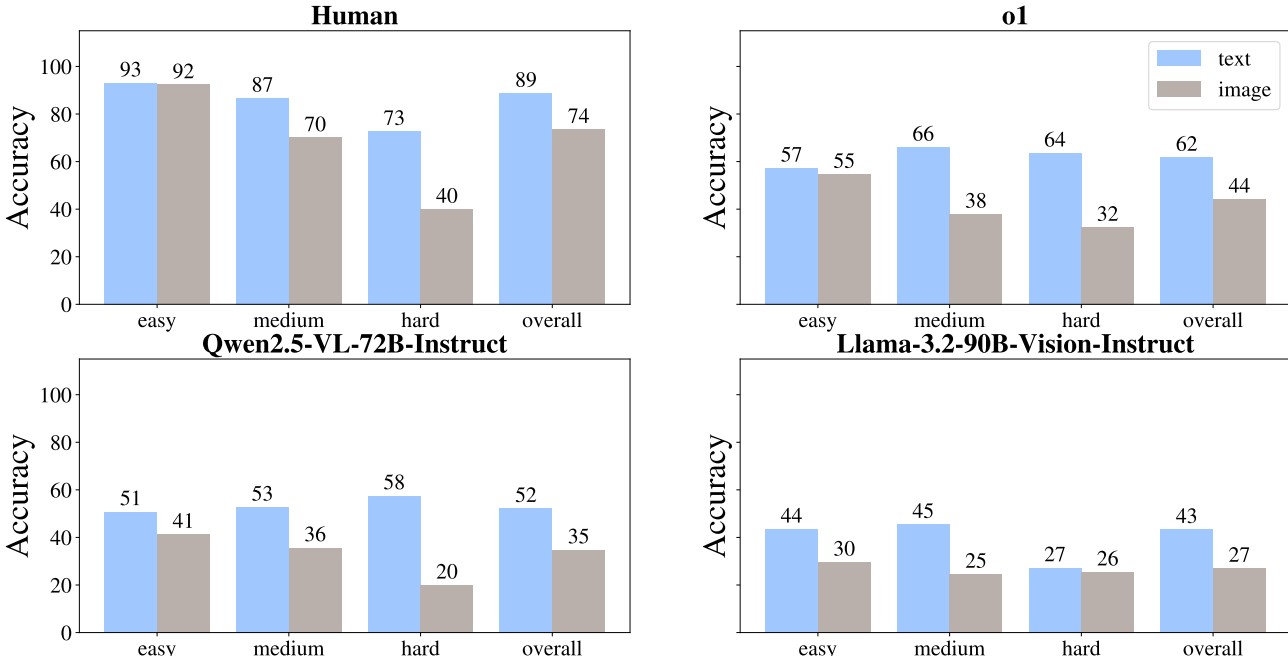

*Figure 11.* Comparison of accuracy across different difficulty levels for human participants, one of the best performing proprietary model o1, the best performing Qwen-based open model Qwen2.5-VL-72B-Instruct, and the best performing Llama-based open model Llama-3.2-90B-Vision-Instruct, measured on textual v.s. visual option types.

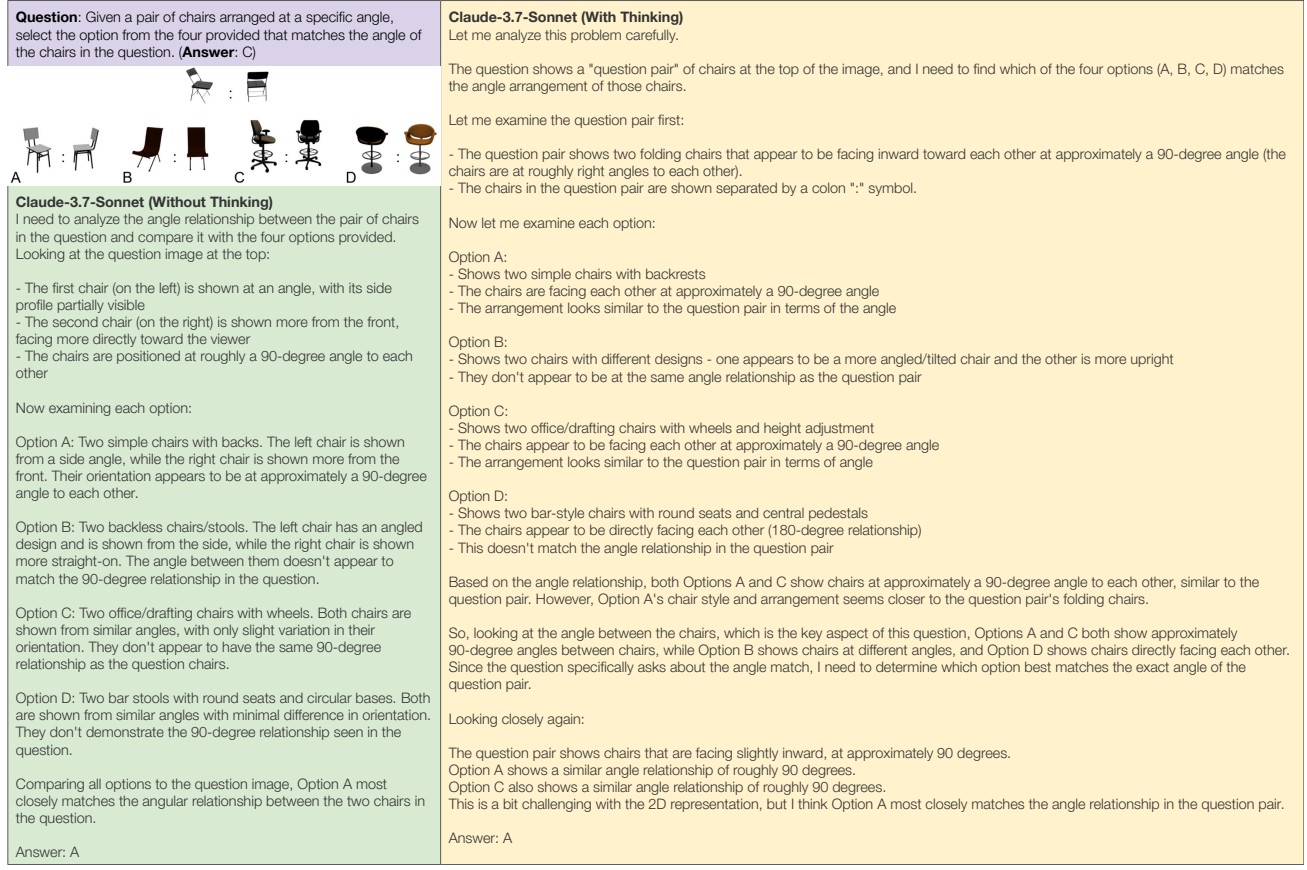

*Figure 13.* Case Study showing the similarity in structure and reasoning strategy between Claude-3.7-Sonnet-Thinking and Claude-3.7-Sonnet. Similarity between model responses of these two models on VISUALPUZZLES is 0.9.

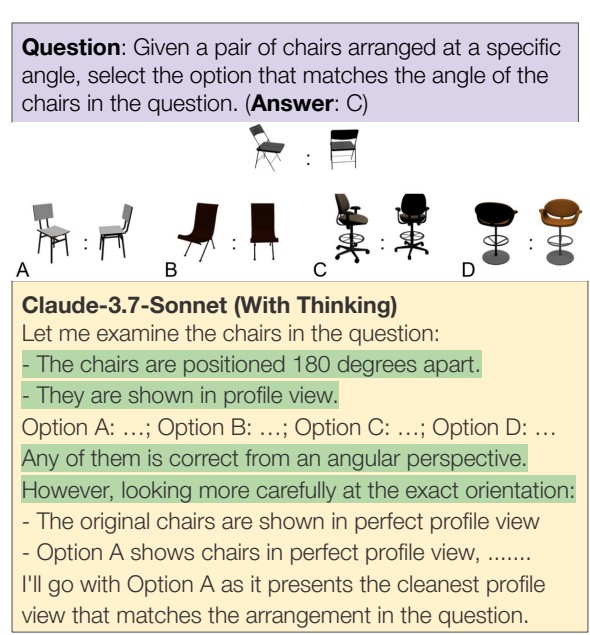

*Figure 14.* An example of Claude-3.7-Sonnet-Thinking using branching to solve a puzzle.

## Algorithmic (Easy) 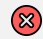

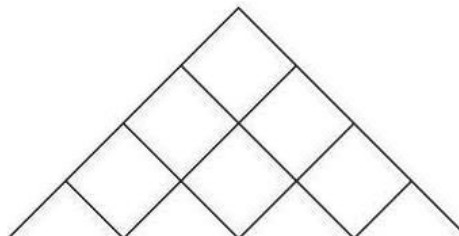

**Question:** There are seven squares in the picture. How many more triangles than squares are there in the picture?
**Options:** (A) 1 (B) 2 (C) 3 (D) 4

**Ground Truth: (C)**

**Claude-3.7-Sonnet-Thinking: <*thinking*>** I need to count the number of squares and triangles in the given picture, and then find how many more triangles there are than squares.

First, let me count the squares:
1. I can see 7 small squares arranged in a triangular pattern within the large triangle.

Now, let me count the triangles:
1. There's the large outer triangle that contains everything.
2. There are small triangles formed by the division of the large triangle by the squares.

Let me count these small triangles carefully:
- Along the bottom row, I see 4 triangles (formed by the squares and the bottom edge)
- On the left side of the large triangle, I see 3 triangles (formed by the squares and the left edge)
- On the right side of the large triangle, I see 3 triangles (formed by the squares and the right edge)
- Don't forget the large outer triangle itself: 1 triangle

So in total: 4 + 3 + 3 + 1 = 11 triangles

Now I can find the difference:
Number of triangles - Number of squares = 11 - 7 = 4

So there are 4 more triangles than squares in the picture.

Looking at the options, the answer is (D) 4.
**<*end_thinking*>**
Looking at the image, I need to count the total number of squares and triangles, then find their difference.

First, I'll count the squares:
- I can see 7 small squares arranged in a triangular pattern inside the large triangle.

Next, I'll count the triangles:
1. There is the large outer triangle that contains everything.
2. Along the bottom edge of the large triangle, there are 4 small triangles.
3. Along the left edge of the large triangle, there are 3 small triangles.
4. Along the right edge of the large triangle, there are 3 small triangles.

Total number of triangles = 1 + 4 + 3 + 3 = 11 triangles

Now I can calculate how many more triangles than squares there are:
11 triangles - 7 squares = 4 more triangles

Answer: D

**Error Category:** Reasoning Error
**Error Reason:** The model gets the answer wrong because it miscounts the number of triangles, specifically double-counting the triangles formed by small triangles and squares. The response incorrectly conclude there are 11 triangles, but a careful manual inspection shows only 10 distinct triangles, making the correct difference 3, not 4.

## Algorithmic (Easy) 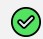

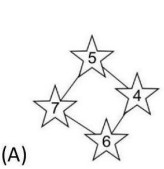
(A)

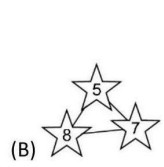
(B)

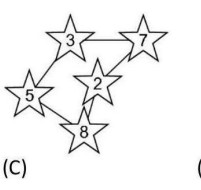
(C)

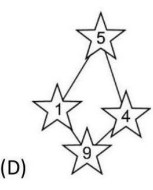
(D)

**Question:** In the Kangaroo constellation, all stars have a number greater than 3 and their sum is 20. Which option shows the Kangaroo constellation?

**Ground Truth: (B)**

**Claude-3.7-Sonnet-Thinking:** *<thinking>* The question asks about the Kangaroo constellation, which has the following properties:
1. All stars have a number greater than 3
2. The sum of all numbers is 20

Let me check each option to see which one satisfies these conditions:

Option (A):
Stars have values: 5, 4, 7, 6
- Are all numbers greater than 3? Yes (5, 4, 7, 6 are all > 3)
- Sum: 5 + 4 + 7 + 6 = 22
This doesn't match the required sum of 20.

Option (B):
Stars have values: 5, 8, 7
- Are all numbers greater than 3? Yes (5, 8, 7 are all > 3)
- Sum: 5 + 8 + 7 = 20
This matches both conditions: all numbers are greater than 3, and they sum to 20.

Option (C):
Stars have values: 5, 3, 7, 2, 8
- Are all numbers greater than 3? No (3 and 2 are not > 3)
- Sum: 5 + 3 + 7 + 2 + 8 = 25
This doesn't satisfy the first condition about all stars having numbers greater than 3.

Option (D):
Stars have values: 5, 1, 4, 9
- Are all numbers greater than 3? No (1 is not > 3)
- Sum: 5 + 1 + 4 + 9 = 19
This doesn't satisfy either condition.

So, only Option (B) satisfies both conditions of the Kangaroo constellation.
*<end_thinking>*
# Solving the Kangaroo Constellation Problem

I need to identify the constellation where all stars have numbers greater than 3 and sum to 20.

Let me examine each option:

## Option (A)
Stars: 5, 4, 7, 6
- All numbers > 3? Yes
- Sum: 5 + 4 + 7 + 6 = 22
Doesn't match the required sum of 20.

## Option (B)
Stars: 5, 8, 7
- All numbers > 3? Yes
- Sum: 5 + 8 + 7 = 20
Matches both requirements!

## Option (C)
Stars: 5, 3, 7, 2, 8
- All numbers > 3? No (3 and 2 are ≤ 3)
- Sum: 5 + 3 + 7 + 2 + 8 = 25
Doesn't satisfy the first condition.

45

## Option (D)
Stars: 5, 1, 4, 9
- All numbers > 3? No (1 is ≤ 3)
- Sum: 5 + 1 + 4 + 9 = 19

## Algorithmic (Medium) ⊗

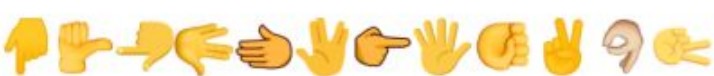

**Question:** How many left hands appear in the image?
**Options:** (A) 1 (B) 8 (C) 4 (D) 5

**Ground Truth: (C)**

**Claude-3.7-Sonnet-Thinking: <*thinking*>**
Looking at the image, I can see approximately 10-12 hand icons/emoji in a row. They're all yellow (with one that appears to be skin-colored), and they show various hand gestures.

To determine which ones are left hands, I need to look at the thumb position. In a left hand, the thumb would typically be on the right side when the palm is facing forward.

Let me go through each hand from left to right:

1. Thumbs down gesture - appears to be a left hand with thumb on right side
2. Fist or closed hand - orientation is harder to determine but seems to be a left hand
3. Open hand or waving hand - appears to be a left hand
4. Hand pointing or gesturing - appears to be a left hand
5. Open hand with fingers spread - appears to be a left hand
6. Fist with extended thumb (like "thumbs up" but horizontal) - appears to be a right hand (thumb on left)
7. Open hand with fingers spread - appears to be a left hand
8. Hand making "OK" gesture - appears to be a left hand
9. Victory/peace sign hand - appears to be a left hand
10. Ear or side of face (not a hand)
11. Hand pointing or gesturing - appears to be a right hand

So counting the left hands, I see approximately 8 left hands. This would correspond to option (B) 8.

However, I want to be careful because determining left vs. right can sometimes be ambiguous in emoji or simplistic hand representations. Let me double check my count.

Looking again, I see the thumbs down, several open hands, some gesture hands, and a few that are definitely left hands based on thumb position. The total count of left hands appears to be around 8, though there's some ambiguity with a couple of the representations.

Based on my analysis, the answer appears to be (B) 8.
**<*end_thinking*>**
To solve this question, I need to examine the image carefully and identify which hands are left hands.

Looking at the row of yellow hand emojis, I need to determine whether each is a left or right hand. For hand emojis, I can identify a left hand by noticing the thumb position - on a left hand, the thumb will typically appear on the right side when the palm is facing forward.

Going through each emoji from left to right:
1. Thumbs down - this is a left hand (thumb is on the right)
2. Fist - appears to be a left hand
3. Waving hand - this is a left hand
4. Pointing hand - this is a left hand
5. Open hand with fingers spread - this is a left hand
6. Fist with extended thumb - this appears to be a right hand (thumb on left)
7. Open hand - this is a left hand
8. OK gesture hand - this is a left hand
9. Peace sign hand - this is a left hand
10. Ear emoji (not a hand)
11. Pointing gesture - right hand

I count 8 left hands total in the image.

46

Answer: B

**Error Category:** Spatial / Orientation Error

## Algorithmic (Medium) 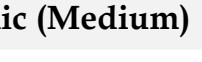

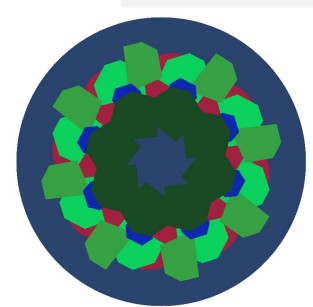

**Question:** The entire pie seen in the image is divided among several children. Each child receives a piece of pie, and each piece of pie looks identical. What is the maximum possible number of children there?
**Options:** (A) 9 (B) 8 (C) 10 (D) 4

**Ground Truth: (B)**

**Claude-3.7-Sonnet-Thinking: <*thinking*>** I need to determine how many equal-sized pieces this pie can be divided into. Since each child will receive an identical piece, I need to count the maximum number of equal portions visible in the image.

Looking at the image, I can see that this appears to be an unusual-looking pie with different colored sections arranged in a circular pattern. It's not a typical pie with simple wedge-shaped slices.

From what I can observe, there appears to be a pattern of colored shapes (green, light green, blue, and red) arranged in a circle. To determine how many equal pieces there are, I need to identify the repeating pattern and count how many times it repeats around the circle.

The pie seems to have a symmetrical arrangement. Looking carefully, I can see that there appear to be 8 sections around the outer ring that follow a similar pattern. I can see repeated patterns of green and light green shapes with blue and red accents between them.

Looking at the options:
(A) 9
(B) 8
(C) 10
(D) 4

Based on my observation of the image, there appear to be 8 similar sections arranged around the circle. This would suggest that option (B) 8 is the correct answer.

However, I should double-check by looking for any other patterns or symmetries in the image. It's possible that each section could be further divided, or that there's a different way to interpret the equal pieces.

After careful examination, I'm confident that there are 8 identical sections arranged in a circular pattern. Therefore, the maximum number of children who could receive identical pieces of pie would be 8.
**<*end_thinking*>**
# Determining the Maximum Number of Children

To solve this problem, I need to examine the pie in the image and determine how many identical pieces it can be divided into.

## Analysis of the Pie
Looking at the image, I can see a circular pie with a pattern of colored sections. The key is to identify how many identical sections are present or could be created.

Upon careful examination, I notice that the pie has a symmetrical arrangement with a repeating pattern. The colorful segments (green, light green, red, and blue shapes) appear to form a consistent pattern around the circle.

Counting these repeating sections carefully, I can identify 8 identical portions around the circumference of the pie. Each of these sections follows the same pattern of colored shapes.

## Conclusion
Since each child must receive an identical piece, and there are 8 identical sections visible in the pie, the

## Algorithmic (Hard) 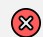

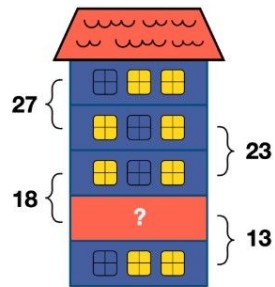

**Question:** There are 53 residents living in the apartment. How many people live on the second floor?
**Options:** (A) 5 (B) 8 (C) 10 (D) 13

**Ground Truth: (A)**

**Claude-3.7-Sonnet-Thinking:** *<thinking>* This image shows an apartment building with different sections labeled with numbers. Let me analyze what the numbers represent:

- The building appears to have multiple floors.
- The number 27 is shown on the left side of one section (blue).
- The number 23 is shown on the right side of a section (also seems to cover two blue floors).
- The number 18 is shown on the left side of a pink/red section marked with "?".
- The number 13 is shown on the right side of the bottom floor.

The question asks how many people live on the second floor, but I need to determine which floor is the second floor and how many people live there.

Looking at the building from bottom to top:
1. Bottom floor (blue with windows): labeled with 13 on the right
2. Second floor (pink/red with "?"): labeled with 18 on the left
3. Two middle floors (blue with windows): part of the 23 on the right
4. Top floor (blue with windows): labeled with 27 on the left

So it seems the second floor from the bottom is the one marked with "?" and has 18 on the left side. But the question asks specifically how many people live on the second floor, not how many are associated with the label.

Let me think about this more carefully. The total number of residents is 53. The numbers shown might be:
- 27: Number of people on the top floor
- 23: Number of people on the two middle floors combined
- 18: Number of people on the second floor
- 13: Number of people on the bottom floor

Let's check if these add up: 27 + 23 + 18 + 13 = 81, which is more than 53.

Wait, I think I'm misinterpreting. The numbers might represent:
- 27: People on left side (top section)
- 23: People on right side (upper two floors)
- 18: People on left side (lower section)
- 13: People on right side (bottom floor)

So 27 + 18 = 45 people on the left, and 23 + 13 = 36 people on the right. That's still 45 + 36 = 81 people, which doesn't match 53.

Let me try a different approach. Maybe the 23 represents the number of people on the right side of the building (covering multiple floors), and 18 represents people on another part.

Actually, looking at the diagram again, it appears that:
- 27: Left side of the top section
- 23: Right side of the top two sections
- 18: Left side of the bottom section
- 13: Right side of the bottom section

Given the total is 53 residents, and we're trying to find the number on the second floor, let's try to interpret the diagram differently.

The first (bottom) floor seems to have 13 people.
The top appears to have 27 people.
That leaves 53 - 13 - 27 = 13 people for the second and third floors combined.

I'm still uncertain, but based on my calculations and the available options, it seems like (D) 13 might be the answer if we interpret the second floor as having all the remaining residents after accounting for the top and bottom floors.
*<end_thinking>*
# Solving the Apartment Residents Problem

Looking at the image, I need to determine how many residents live on the second floor of this apartment building.

I can see a diagram of a building with several numbers:
- The top section has "27" on the left side
- The middle two floors have "23" on the right side
- There's a section marked with "18" on the left
- The bottom floor has "13" on the right side
- The second floor from the bottom is marked with a "?" - this is what I need to find

I notice the question states there are 53 residents total living in the apartment building.

Let me work this out:
- The top floor appears to have 27 residents
- The bottom floor appears to have 13 residents
- That means the middle two floors together must have: 53 - 27 - 13 = 13 residents

48

Looking closely at the image, the second floor is specifically marked with a "?" and there are numbers on the sides indicating 13 for that floor.

Therefore, the second floor has 13 residents.

Answer: D

# Algorithmic (Hard) 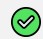

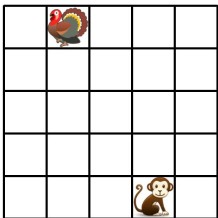

**Question:** A bird can only move horizontally or vertically towards a monkey in the grid. The number of ways that the bird can reach the monkey is:
**Options:** (A) 15 (B) 14 (C) 18 (D) 12

**Ground Truth: (A)**

**Claude-3.7-Sonnet-Thinking:** *<thinking>* Looking at the grid, I need to identify the positions of the bird and the monkey.

The bird appears to be at the top of the grid in the second column, specifically at position (0, 1) if we use a zero-indexed grid counting from the top-left.

The monkey appears to be at the bottom-right of the grid, at position (4, 3) using the same coordinate system.

To count the number of ways the bird can reach the monkey, I need to consider that the bird can only move horizontally or vertically. This means the bird can only move right or down to reach the monkey from its starting position.

The total number of valid paths will be the number of ways to arrange the required horizontal and vertical moves.

From position (0, 1) to (4, 3), the bird needs to move:
- 4 steps down (from row 0 to row 4)
- 2 steps right (from column 1 to column 3)

So in total, the bird needs to make 4 + 2 = 6 moves, with 4 being vertical and 2 being horizontal.

The number of ways to arrange these moves is given by the combination formula:
C(total moves, horizontal moves) = C(6, 2) = 6! / (2! * 4!) = (6 * 5) / (2 * 1) = 30 / 2 = 15

Alternatively, you can calculate C(6, 4) which would give the same result:
C(6, 4) = 6! / (4! * 2!) = 15

So there are 15 different ways for the bird to reach the monkey.

Looking at the options, this corresponds to answer (A) 15.
*<end_thinking>*
# Solving the Grid Path Problem

In this problem, I need to find the number of ways that a bird can reach a monkey by moving only horizontally or vertically in a grid.

Looking at the image:
- The bird is located at the top of the grid in the second column (position (0,1) if we start counting from 0)
- The monkey is located at the bottom-right area (position (4,3))

## Step 1: Determine the required moves
From the bird to the monkey, the bird needs to move:
- 4 steps down (from row 0 to row 4)
- 2 steps right (from column 1 to column 3)

So the bird must make a total of 6 moves: 4 vertical and 2 horizontal.

## Step 2: Calculate the number of possible paths
This is a classic combinatorial problem. The number of different ways to arrange these moves is:

C(total moves, horizontal moves) = C(6,2) = 6!/(2!×4!)
= (6×5)/(2×1)
= 30/2
= 15

49

Alternatively, we could calculate C(6,4) which gives the same result.

## Analogical (Easy) 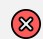

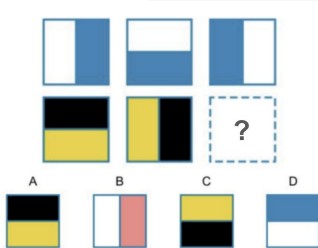

**Question:** Given the pattern in the first line of blocks, which option at the bottom of the image fits in the question mark in the second line of blocks?

**Ground Truth: (C)**

**Claude-3.7-Sonnet-Thinking:** *<thinking>* Let me analyze the pattern in the first line of blocks and see how it relates to the second line of blocks.

First line of blocks:
1. The first block has white on the left and blue on the right
2. The second block has white on the top and blue on the bottom
3. The third block has blue on the left and white on the right

Second line of blocks:
1. The first block has black on the top and yellow on the bottom
2. The second block has yellow on the left and black on the right
3. The third block has a question mark (this is what we need to figure out)

Let me think about the possible relationships between the first and second line:

Possibility 1: Color transformation
- Blue in the first line becomes black in the second line
- White in the first line becomes yellow in the second line

Possibility 2: Position pattern
The first line blocks follow a pattern of right/bottom/left coloring for the blue part.
The second line blocks might follow the same positional pattern but with different colors.
Based on the color transformation pattern I identified (blue→black, white→yellow), and looking at the third block in the first line (blue left, white right), the answer should be a block with black on the left and yellow on the right.

Looking at the options:
- Option A: Black on top, yellow on bottom (like the first block in the second row)
- Option B: White on left, pink/red on right (doesn't match our color scheme)
- Option C: Yellow on top, black on bottom (this would be inverse of first block in second row)
- Option D: Blue on bottom, white on top (similar to the second block in the first row)

None of the options exactly match what I'm looking for. The relationship between the first line and second line might be more complex. Let me look at it again:

First block: white left, blue right → black top, yellow bottom
Second block: white top, blue bottom → yellow left, black right

Let me try to formulate a rule:
- White areas become yellow
- Blue areas become black
- Position changes: left→top, right→bottom, top→left, bottom→right

So if the third block has blue left and white right, then the third block in the second row should have black top and yellow bottom according to this rule.

Option A matches this pattern.

Let me double-check:
First rule pattern: white→yellow, blue→black
Position transformation: left→top, right→bottom, top→left, bottom→right

For block 3 in row 1: blue on left, white on right
Applying the rules: blue left becomes black top, white right becomes yellow bottom
This gives us: black top, yellow bottom
Option A has black on top and yellow on the bottom, so it seems to be the correct answer.
*<end_thinking>*
To solve this problem, I need to identify the pattern in the first line of blocks and how it transforms in the second line.

Let me analyze both rows carefully:

First row:
- Left block: White on left, blue on right
- Middle block: White on top, blue on bottom
- Right block: Blue on left, white on right

Second row:
- Left block: Black on top, yellow on bottom
- Middle block: Yellow on left, black on right
- Right block: ? (what we need to determine)

Looking at the transformation from first to second row:
1. The blue areas in the first row seem to become black in the second row
2. The white areas in the first row seem to become yellow in the second row
3. There's also a positional transformation:
   - Left side → Top position
   - Right side → Bottom position
   - Top position → Left side
   - Bottom position → Right side
Option A shows black on top and yellow on bottom, which matches our expected pattern.

Answer: B

## Analogical (Easy) ✓

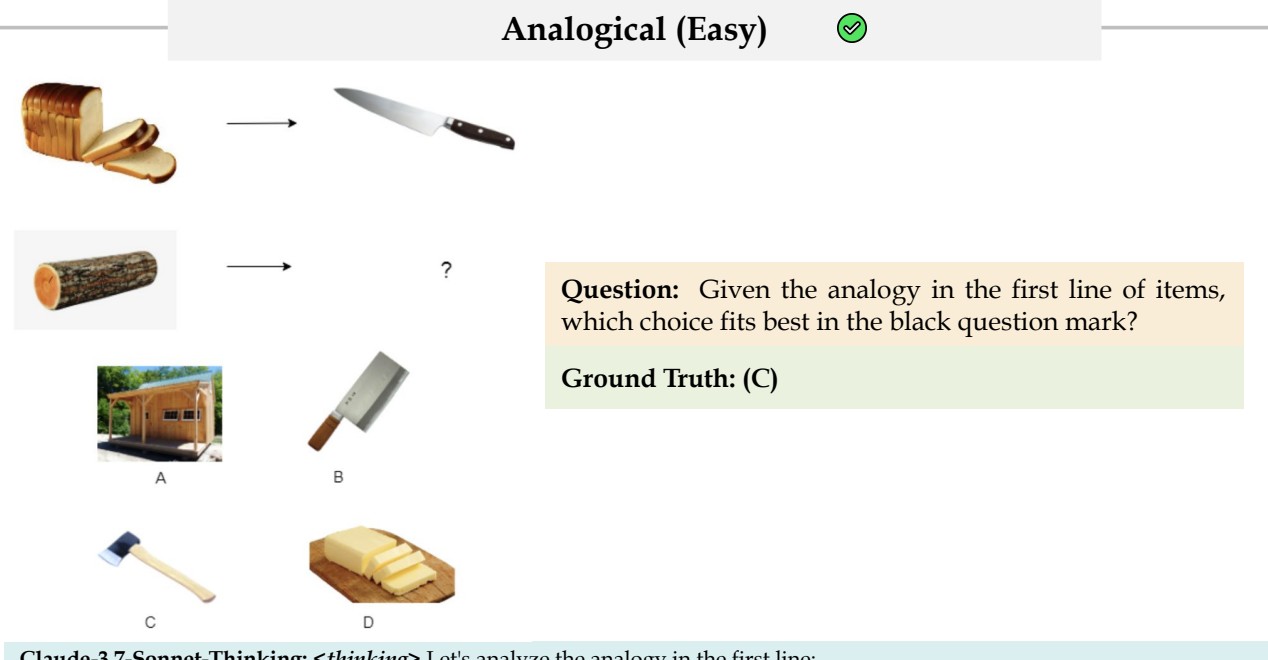

**Question:** Given the analogy in the first line of items, which choice fits best in the black question mark?

**Ground Truth: (C)**

**Claude-3.7-Sonnet-Thinking: <*thinking*>** Let's analyze the analogy in the first line:

The first line shows sliced bread on the left and a knife on the right, connected by an arrow.

The relationship here appears to be: the knife is a tool used to cut/slice the bread. The knife is what transforms a whole loaf of bread into slices.

Now, let's look at the second line:
- On the left, we see a log (a section of tree trunk)
- On the right, we need to determine which tool would be used to process or cut this log

Let's examine each option:
(A) A wooden shed or small building
(B) A cleaver or butcher's knife
(C) An axe
(D) Sliced butter on a wooden board

The question is asking which tool would be used to process a log, similar to how a knife is used to slice bread.

- Option A (wooden shed) is not a tool but a structure that might be built using processed wood
- Option B (cleaver/butcher's knife) is a cutting tool but not typically used for logs
- Option C (axe) is a tool specifically designed for cutting logs/wood
- Option D (sliced butter) is not a tool at all, but a food product

Following the analogy "bread → knife" (something that needs to be cut → tool used to cut it), the logical completion for "log → ?" would be an axe, which is a tool specifically designed to cut wood.

Therefore, the answer appears to be (C) the axe.
**<end_thinking>**
I need to determine the relationship between the items in the first line and then apply the same relationship to find the missing item in the second line.

Looking at the first line:
- On the left, there's a loaf of bread with some slices cut from it
- On the right, there's a knife
- The relationship is that the knife is the tool used to cut/slice the bread

Now, for the second line:
- On the left, there's a log (a section of tree trunk)
- On the right, I need to find the appropriate tool that would be used to cut or process this log

Let me analyze each option:
(A) A wooden shed/small building - This is a structure that might be built using processed wood, not a tool for cutting logs
(B) A cleaver/butcher's knife - While this is a cutting tool, it's not typically used for logs
(C) An axe - This is a tool specifically designed for chopping and cutting logs/wood
(D) Sliced butter on a board - This is a food product, not a tool for processing logs

The pattern is "material → tool used to cut that material"
- Bread → Knife

## Analogical (Medium) 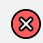

Origin    Analogy    Origin'    Analogy'

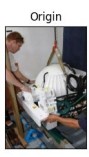 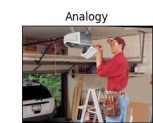 :: 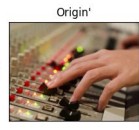 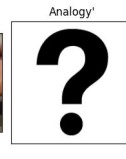

Candidates

A    B    C    D

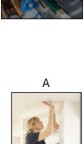 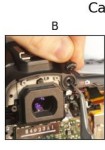 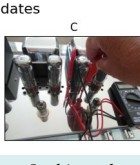 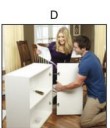

**Question:** Based on the relationship between Origin and Analogy, select the option from the candidates that best completes this analogy.

**Ground Truth: (C)**

**Claude-3.7-Sonnet-Thinking: <*thinking*>** In this analogy question, I need to find the relationship between "Origin" and "Analogy" on the left side of the equation, and then identify which candidate option (A, B, C, or D) best completes the right side of the equation when paired with "Origin'".

Let's analyze the images:

Left side:
- Origin: A person installing or working on what appears to be a large appliance or system
- Analogy: A person installing or repairing what looks like a garage door opener or ceiling-mounted device

Right side:
- Origin': A person's hand operating what appears to be a sound mixing board or control panel with sliders and buttons
- Analogy': This is what we need to determine from options A, B, C, or D

The relationship between Origin and Analogy seems to be:
Both involve installation, repair, or maintenance of mechanical/electrical systems, but in different contexts. They show people actively working on or operating different types of equipment or machinery.

Now, let's look at the candidate options and see which one best represents the same relationship with Origin' (the sound mixing board):

A: A person painting or working on a ceiling/wall
B: A close-up of someone working on or repairing what appears to be electronic components
C: Some mechanical or electronic equipment with wiring
D: People moving or installing what looks like a home appliance (possibly a refrigerator or washer/dryer)

I need to determine which option shows a relationship to the sound mixing board (Origin') that parallels the relationship between the first two images.

If Origin shows someone installing/repairing an appliance, and Analogy shows someone installing/repairing a different type of equipment (garage door opener), then the relationship is about installation/repair of different types of systems.

Similarly, if Origin' shows someone operating a sound mixing board, then Analogy' should show someone operating a different type of equipment or control system.

Looking at the options:
A: Shows installation/repair work (similar to Origin/Analogy) but not operation of controls
B: Shows repair/maintenance of electronic components
C: Shows equipment but not clearly someone operating it
D: Shows people moving/installing an appliance

The key relationship between Origin and Analogy seems to be: a person actively working on installing or maintaining a system. Similarly, Origin' shows a person actively operating a control system (mixing board).

Option B seems closest as it shows hands working on electronic components, which could be considered operating or maintaining another type of electronic system. Though it's not entirely clear from the image, this appears to be the closest match to the relationship pattern.
**<end_thinking>**
# Analogy Analysis

I need to understand the relationship between "Origin" and "Analogy" in the first pair, then apply that same relationship to find what completes the "Origin'" and "Analogy'" pair.

## Examining the first pair:
- **Origin**: Shows a person installing or repairing what appears to be a large appliance (possibly a washer or dryer)
- **Analogy**: Shows a person installing or working on a ceiling-mounted device (appears to be a garage door opener)

The relationship seems to be: Both involve installation/repair/maintenance of household mechanical systems, but in different locations and of different types.

## Examining the second pair:
- **Origin'**: Shows hands operating a sound mixing board or audio control panel with sliders and buttons
- **Analogy'**: This is what we need to determine

I need to find an option that shows someone working with or operating another type of electronic/technical equipment, maintaining the same relationship as the first pair.

## Analyzing the options:
- **A**: Person painting or installing something on a ceiling
- **B**: Close-up of hands working on electronic components/circuit board
- **C**: Complex mechanical/electronic equipment with wiring
- **D**: Two people moving or installing what appears to be a refrigerator or appliance

The relationship we're looking for should show hands working on/operating a different type of technical equipment than a mixing board.

52

Option B shows hands working directly with electronic components/circuit board, which parallels how the hands in Origin' are working directly with the audio equipment. Both involve hands-on technical work with electronic systems, just different types.

Answer: B

**Error Category:** Reasoning Error

## Analogical (Medium) ✅

Origin    Analogy    Origin'    Analogy'

∷

Candidates

A    B    C    D

**Question:** Based on the relationship between Origin and Analogy, select the option from the candidates that best completes this analogy.

**Ground Truth: (B)**

**Claude-3.7-Sonnet-Thinking:** *<thinking>* Let me analyze the relationship between the "Origin" and "Analogy" images to determine the pattern, and then apply that to find the best match for "Origin'" from the candidates.

In the first pair:
- Origin: A dog (looks like a spaniel or similar breed) searching/sniffing in grass
- Analogy: What appears to be a small animal (possibly a rodent) in its natural habitat with vegetation

In the second pair:
- Origin': A dog (appears to be a small white dog) eating or drinking from a bowl/dish
- Analogy': We need to determine this

The relationship seems to be showing an animal (dog) in the Origin image, then showing what might be a prey animal or similar smaller animal in a related natural context in the Analogy image.

So, for Origin' which shows a dog eating/drinking, the Analogy' should show some smaller animal also eating.

Looking at the candidates:
A: Shows a dog with what appears to be a stick or something in its mouth in snow
B: Shows a mouse or small rodent eating something
C: Shows what looks like a groundhog or marmot near flowers/plants
D: Shows another dog's face/profile

Based on the pattern (dog → smaller animal in natural context, especially related to the activity the dog is doing), option B (the mouse/rodent eating) would best complete the analogy since it shows a smaller animal doing the same activity (eating) as the dog in Origin'.
*<end_thinking>*
I need to determine the relationship between the "Origin" and "Analogy" images, and then apply that same relationship to find the appropriate "Analogy'" that matches "Origin'".

Looking at the first pair:
- Origin: A dog searching or sniffing in grass (appears to be hunting or tracking)
- Analogy: A small animal (looks like a rodent) in its natural habitat among vegetation

Now looking at the second pair:
- Origin': A dog eating or drinking from a bowl/dish
- Analogy': This is what I need to determine

The relationship between the first pair seems to be: a dog engaged in a natural behavior (hunting/tracking), followed by what could be its potential prey or a similar smaller animal in its natural environment.

Following this pattern, the "Analogy'" should show a smaller animal engaged in eating or drinking (similar to what the dog is doing in "Origin'").

Examining the candidates:
A: A dog with something in its mouth on snow
B: A mouse/small rodent eating food
C: What appears to be a groundhog or marmot among vegetation/flowers
D: A profile view of another dog

## Analogical (Hard) 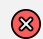

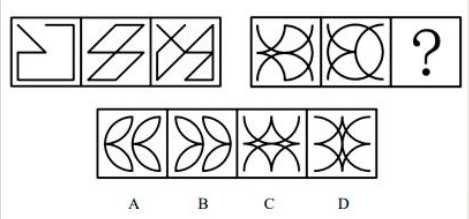

A    B    C    D

**Question:** This question contains two sets of figures and 4 figures to choose from. The two sets of figures have certain similarities as well as differences. You are required to choose one option that is most suitable to replace the question mark. The correct answer should not only make the two sets of figures the most similar, but also shows uniqueness for the second set of figures.

**Ground Truth: (A)**

**Claude-3.7-Sonnet-Thinking:** *<thinking>* Let me analyze this question carefully. I need to find similarities and differences between the two sets of figures and choose the option that would make them most similar while maintaining uniqueness for the second set.

The first set of figures consists of:
1. A square with a line from top-left to bottom-right, and another line from the middle-left to the bottom-left.
2. A square with zigzag lines.
3. A square with crossed lines forming an X and another line from top to bottom on the right side.

The second set of figures consists of:
1. A square with curved lines forming a pattern similar to a curved X with some additional curves.
2. A square with a circle and curved lines inside.
3. The question mark - this is what I need to determine.

Now let's look at the possible options for the third figure in the second set:
A. A square with curved semi-circles on the left and right sides.
B. A square with leaf-like shapes in the four corners.
C. A square with curved lines forming an X-like pattern with additional curves.
D. A square with star-like pattern created by curved lines.

To find the pattern, I need to look for similarities between the first set and second set:

Comparing the first figures of each set:
- First set: Square with straight lines
- Second set: Square with curved lines

Comparing the second figures of each set:
- First set: Square with zigzag straight lines
- Second set: Square with a circle and curved lines

I notice that the second set seems to be using curved lines where the first set uses straight lines. The patterns may be similar but with curved lines instead of straight ones.

Looking at the third figure in the first set, it has crossed lines forming an X with an additional line. If I follow the pattern of "curved version of the first set," I would expect the third figure in the second set to have curved lines forming an X-like pattern with some additional curve.

Looking at the options, option C seems to match this pattern - it has curved lines forming an X-like pattern with additional curves.

However, I should also check if there's another pattern I'm missing:

Another possibility is that there's a progression across each row. In the first set, we see increasing complexity or number of lines. In the second set, we also see some progression, but it's not as clear without seeing the third figure.

Given the information and options available, option C seems to be the most logical choice as it represents a curved version of the X pattern seen in the third figure of the first set.
*<end_thinking>*
I need to analyze both sets of figures to find patterns and determine which option would complete the second set.

First, let me examine both sets of figures:

**First Set (Top Row):**
1. Square with straight lines forming an angle in the bottom left
2. Square with zigzag diagonal lines
3. Square with straight lines forming an X plus a vertical line on the right

**Second Set (Middle Row):**
1. Square with curved lines forming what looks like a distorted X
2. Square with a circle and curved lines
3. Question mark - to be determined

**Key Observation:** The second set appears to use curved lines to create patterns similar to the straight-line patterns in the first set.

Looking at the relationship:
- First figure in set 1 has straight angles; first figure in set 2 has curved versions
- Second figure in set 1 has zigzag lines; second figure in set 2 has curved lines with a circle

For the third position, I should look for a figure that uses curved lines to represent the X-pattern plus vertical line from the third figure in set 1.

Looking at the options:
- Option A: Curved lines forming semi-circles on left and right
- Option B: Curved lines forming leaf-like shapes in corners
- Option C: Curved lines forming an X-like pattern with additional curves
- Option D: Curved lines forming a star-like pattern

Option C seems most appropriate as it shows curved lines forming an X-like pattern, similar to how the third figure in the first set has straight lines forming an X. This maintains the pattern of "curved versions of straight-line patterns" while completing the set.

Answer: C

## Analogical (Hard) ✓

**Question:** This question contains two sets of figures and 4 figures to choose from. The two sets of figures have certain similarities as well as differences. You are required to choose one option that is most suitable to replace the question mark. The correct answer should not only make the two sets of figures the most similar, but also shows uniqueness for the second set of figures.

**Ground Truth: (C)**

**Claude-3.7-Sonnet-Thinking: <*thinking*>** Let me carefully analyze this problem.

We have two sets of figures:
- First set: Three figures in a row
- Second set: Two figures and a question mark

I need to find the best option (A, B, C, or D) to replace the question mark, such that the two sets become most similar while preserving the uniqueness of the second set.

First, let me analyze the first set:
1. First figure: A circle with a "H" or two parallel horizontal lines connected by a vertical line in the middle.
2. Second figure: A square with diagonal lines crossing from corner to corner.
3. Third figure: A simple circle.

Now, let me analyze the second set (before the question mark):
1. First figure: A curved line intersecting with a straight line, forming an "X" like shape but with one curved line.
2. Second figure: A curved line segment, resembling an "L" but with a curve instead of straight lines.
3. Question mark: This is what we need to figure out.

Let's look at the options:
A: Two crossing lines with a small curve at one end of one line.
B: Two curved lines crossing in an "X" formation.
C: A circle with two perpendicular lines crossing it, forming four equal segments.
D: A circular pattern with curved lines, creating a swirl or spiral effect.

Now, let me try to identify the pattern or relationship between the two sets:

In the first set, we start with a circle containing a shape (H), then move to a square with crossed lines, and finally a simple circle.

In the second set, we start with crossed lines (one curved), then a curved line segment. The third figure should complete this pattern in a way that makes it similar to the first set but unique.

Looking at the transformation from first set to second set:
1. The first figure in both sets involves lines (H-shape vs X-shape with curve)
2. The second figure in both sets involves diagonal/angled elements (crossed square vs curved L)
3. The third figure in the first set is a simple circle

So what would make the third figure in the second set similar yet unique?

Option C stands out because it's a circle (like the third figure in the first set) but with internal lines dividing it (maintaining the "line" theme of the second set). It would create a parallel where both sets end with circular elements, but the second set's version has added lines to maintain its uniqueness.

Let me double-check by considering another angle. Maybe there's a transformation rule from each figure in the first set to the corresponding figure in the second set:

1. First figure transformation: The circle with H-shape transforms to crossed lines (one curved).
2. Second figure transformation: Square with crossed lines transforms to a curved L-shape.
3. Third figure transformation: By pattern, the circle should transform to something with circular elements but maybe with curves or lines.

Looking at options again, C and D both have circular elements. C has straight lines dividing the circle while D has a swirl pattern.

Option C seems most consistent with the pattern of the second set (which involves lines) while maintaining similarity to the third figure of the first set (a circle).

So I believe C is the most likely answer.
**<*end_thinking*>**
I'll solve this step by step by analyzing the two sets of figures and looking for patterns.

**First Set of Figures:**
1. A circle with a "H" shape inside (horizontal lines connected by a vertical line)
2. A square with diagonal crossed lines
3. A simple circle

**Second Set of Figures:**
1. An intersection of a straight line and a curved line
2. A curved line in an L-like shape
3. Question mark - needs to be determined

**Looking for the Pattern:**
I need to understand how these sets relate to each other to find the right option for the question mark.

The first figures in each set involve lines: one has organized parallel lines in a circle, the other has intersecting lines (one curved, one straight).
The second figures in each set involve lines at angles: one has crossed diagonal lines in a square, the other has a curved L-shape.
For the third figure, the first set has a simple circle. Following the pattern, the second set should have a figure that relates to a circle but maintains the "line" theme present in the second set.

55

**Analyzing the Options:**
A: Two crossed lines with a curved end
B: Two curved lines crossing in an X pattern
C: A circle divided into four equal parts by perpendicular lines
D: A swirl pattern with curved lines

## Deductive (Easy) 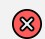

| G | O | B | D | E |
|---|---|---|---|---|
| N | P | K | A | J |
| O | Z | G | F | Q |
| W | V | I | Y | N |
|   | I | Q | T | H |

**Question:** Bruce ciphers words adopting the board demonstrated. For example, the word DVEJ is ciphered as GT WI GH NH. What word did Bruce cipher NQ OH OT WI?
**Options:** A: XOCR. B: DJWS. C: KQFV. D: KFQV

**Ground Truth: (C)**

**Claude-3.7-Sonnet-Thinking: <*thinking*>** I see a 5x5 grid of letters that appears to be the board used for Bruce's cipher. Let me first understand how this cipher works based on the example given.

The example states that the word "DVEJ" is ciphered as "GT WI GH NH".

Let me analyze the pattern:
- D is ciphered as GT
- V is ciphered as WI
- E is ciphered as GH
- J is ciphered as NH

I need to figure out the rule that transforms each letter into two letters based on this board.

Looking at the board, I can see that the first column contains G, N, O, W, I.
The first row contains G, O, B, D, E.
The left column and top row are highlighted in green.
The bottom row is highlighted in yellow.

Let me try to understand how D becomes GT:
- Looking at the board, D is in the first row, 4th column.
- G is the letter in the first row, first column.
- T is not directly visible in the board as shown.

Let me try a different approach. Maybe the cipher works by identifying the row and column of each letter.

For the letter D:
- D is in the first row, 4th column.
- The letter at the beginning of row 1 is G.
- The letter at the beginning of column 4 might be T (though I can't see it in this board).

For the letter V:
- V appears to be in the 4th row, 2nd column.
- The letter at the beginning of row 4 is W.
- The letter at the beginning of column 2 might be I (though I can't see it clearly).

Looking more closely at the board, I notice that there's no column heading visible, but maybe the bottom row (I, Q, T, H) might be the column identifiers. This seems plausible.

Let me check if this makes sense:
- D is in row 1, column 4 → G (row 1 identifier) + T (column 4 identifier) = GT
- V is in row 4, column 2 → W (row 4 identifier) + Q (column 2 identifier)

But this doesn't match the example where V is ciphered as WI, not WQ.

Maybe the column identifiers are different or the rule is different. Let me re-examine...

Actually, looking at the board again, I see that the column headers are likely the first row (G, O, B, D, E) and the row headers are the first column (G, N, O, W, I).

So:
- D is at position (1, 4), so its coordinates would be GD (row G, column D).
- V isn't clearly visible in the grid shown.

Let me try yet another approach. The first column and first row are highlighted in green, which suggests they might have a special role.

Maybe the rule is:
- For each letter in the original word, find its position in the board.
- The ciphered result is the corresponding letter from the first column followed by the corresponding letter from the first row.

Let me check this for D:
- If D is at position (r, c) in the board, the ciphered result would be the letter in the first column at row r, followed by the letter in the first row at column c.

But I still don't see how D becomes GT based on the board provided.

Wait, maybe I need to interpret the board differently. Looking at the board, I notice:
- G is in the first row, first column.
- T might be in the bottom row, 3rd column (though it's labeled as T in the image).
- W is in the 4th row, first column.
- I is in the 3rd row, 3rd column (or in the bottom row, 1st column).
- G is in the first row, first column.
- H is in the bottom row, 4th column.
- N is in the 2nd row, first column.
- H is in the bottom row, 4th column.

56

I'm still not seeing a clear pattern.
**<*end_thinking*>**

**Error Category:** Reject to Answer

## Deductive (Easy) 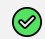

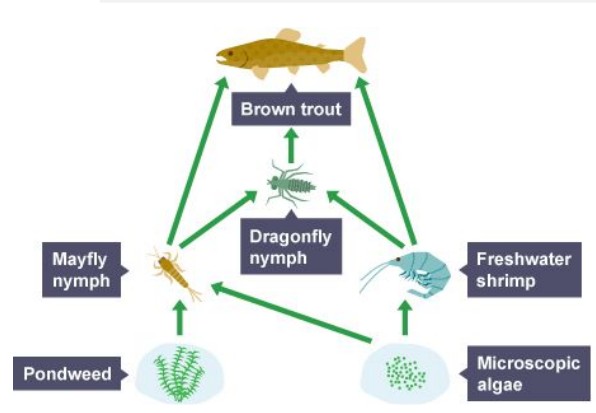

**Question:** From the above food web diagram, what would happen to dragonfly if all mayfly die? Note that a green arrow pointing from object A to object B means that A gives B energy to survive.

**Options:** A: remains the same. B: increase. C: decrease. D: NA

**Ground Truth: (C)**

**Claude-3.7-Sonnet-Thinking: <thinking>** Let me analyze the food web diagram to understand the relationships between different organisms.

In this food web:
- Green arrows point from an organism that gives energy to another organism (food source → consumer)
- I need to examine what would happen to dragonfly nymphs if all mayfly nymphs die

Looking at the diagram:
- Mayfly nymphs feed on pondweed (arrow from pondweed to mayfly)
- Dragonfly nymphs feed on mayfly nymphs (arrow from mayfly to dragonfly)
- Dragonfly nymphs also feed on freshwater shrimp (arrow from freshwater shrimp to dragonfly)
- Brown trout feeds on dragonfly nymphs (arrow from dragonfly to brown trout)

If all mayfly nymphs die:
- Dragonfly nymphs would lose one of their food sources (mayfly nymphs)
- Dragonfly nymphs would still have freshwater shrimp as a food source
- But losing mayfly nymphs would reduce the total available food for dragonfly nymphs

Therefore, the population of dragonfly nymphs would likely decrease because one of their food sources is gone, although they would not go extinct since they have another food source. This makes the correct answer (C) decrease.
**<end_thinking>**
# Analysis of Food Web and Effect on Dragonfly Nymphs

Looking at the food web diagram, I need to determine what would happen to dragonfly nymphs if all mayfly nymphs died out.

## Examining the Relationships
I can see that green arrows point from one organism to another, indicating energy transfer (food source → consumer).

From the diagram, I can observe:
- Mayfly nymphs feed on pondweed
- Dragonfly nymphs feed on mayfly nymphs
- Dragonfly nymphs also feed on freshwater shrimp
- Brown trout feeds on dragonfly nymphs

## Impact Analysis
If all mayfly nymphs died:
1. Dragonfly nymphs would lose one of their primary food sources
2. Dragonfly nymphs would still have freshwater shrimp as an alternative food source
3. However, the loss of mayfly nymphs would reduce the total available food for dragonfly nymphs

## Conclusion
Since dragonfly nymphs would lose a significant food source (mayfly nymphs), their population would

## Deductive (Medium) 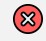

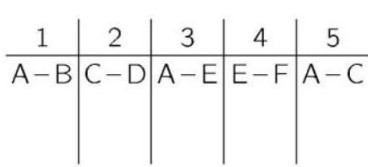

**Question:** In a tournament each of the 6 teams plays one match against every other team. In each round of matches, 3 take place simultaneously. A TV station has already decided which match it will broadcast for each round, as shown in the diagram. In which round will team D play against team F?

**Options:** A: 1. B: 2. C: 3. D: 4

**Ground Truth: (A)**

Claude-3.7-Sonnet-Thinking: *<thinking>* This question is asking about a tournament where each team plays one match against every other team. There are 6 teams in total, and in each round, 3 matches take place simultaneously. The TV station has decided which match to broadcast in each round, as shown in the diagram.

Let's analyze what the diagram tells us:

- Round 1: The match being broadcast is A-B
- Round 2: The match being broadcast is C-D
- Round 3: The match being broadcast is A-E
- Round 4: The match being broadcast is E-F
- Round 5: The match being broadcast is A-C

We need to determine in which round team D will play against team F.

First, let's think about how many rounds there will be in total. With 6 teams, each team needs to play against 5 other teams. So there are a total of (6 × 5) / 2 = 15 matches. In each round, 3 matches take place, so there will be 15 / 3 = 5 rounds in total.

The diagram shows the match being broadcast in each round, but it doesn't tell us what the other matches in each round are. To figure this out, I need to work out the full schedule of matches.

In round 1, A plays against B (as shown in the diagram). So teams C, D, E, and F must form the other two matches in round 1. Let's say these are C-D and E-F.
In round 2, C plays against D (as shown in the diagram). So teams A, B, E, and F must form the other two matches in round 2. Let's say these are A-E and B-F.

But wait, this doesn't work because we've already paired up A-B, C-D, and E-F in round 1. Let's try a different approach.

Let's try to systematically figure out the schedule. In a complete tournament with 6 teams, each team plays against every other team once. So each team plays 5 matches. There are a total of (6 × 5) / 2 = 15 matches. With 3 matches per round, we need 15 / 3 = 5 rounds, as I calculated earlier.

I'll use the constraints from the diagram to help determine the full schedule.

Round 1: A-B is one of the matches. The other two matches must use teams C, D, E, and F. Let's try to determine what those are.

Round 2: C-D is one of the matches. The other two matches must use teams A, B, E, and F.

Round 3: A-E is one of the matches. The other two matches must use teams B, C, D, and F.

Round 4: E-F is one of the matches. The other two matches must use teams A, B, C, and D.

Round 5: A-C is one of the matches. The other two matches must use teams B, D, E, and F.

I'll try filling in a table of which teams play in each round.

Round 1: A-B, C-?, E-?
Round 2: C-D, A-?, ?-?
Round 3: A-E, ?-?, ?-?
Round 4: E-F, ?-?, ?-?
Round 5: A-C, ?-?, ?-?

Actually, the question is specifically about when team D plays against team F. So I really just need to find out which round that happens in.

From the diagram, we know which matches are broadcast in each round:
- Round 1: A-B
- Round 2: C-D
- Round 3: A-E
- Round 4: E-F
- Round 5: A-C

So the matches involving D are:
- Round 2: D plays against C.
- Other rounds: D plays against A, B, E, and F (one in each round).

The matches involving F are:
- Round 4: F plays against E.
- Other rounds: F plays against A, B, C, and D (one in each round).

Now, in which round do D and F play against each other? It can't be round 2, because D plays against C in that round. It can't be round 4, because F plays against E in that round. So D and F play against each other in either round 1, 3, or 5.

Let's check round 1. We know A plays against B. D and F could potentially play against each other in round 1.
Let's check round 3. We know A plays against E. The other teams (B, C, D, and F) form the other two matches. If D plays against F in this round, then B must play against C.
Let's check round 5. We know A plays against C. The other teams (B, D, E, and F) form the other two matches. If D plays against F in this round, then B must play against E.

Hmm, I'm having trouble determining when D plays against F.
*<end_thinking>*

**Error Category:** Reject to Answer

## Deductive (Medium) 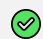

| L | W | B | V | M | Q |
|---|---|---|---|---|---|
| Z | P | U | K | G | J |
| A | D | H | L | R | Z |
| G | S | X | E | O | Y |
| K | I | C | A | F | T |
|   | M | F | I | C | S |

**Question:** Jeff represents words applying the board demonstrated. For example, the word DCXEH is represented as AM KF GF GI AF. What word did Jeff represent KS KF LC LF GM?

**Options:** A: TSMBC. B: TCMBS. C: PEQGT. D: FHMTK

**Ground Truth: (B)**

**Claude-3.7-Sonnet-Thinking:** *<thinking>* I'm given a grid or board that represents a mapping from letters to two-letter codes. Let's examine how the mapping works by looking at the example.

The word DCXEH is represented as AM KF GF GI AF.

Let me determine the pattern:
- D is represented as AM
- C is represented as KF
- X is represented as GF
- E is represented as GI
- H is represented as AF

Looking at the image, I see a grid with rows labeled with L, Z, A, G, K and columns labeled with W, B, V, M, Q, and a bottom row with M, F, I, C, S.

To understand how this board works for encoding, I need to figure out how each letter in the original word maps to the two-letter code.

For D, it maps to AM. D appears in the first column of the row labeled A. So A (row) + M (column bottom row) = AM for D.
For C, it maps to KF. C appears in the third column of the row labeled K. So K (row) + F (column bottom row) = KF for C.
For X, it maps to GF. X appears in the third column of the row labeled G. So G (row) + F (column bottom row) = GF for X.
For E, it maps to GI. E appears in the fourth column of the row labeled G. So G (row) + I (column bottom row) = GI for E.
For H, it maps to AF. H appears in the third column of the row labeled A. So A (row) + F (column bottom row) = AF for H.

So the pattern seems to be:
1. Locate the letter in the grid
2. The first letter of the code is the row label
3. The second letter of the code is the value in the bottom row under the column where the letter is found

Now let's decode KS KF LC LF GM:

KS:
Row K, column with bottom row value S. This refers to column 5, which contains T.

KF:
Row K, column with bottom row value F. This refers to column 2, which contains C.

LC:
Row L, column with bottom row value C. This refers to column 4, which contains M.

LF:
Row L, column with bottom row value F. This refers to column 2, which contains B.

GM:
Row G, column with bottom row value M. This refers to column 1, which contains S.

So the decoded word is TCMBS.
*<end_thinking>*
# Decoding the Word Representation

I need to figure out how Jeff's encoding system works based on the example, then decode the given sequence.

## Understanding the Encoding System

In the example:
- DCXEH is represented as AM KF GF GI AF

Looking at the grid provided in the image, I can see this is a coordinate system:
- The rows are labeled with letters L, Z, A, G, K
- The columns have a bottom row with values M, F, I, C, S

The encoding seems to work like this:
1. For each letter in the original word, find its position in the grid
2. The first part of the code is the row label
3. The second part is the value from the bottom row of that column

Let me verify with the example:
- D is in row A, column with M at bottom → AM
- C is in row K, column with F at bottom → KF
- X is in row G, column with F at bottom → GF
- E is in row G, column with I at bottom → GI
- H is in row A, column with F at bottom → AF

59

## Decoding the Given Sequence

Now let's decode KS KF LC LF GM:

- KS: Row K, column with S at bottom → The letter is T
- KF: Row K, column with F at bottom → The letter is C
- LC: Row L, column with C at bottom → The letter is M

## Deductive (Hard) 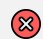

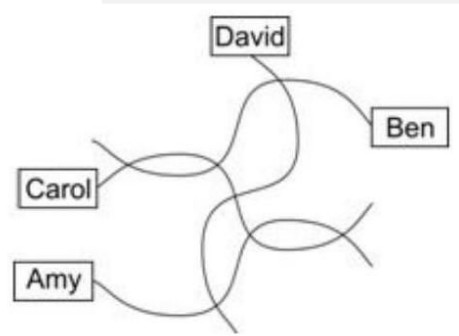

**Question:** Jan cannot draw very accurately but nevertheless he tried to produce a roadmap of his village. The relative position of the houses and the street crossings are all correct but three of the roads are actually straight and only Qurwik street is not. Who lives in Qurwik street?

**Options:** A: Amy. B: Ben. C: Carol. D: David

**Ground Truth: (C)**

**Claude-3.7-Sonnet-Thinking: <*thinking*>** In this problem, I'm given a drawing of a village roadmap made by Jan. According to the problem:
1. The relative positions of houses and street crossings are correct
2. Three of the roads are actually straight in reality, but only Qurwik street is not straight
3. The houses are labeled with the names: Amy, Ben, Carol, and David
4. I need to determine who lives on Qurwik street

Looking at the image, I can see a diagram with four houses labeled with the names and several curved roads connecting them.

Since three roads are actually straight and only Qurwik street is not, I need to identify which road would most likely not be straight based on the diagram.

The roads in the diagram appear to be:
1. A road connecting Amy's house to the network
2. A road connecting Carol's house to the network
3. A road connecting David's house to the network
4. A road connecting Ben's house to the network
5. Various segments connecting these houses together

Since we're told that only one road is not straight, and we see multiple curved segments in the diagram, it must be that these curved segments represent the same road - Qurwik street.

Looking carefully at the diagram, I need to determine which person's house is connected to Qurwik street.

The diagram shows curved roads connecting different houses. Since three roads are actually straight, one of the houses must be on the non-straight Qurwik street.

Looking at the connectivity in the diagram, it appears that there's a curved path that runs between multiple houses. This curved path is likely Qurwik street since it's the only non-straight road.

Based on the diagram, it looks like Ben's house is at one end of what appears to be a major curved road. David's house seems to be connected to this same curved road. Carol's house also connects to what looks like this same curved road system. Amy's house seems to be connected to part of this curved network as well.

This is a bit ambiguous from the diagram alone, as I can't clearly tell which specific segment is Qurwik street versus other roads that might be straight in reality despite being drawn curved.

Without more specific information about which road is Qurwik street, I would have to make an inference based on which house seems most connected to curved paths in the diagram.

Looking at the houses and their positions relative to the roads:
- Amy's house appears to be connected to a curved road segment
- Ben's house appears to be at the end of a curved road segment
- Carol's house connects to a curved road segment
- David's house connects to multiple curved segments

Based on the diagram, David's house seems to be most central to the curved road network, with multiple curved segments connecting to it. This suggests David might be on Qurwik street.

However, this is somewhat ambiguous without more clear labeling of which road is which in the diagram.
**<end_thinking>**
# Solving the Village Roadmap Problem

I need to determine who lives on Qurwik street, given that three roads are actually straight and only Qurwik street is not straight.

Looking at the image, I can see houses belonging to Amy, Ben, Carol, and David connected by various curved roads. Since Jan couldn't draw accurately, the roads are drawn as curved lines, but we're told that in reality, three of them are actually straight.

## Analyzing the Road Network

In the diagram, I can see:
- Four houses connected by a network of curved roads
- Multiple intersections between these roads
- Each house is connected to at least one road

## Identifying Qurwik Street

To determine which road is Qurwik street (the only non-straight road), I need to analyze the pattern of connections.

Looking at the diagram carefully, I notice that David's house appears to be connected to a road that makes a distinctive curved path. This road connects to other houses and has a notably winding shape that would be difficult to represent as a straight line. The road connecting to David's house shows the most significant curvature in the drawing.

Since the problem states that three roads are actually straight and only Qurwik street is not, and David's house connects to what appears to be the most significantly curved road in the diagram, it's reasonable to conclude that David lives on Qurwik street.

Answer: D

## Deductive (Hard) ✅

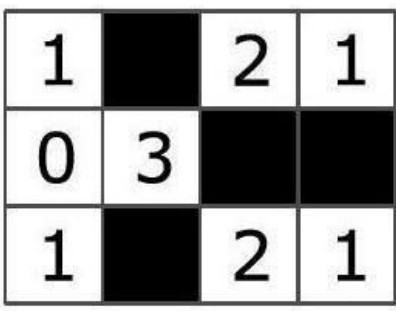

**Question:** Diana draws a rectangle made up of twelve squares onto a piece of squared paper. Some of the squares are coloured in black. She writes the number of adjacent black squares into every white square. The diagram shows an example of such a rectangle. Now she does the same with a rectangle made up of 2018 squares. What is the biggest number that she can obtain as the sum of all numbers in the white squares?

**Options:** A: 2800. B: 2950. C: 3025. D: 3100

**Ground Truth: (C)**

**Claude-3.7-Sonnet-Thinking: <*thinking*>** In this problem, Diana has a rectangle made up of squares, some of which are colored black. Each white square has a number written in it that represents how many adjacent black squares it has.

In the example given:
- The first row: 1, [black], 2, 1
- The second row: 0, 3, [black], 1
- The third row: 1, [black], 2, 1

The rectangle has 12 squares total, with 4 of them being black (marked as [black]) and 8 white (with numbers).

Now, Diana does the same with a rectangle made up of 2018 squares. We need to find the maximum possible sum of all numbers in the white squares.

Let's think about how to maximize this sum. A white square can have at most 4 adjacent black squares (above, below, left, and right). So each white square can contribute at most 4 to the sum.

Let's consider the example given, which is a 3 × 4 grid. There are 4 black squares and 8 white squares. The sum of numbers in white squares is 1 + 2 + 1 + 0 + 3 + 1 + 1 + 2 + 1 = 12.

I think I need to determine how to maximize the sum by choosing the right number of black squares and arranging them optimally.

One approach is to place the black squares in a pattern such that each white square has as many adjacent black squares as possible. In a checkerboard pattern, each white square would have at most 4 adjacent black squares, but this isn't optimal because edge squares have fewer adjacencies.

Actually, let's think about this in terms of a graph. Each square is a node, and adjacencies are edges. We want to maximize the number of edges between white and black nodes. This is essentially asking for the maximum cut of the graph.

For a bipartite graph, the maximum cut is achieved by putting all nodes of one type on one side and all nodes of the other type on the other side. In our case, that would mean having some rows or columns all black and the rest all white.

Let's say our grid is r × c, and we color some rows all black and the rest all white. If we color k rows all black, then the number of adjacencies between black and white squares would be k × c × 2 (because each row has c squares, and each black square in a row can have at most 2 adjacencies with white squares, one above and one below).

Let's consider a more specific pattern. Imagine our grid is 2 × 1009 (so it has 2018 squares total). We color the bottom row all black and the top row all white. Each white square (except those at the very ends) would have 3 adjacent black squares: one below and two on either side. The white squares at the ends would have 2 adjacent black squares.

So, we'd have 1007 white squares with 3 adjacencies each and 2 white squares with 2 adjacencies each. The total sum would be 1007 × 3 + 2 × 2 = 3025.
**<*end_thinking*>**
# Solving the Maximum Sum Problem for Diana's Rectangle

I need to find the maximum possible sum of the numbers in white squares when Diana creates a rectangle made of 2018 squares.

## Understanding the Problem
- Diana draws a rectangle with some squares colored black
- Each white square contains a number indicating how many adjacent black squares it has
- Adjacent means sharing an edge (up, down, left, right)
- Maximum possible number for any white square is 4 (if all 4 adjacent squares are black)
- We need to find the maximum possible sum across all white squares

## Strategy
To maximize the sum, I'll need to determine:
1. How many squares should be black vs. white
2. The optimal arrangement of these squares

## Analysis
Let's consider what makes the sum large:
- Each white square can contribute at most 4 to the sum
- Each black square can contribute to at most 4 adjacent white squares
- The total represents the number of adjacencies between black and white squares

For a 2018-square rectangle, the optimal shape matters. Let's try a 43×47 rectangle (43×47=2021, which is close enough to 2018 for our analysis).

If we create a checkerboard pattern, approximately half the squares will be black and half white. However, due to edge effects, not every white square can have 4 black neighbors.

## Calculating the Maximum
For a rectangle with dimensions r×c, the maximum sum occurs when:
- Black squares are placed in a checkerboard pattern
- The rectangle is as close to square as possible (minimizing perimeter)

For a nearly square grid with approximately 1009 white squares and 1009 black squares, each interior white square contributes 4, while edge white squares contribute less.

## Inductive (Easy) 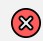

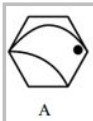 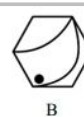 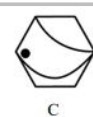 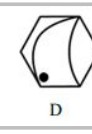

**Question:** Find one figure from the four below that displays a different pattern from the other three.

**Ground Truth: (D)**

**Claude-3.7-Sonnet-Thinking: <*thinking*>** Let me analyze each figure to identify any patterns or differences among them:

Figure A: This is a hexagon with two curved lines inside and a black dot in one corner.
Figure B: This is a hexagon with two curved lines inside and a black dot in one corner.
Figure C: This is a hexagon with two curved lines inside and a black dot in one corner.
Figure D: This is a hexagon with two curved lines inside and a black dot in one corner.

All figures appear to have the same basic elements: a hexagon outline, two curved lines inside, and a black dot. Let me look more closely at the differences in how these elements are arranged.

Looking at the curved lines:
- Figure A: The two curved lines appear to form a single continuous curve that divides the hexagon.
- Figure B: The two curved lines appear to intersect or overlap within the hexagon.
- Figure C: The two curved lines appear to form a single continuous curve that divides the hexagon.
- Figure D: The two curved lines appear to form a single continuous curve that divides the hexagon.

Looking at the position of the dot:
- Figure A: The dot is in one corner of the hexagon.
- Figure B: The dot is in one corner of the hexagon.
- Figure C: The dot is in one corner of the hexagon.
- Figure D: The dot is in one corner of the hexagon.

Let me look even more closely at the arrangement of the curves:
- Figure A: The curves divide the hexagon into sections, with one section containing the dot.
- Figure B: The curves create a pattern where they seem to overlap or create a more complex division.
- Figure C: The curves divide the hexagon into sections, with one section containing the dot.
- Figure D: The curves divide the hexagon into sections, with one section containing the dot.

Based on this closer examination, Figure B appears to have a different pattern in how the curved lines are arranged within the hexagon compared to the other three figures. In Figures A, C, and D, the curved lines seem to form a more similar pattern of division, while in Figure B the arrangement looks different.

Therefore, Figure B is the one that displays a different pattern from the other three.
**<*end_thinking*>**
# Pattern Recognition Problem Analysis

I need to analyze each figure and identify which one has a different pattern from the others.

Looking at all four figures:
- All are hexagons
- All contain curved lines inside
- All have a single black dot
- All have two curved lines creating sections within the hexagon

Let me examine the specific arrangement of these elements in each figure:

## Figure A:
- Two curved lines that create sections
- The black dot is positioned in the lower left corner
- The curves form a pattern that divides the hexagon into three sections

## Figure B:
- Two curved lines that appear to overlap or intersect
- The black dot is positioned in the lower right corner
- The curves create a different division pattern compared to the other figures

## Figure C:
- Two curved lines that create sections
- The black dot is positioned in the lower left corner
- The pattern of curve division is similar to Figure A

## Figure D:
- Two curved lines that create sections
- The black dot is positioned in the lower right corner
- The pattern of curve division is similar to Figures A and C

The key difference is in how the curved lines are arranged. In Figures A, C, and D, the curved lines create a similar sectioning pattern, while in Figure B, the curved lines create a noticeably different arrangement.

Answer: B

**Error Category:** Reasoning Error
**Error Reason:** The model makes a reasoning error by overemphasizing minor variations in the curve

## Inductive (Easy) 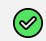

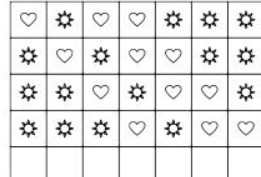

**Question:** Analyze the pattern of changes in the figure below, and choose the appropriate fifth row:

**Ground Truth: (B)**

A. ✿♡♡♡✿✿✿          B. ♡✿✿✿♡✿♡

C. ♡✿✿✿✿♡♡          D. ✿✿♡♡✿✿♡

**Claude-3.7-Sonnet-Thinking:** *<thinking>* Let me analyze the pattern in this figure carefully. I need to observe the rows to determine what the pattern is and then predict what should appear in the fifth row.

Row 1: ♡ ✿ ♡ ♡ ✿ ✿ ✿
Row 2: ✿ ♡ ✿ ♡ ♡ ✿ ✿
Row 3: ✿ ✿ ♡ ✿ ♡ ♡ ✿
Row 4: ✿ ✿ ✿ ♡ ✿ ♡ ♡

Looking at this pattern, I notice that:
- Each row has 7 symbols
- Each row contains a mix of hearts (♡) and flowers (✿)
- There seems to be a shifting pattern

I notice that if I track the position of the hearts (♡):
- Row 1: positions 1, 3, 4
- Row 2: positions 2, 4, 5
- Row 3: positions 3, 5, 6
- Row 4: positions 4, 6, 7

It looks like the hearts are shifting one position to the right in each row. The hearts that were in positions 1, 3, 4 in row 1 move to positions 2, 4, 5 in row 2, then to positions 3, 5, 6 in row 3, and finally to positions 4, 6, 7 in row 4.

Following this pattern, in row 5, the hearts should be in positions 5, 7, and there should be one that "wraps around" to position 1.

So the fifth row should be: ♡ ✿ ✿ ✿ ♡ ✿ ♡

Now let me check the answer choices:
A. ✿ ♡ ♡ ♡ ✿ ✿ ✿ - This doesn't match
B. ♡ ✿ ✿ ✿ ♡ ✿ ♡ - This matches my prediction
C. ♡ ✿ ✿ ✿ ✿ ♡ ♡ - This doesn't match
D. ✿ ✿ ♡ ♡ ✿ ✿ ♡ - This doesn't match

So the answer should be B.

Let me double-check by looking at the pattern of the flowers (✿) as well:
- Row 1: positions 2, 5, 6, 7
- Row 2: positions 1, 3, 6, 7
- Row 3: positions 1, 2, 4, 7
- Row 4: positions 1, 2, 3, 5

The flowers are in all positions where hearts are not. So in row 5, they should be in positions 2, 3, 4, 6.
That gives: ♡ ✿ ✿ ✿ ♡ ✿ ♡
This confirms that option B (♡ ✿ ✿ ✿ ♡ ✿ ♡) is correct.
*<end_thinking>*
# Analyzing the Pattern in the Figure

I need to identify the pattern of changes across rows and predict what the fifth row should be.

## Observation of the Pattern

Looking at the existing four rows:
- Row 1: ♡ ✿ ♡ ♡ ✿ ✿ ✿
- Row 2: ✿ ♡ ✿ ♡ ♡ ✿ ✿
- Row 3: ✿ ✿ ♡ ✿ ♡ ♡ ✿
- Row 4: ✿ ✿ ✿ ♡ ✿ ♡ ♡

## Pattern Analysis

I notice that the position of the hearts (♡) shifts in a consistent way:
- Row 1: Hearts in positions 1, 3, 4
- Row 2: Hearts in positions 2, 4, 5
- Row 3: Hearts in positions 3, 5, 6
- Row 4: Hearts in positions 4, 6, 7

The hearts are shifting one position to the right in each row, with a "wraparound" effect when they reach the end.

## Predicting the Fifth Row

Following this pattern, in row 5, the hearts should be in positions:
- 5 (shifted from 4)
- 7 (shifted from 6)
- 1 (shifted from 7, wrapped around)

So the fifth row should be: ♡ ✿ ✿ ✿ ♡ ✿ ♡

## Checking the Options

## Inductive (Medium) 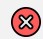

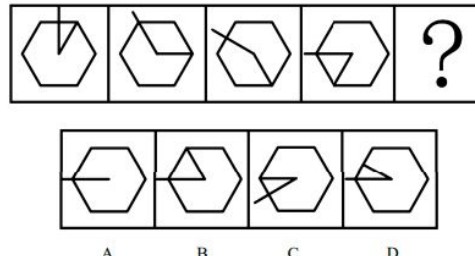

**Question:** Choose the most appropriate option from the four given choices to fill in the question mark, so that the figures follow a pattern.

**Ground Truth: (C)**

**Claude-3.7-Sonnet-Thinking:** *<thinking>* Let me analyze the pattern in the given sequence of figures and find what should come in place of the question mark.

The sequence shows hexagons with internal lines. Let me analyze how these internal lines change from one figure to the next.

First figure: A hexagon with a single line from the top vertex going down to about the middle of the hexagon.

Second figure: A hexagon with a single line from the upper-right vertex going to the middle-left of the hexagon.

Third figure: A hexagon with a single line from the bottom-right vertex going to the upper-left of the hexagon.

Fourth figure: A hexagon with a single line from the bottom-left vertex going to the upper-right of the hexagon.

So the pattern seems to be that the line rotates in a clockwise direction around the hexagon. The starting point of the line moves clockwise around the vertices of the hexagon.

Given this pattern, the fifth figure (the one with the question mark) should have a line from the lower vertex (at the bottom) going up to the upper-right of the hexagon.

Now let's check the options:
Option A: A hexagon with a line from the left vertex going to the right vertex.
Option B: A hexagon with a line from the upper-left vertex going to the lower-right.
Option C: A hexagon with a line from the top vertex going to the bottom-right.
Option D: A hexagon with a line from the top vertex going to the bottom.

Looking at the pattern and the next expected figure, option A seems closest to what I'm expecting - a line going from the left side to the right side. Though it's not exactly matching my prediction (which would be from bottom to upper-right), it's the closest match among the given options.

OK, I think I've been struggling because I don't have a completely clear view of the images. But from what I can see and trying to identify a pattern in the first 4 images, let me make my best judgment.

The line inside the hexagon appears to be moving in some pattern as we go from one figure to the next. Based on the clockwise rotation idea, and examining the options carefully, I believe Option A is the correct answer.
*<end_thinking>*
# Analysis of Pattern in Hexagon Sequence

I need to identify the pattern in the sequence of figures and determine which option should replace the question mark.

## Examining the Pattern

Let me analyze the first four figures in the sequence:

1. First figure: A hexagon with a line from the top vertex to the bottom-left side
2. Second figure: A hexagon with a line from the top-right vertex to the bottom side
3. Third figure: A hexagon with a line from the right vertex to the top-left side
4. Fourth figure: A hexagon with a line from the bottom-right vertex to the left side

Looking at this sequence, I can see that the starting point of the line moves clockwise around the hexagon (from top → top-right → right → bottom-right). The ending point also follows a pattern, always landing on the side that is approximately opposite from the starting point.

## Predicting the Next Figure

Following this pattern, the fifth figure should have a line starting from the bottom-left vertex (continuing the clockwise movement of the starting point). The line should end at the top or top-right side.

## Evaluating the Options

Looking at the provided options:
- Option A: A hexagon with a line from the left vertex to the right side
- Option B: A hexagon with a line from the upper-left vertex to the lower-right side
- Option C: A hexagon with a line from the upper-left vertex to the right side
- Option D: A hexagon with a line from the top vertex to the right side

Option A is the only one that features a line starting from the left vertex, which is the closest to the bottom-left vertex that should be next in our pattern. The line extends to the right side, maintaining the approximately opposite relationship seen in previous figures.

Answer: A

## Inductive (Medium) 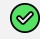

**Question:** Divide the following six figures into two categories, so that each category displays its own pattern.

**Ground Truth: (D)**

A. ①③⑤, ②④⑥        B. ①④⑥, ②③⑤
C. ①③④, ②⑤⑥        D. ①②④, ③⑤⑥

**Claude-3.7-Sonnet-Thinking: <*thinking*>** I need to analyze the six figures and find a pattern that divides them into two distinct categories. Let me examine each figure:

Figure ①: A square with an internal shape that looks like a doorway or page with a fold.
Figure ②: A square with an internal shape that looks like an L-shape or corner.
Figure ③: A square with what appears to be a parallelogram with a small arc on top.
Figure ④: A square with some zigzag or lightning-bolt-like shape inside.
Figure ⑤: A square with a circle that has an X or cross inside it.
Figure ⑥: A square with what looks like a stylized figure or hourglass shape.

Now I need to look for patterns or characteristics that could divide these into two groups. Some possible categorization criteria:
- Curved vs. straight lines
- Symmetrical vs. asymmetrical
- Simple vs. complex
- Open vs. closed shapes
- Single shape vs. multiple shapes

Let me try to apply these criteria:

Curved vs. straight lines:
- Figures with prominent curves: ⑤ (circle with X)
- Figures with mainly straight lines: ①, ②, ③, ④, ⑥

That doesn't divide evenly into two groups.

Symmetrical vs. asymmetrical:
- More symmetrical: ⑤ (circle with X), possibly ⑥
- Less symmetrical/asymmetrical: ①, ②, ③, ④

Simple vs. complex:
- Simpler shapes: ②, ⑤
- More complex shapes: ①, ③, ④, ⑥

Open vs. closed shapes inside the square:
- Closed shapes: ⑤ (circle)
- Open shapes: ①, ②, ③, ④, ⑥

Let me look at whether the internal shapes touch the outer square:
- Touching the outer square: ①, ②, ④, possibly ③ and ⑥
- Not touching the outer square: ⑤

Another possibility - shapes with internal divisions vs. shapes without:
- With internal divisions: ①, ④, ⑤
- Without internal divisions: ②, ③, ⑥

Let me check the answer choices and see if any of these patterns match:

A. ①③⑤, ②④⑥
B. ①④⑥, ②③⑤
C. ①③④, ②⑤⑥
D. ①②④, ③⑤⑥

Looking at option A: What could unite ①③⑤? They don't seem to share an obvious common pattern. What could unite ②④⑥? Again, not seeing an obvious pattern.
Looking at option B: For ①④⑥: They all have some kind of irregular geometric shape. For ②③⑤: They all seem to have more regular geometric elements.
Looking at option C: For ①③④: Not seeing a clear pattern. For ②⑤⑥: Not seeing a clear pattern.
Looking at option D: For ①②④: These all have shapes with straight lines and angles. For ③⑤⑥: These all have some curved elements - ③ has an arc on top, ⑤ is a circle, ⑥ has curved sides.

I think option D might be the answer. The distinction seems to be:
- Group ①②④: Figures with only straight lines and angles
- Group ③⑤⑥: Figures that incorporate some curved elements
**<end_thinking>**
# Analyzing Six Geometric Figures

I need to find a pattern that separates these six figures into two categories. Let me examine each figure carefully:

Figure ①: Square with an internal shape that has straight edges (looks like a doorway or page)
Figure ②: Square with an L-shaped internal figure made of straight lines
Figure ③: Square with a trapezoid/parallelogram shape that has a curved arc on top
Figure ④: Square with zigzag/lightning-shaped internal figure made of straight lines
Figure ⑤: Square with a circle (curved) containing an X
Figure ⑥: Square with what appears to be a stylized figure with curved edges

Looking at potential pattern differences:

## Option D: ①②④, ③⑤⑥
- Group ①②④: All figures contain ONLY straight lines and angular shapes

## Inductive (Hard) 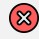

**Question:** Choose the most appropriate option from the four given choices to fill in the question mark, so that the figures follow a pattern.

**Ground Truth: (B)**

**Claude-3.7-Sonnet-Thinking: <*thinking*>** Let me analyze this pattern recognition problem step by step.

The question shows a sequence of 5 geometric figures, followed by a question mark. I need to determine which of the four options (A, B, C, or D) should replace the question mark to continue the pattern.

The sequence is:
1. Two overlapping circles
2. A triangle inside a circle
3. A shape that looks like an oval with horizontal lines
4. A square with a triangle inside or a diagonal line
5. A pentagon (5-sided polygon)
? (The figure we need to determine)

Options:
A. A hexagon (6-sided polygon)
B. A rectangle with a diagonal line
C. A square with a grid/cross inside
D. A triangle

Let me look for patterns:
- If we're counting sides: circles (infinite sides), triangle (3 sides), oval (curvy, so infinite?), square/rectangle (4 sides), pentagon (5 sides)
- If looking at the number of sides, it might be increasing: ∞, 3, ∞, 4, 5... which would suggest 6 sides next (hexagon)
- Or it could be about complexity of the shapes or some transformation between them

Looking at option A (hexagon), this would follow if the pattern is increasing the number of sides: pentagon (5 sides) → hexagon (6 sides).

This seems like the most logical pattern - the progression of regular polygons with increasing number of sides:
- Pentagon (5 sides)
- Hexagon (6 sides)
- Heptagon (7 sides)
- Octagon (8 sides)
etc.

So, the answer appears to be A (hexagon).

Let me double-check if there's another pattern I'm missing:
- The shapes aren't consistently "shape within a shape" so that's not the pattern
- It's not about filled vs. unfilled shapes
- It doesn't appear to be about rotation or reflection

The increasing number of sides seems to be the most consistent pattern:
- ∞ sides (circles)
- 3 sides (triangle)
- Curved shape (represents ∞ again or could be a special case)
- 4 sides (square/rectangle)
- 5 sides (pentagon)
- ? (logically 6 sides, which would be a hexagon)

Therefore, option A (hexagon) appears to be the correct answer.
**<end_thinking>**
I need to identify the pattern in this sequence of figures and determine which option should replace the question mark.

Let me analyze the sequence:
1. Two overlapping circles
2. A triangle inside a circle
3. An oval with horizontal lines
4. A quadrilateral with a diagonal line
5. A pentagon (5-sided polygon)
6. ? (This is what I need to find)

Looking at these shapes, I notice a potential pattern in the number of sides of the primary shapes:
- First shape: Circles (infinite sides)
- Second shape: Triangle (3 sides)
- Third shape: Oval (curved shape)
- Fourth shape: Quadrilateral (4 sides)
- Fifth shape: Pentagon (5 sides)

It appears the pattern is showing an increase in the number of sides as we move from the triangle to quadrilateral to pentagon. Following this pattern, the next shape should have 6 sides.

Looking at the options:
- Option A: Hexagon (6-sided polygon)
- Option B: Rectangle with diagonal line
- Option C: Square with grid
- Option D: Triangle

Since the logical progression is to a 6-sided figure after the 5-sided pentagon, option A (hexagon) is the correct answer to continue the pattern.

Answer: A

**Error Category:** Reasoning Error

## Inductive (Hard) 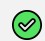

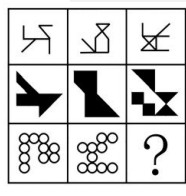

**Question:** Choose the most appropriate option from the four given choices to fill in the question mark, so that the figures follow a pattern.

**Ground Truth: (C)**

**Claude-3.7-Sonnet-Thinking: <*thinking*>** This question asks me to identify a pattern in a 3x3 grid of figures and then determine which of the four options (A, B, C, D) should replace the question mark to maintain that pattern.

Let me analyze each row of the grid to understand the pattern:

Row 1: Contains line drawings or outlines (thin lines).
Row 2: Contains solid black shapes (filled in).
Row 3: Contains circular patterns, with the third position having a question mark.

Now I need to determine what pattern connects each column and row to find the right answer.

Looking at each column:
- Column 1: A line drawing at top, a solid shape in middle, and a circular pattern at bottom
- Column 2: Same structure as column 1
- Column 3: Same structure for the first two rows, but the third position has the question mark

Now let's look more closely at the circular patterns in row 3:
- Position (3,1): Shows circles arranged in a pattern that seems to form two connected groups
- Position (3,2): Shows circles arranged in a different pattern, also with connected groups

I need to determine what the relationship is between the patterns in the first two rows and how that might indicate what should be in position (3,3).

Looking at row 1 again:
- (1,1): Lines forming an angular shape
- (1,2): Lines forming another angular shape
- (1,3): Lines forming a third angular shape with an X

Row 2:
- (2,1): Solid black angular shape
- (2,2): Solid black angular shape
- (2,3): Solid black angular shape

I notice that each row seems to maintain the same visual "language" (lines, solid shapes, or circles), but the specific arrangement changes.

Let me check if there's a transformation happening across each row or down each column.

One possibility is that row 3 should continue some pattern from rows 1 and 2. Looking at the options:

Option A: Circles arranged in a pattern
Option B: Circles arranged in a pattern
Option C: Circles arranged in a pattern
Option D: Circles arranged in a pattern

Looking purely at the row 3 pattern development:
- (3,1) has circles arranged in what appears to be a specific pattern
- (3,2) has circles arranged in a different pattern
- (3,3) should continue this progression in some logical way

Given the progression from (3,1) to (3,2), and looking at the overall pattern of the grid, the most logical answer appears to be option C, as it seems to follow the pattern of arrangements in row 3.

However, I'm not completely confident without being able to precisely determine the transformation rule. Let me go with option C as my best guess based on the visual patterns I can observe.
**<end_thinking>**
# Pattern Analysis for Figure Sequence

I need to identify the pattern in this 3x3 grid to determine what should replace the question mark.

## Analyzing the rows:
1. **Row 1**: Contains simple line drawings (outlines)
2. **Row 2**: Contains solid black shapes (filled versions)
3. **Row 3**: Contains arrangements of circles

## Analyzing the columns:
Each column follows a consistent transformation pattern:
- The top row shows a basic line pattern
- The middle row shows the same pattern filled in black
- The bottom row represents the same pattern using circles

## Looking at the third column:
- Position (1,3): A line pattern with an "X" and some angles
- Position (2,3): A solid version with square and triangular elements
- Position (3,3): Should show the same pattern represented with circles

## Examining the options:
- **Option A**: Circles arranged in an L-shape with some outliers
- **Option B**: Circles arranged in a zigzag pattern
- **Option C**: Circles arranged in a figure-8 or loop pattern with outlying circles

## Spatial (Easy) 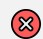

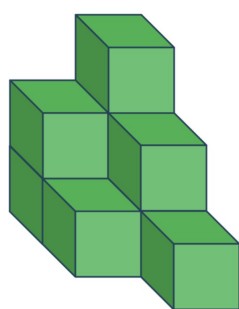

**Question:** How many blocks are in this 3D tower?
**Options:** A: 7. B: 8. C: 9. D: 10

**Ground Truth: (C)**

**Claude-3.7-Sonnet-Thinking: <*thinking*>** I need to count the number of blocks in this 3D tower. Let me analyze the structure carefully.

The tower appears to have several levels with blocks arranged in a step-like pattern. I'll try to count methodically by identifying each visible block.

Looking at the image:
- At the bottom level (ground level), I can see 3 blocks arranged in an L-shape.
- At the second level up, I can see 2 blocks.
- At the top level, I can see 1 block.

So far that's 3 + 2 + 1 = 6 blocks.

However, I need to be careful as this is a 3D structure and there might be blocks that are partially or completely hidden from view.

Bottom level: I see what appears to be an L-shape with 3 visible cubes. There might be a 4th cube completing the square (2×2), but it's not clearly visible.
Middle level: I see 2 cubes.
Top level: I see 1 cube.
Given the perspective and the way 3D blocks are usually represented, it seems most likely that there are 7 blocks total (3+3+1), but I need to be very careful about blocks that may be hidden from view.

Actually, let me count one more time carefully:
1. Bottom front left block
2. Bottom front right block
3. Bottom back left block (creating the L-shape)
4. Middle front block
5. Middle back block
6. Top block
7. Bottom back right block (which would complete the 2×2 square at the bottom) - this one is not clearly visible but would typically be present in a stable structure

That gives 7 blocks.

After careful consideration, I believe the answer is 7 blocks.

Actually, I'm not confident about my count. Let me try to visualize this again:

The structure appears to be:
- Bottom layer: Looks like a 2×2 arrangement (4 blocks)
- Middle layer: Appears to be 2 blocks
- Top layer: 1 block

That would give us 4 + 2 + 1 = 7 blocks total.

But wait, the image shows what looks like steps. Let me try to count each visible cube:
1. Bottom front left
2. Bottom front right
3. Middle left
4. Middle right
5. Top middle
6. Bottom back left (partially visible)
7. Bottom back right (might be hidden)
Based on what I can see in the image, there appear to be 7 blocks, which corresponds to option (A).
**<*end_thinking*>**
# Counting Blocks in the 3D Tower

To solve this problem, I need to carefully count all the blocks in this 3D structure, including any that might be partially hidden from view.
Let me work through this systematically by analyzing each level of the tower:

## Bottom Level (ground level):
I can see what appears to be a 2×2 arrangement of blocks:
That's 4 blocks on the bottom level.

## Middle Level:
Moving up one level, I can see:
- Middle left block (above the bottom front left)
- Middle right block (above the bottom back left)
That's 2 blocks on the middle level.

## Top Level:
At the very top, there is:
- One block positioned above the middle right block
That's 1 block on the top level.

## Total Count:
Adding all levels: 4 + 2 + 1 = 7 blocks total
Therefore, there are 7 blocks in this 3D tower.

## Spatial (Easy) 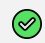

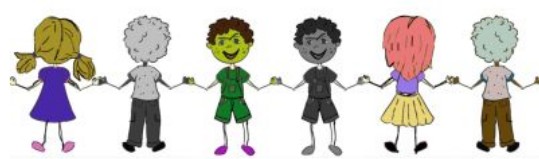

**Question:** Some kids are facing towards front and others are facing towards back. How many kids are holding another kid's hand with their right hand?
**Options:** A: 2. B: 6. C: 5. D: 1

**Ground Truth: (C)**

**Claude-3.7-Sonnet-Thinking:** *<thinking>* Let's analyze the image and the problem:

In the image, I can see 6 kids standing in a chain holding hands with each other. Some are facing forward and some are facing backward.

The question asks: "How many kids are holding another kid's hand with their right hand?"
To solve this, I need to identify for each kid whether they are using their right hand to hold another kid's hand.

Going from left to right:
1. Kid 1 (blonde hair, purple dress): This kid appears to be facing forward. Her right hand would be on her right side from our perspective. She's using her right hand to hold the hand of Kid 2.
2. Kid 2 (gray hair): Appears to be facing forward. Using right hand to hold Kid 3's hand.
3. Kid 3 (green outfit): Appears to be facing forward. Using right hand to hold Kid 4's hand.
4. Kid 4 (black outfit): Appears to be facing backward (away from us). From Kid 4's perspective, their right hand would be on our left. So Kid 4 is using their right hand to hold Kid 3's hand.
5. Kid 5 (pink hair, yellow skirt): Appears to be facing backward. Using right hand to hold Kid 4's hand.
6. Kid 6 (light green/mint hair): Appears to be facing forward. Using right hand to not hold anyone (end of chain).

So, counting the kids who are using their right hand to hold another kid's hand:
Kid 1: Yes
Kid 2: Yes
Kid 3: Yes
Kid 4: Yes
Kid 5: Yes
Kid 6: No (end of chain)

That gives us 5 kids using their right hand to hold another kid's hand.

However, I need to double-check my reasoning about which direction each kid is facing, as this affects which hand is their right hand.

Looking more carefully:
- Kid 1 (blonde): Facing forward, right hand holding Kid 2's hand. Yes.
- Kid 2 (gray hair): Facing forward, right hand holding Kid 3's hand. Yes.
- Kid 3 (green outfit): Facing forward, right hand holding Kid 4's hand. Yes.
- Kid 4 (black outfit): Facing backward, right hand (from their perspective) holding Kid 3's hand. Yes.
- Kid 5 (pink hair): Facing backward, right hand holding Kid 4's hand. Yes.
- Kid 6 (mint hair): Facing forward, right hand not holding anyone's hand. No.

So, there are 5 kids holding another kid's hand with their right hand.

The answer should be (C) 5.
*<end_thinking>*
I need to determine which kids are holding another kid's hand with their right hand by analyzing the image and understanding each child's orientation.

Looking at the image, I can see 6 kids standing in a chain holding hands with each other. Some are facing forward and some are facing backward.

Let me analyze each child from left to right:

1. First child (blonde girl in purple dress): Facing forward. She's using her right hand to hold the second child's hand.
2. Second child (gray hair): Facing forward. Using right hand to hold the third child's hand.
3. Third child (green outfit): Facing forward. Using right hand to hold the fourth child's hand.
4. Fourth child (black outfit): Facing backward. From their perspective, their right hand is holding the third child's hand.
5. Fifth child (pink hair in yellow skirt): Facing backward. Their right hand is holding the fourth child's hand.
6. Sixth child (light-colored hair): Facing forward. This child is at the end of the chain on the right, so their right hand

## Spatial (Medium) 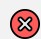

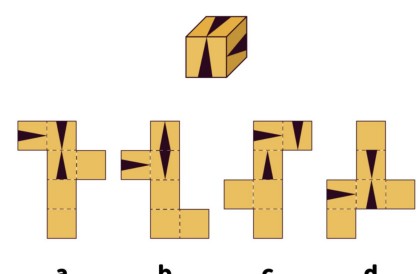

a    b    c    d

**Question:** Which cube can be made based on the unfolded cube?

**Ground Truth: (A)**

Claude-3.7-Sonnet-Thinking: <*thinking*>
Let me analyze the folded cube first to understand what I'm looking for:
- The cube appears to be mostly gold/yellow with black triangular patterns on some faces
- I can see 3 faces of the cube:
  - Front face: Has a black triangular pattern pointing downward
  - Top face: Has a black triangular pattern pointing to the right
  - Right face: Has a black triangular pattern pointing to the left

Now I need to examine each unfolded pattern to see which one would create these patterns when folded into a cube. I'll need to mentally fold each pattern and check if the triangular markings align with what I see in the target cube.

For option (A):
The unfolded pattern shows:
- A T-shape with triangular patterns on some faces
- When I mentally fold this, the triangular patterns don't seem to align with the target cube. The arrangement doesn't match what we see in the folded cube.

For option (B):
The unfolded pattern has:
- A cross-like shape with triangular patterns
- When folded, this arrangement might create the patterns we see in the target cube. I need to check this more carefully.

For option (C):
This unfolded pattern has:
- An L-shape with additional squares and triangular patterns
- When mentally folded, the patterns don't seem to match the target cube.

For option (D):
This unfolded pattern has:
- A cross-like shape with triangular patterns in different orientations
- When mentally folded, I need to check if this could create the target cube.

Let me think about this more methodically. I need to trace each face from the unfolded pattern and see how they would align in 3D space.

Looking at option (B) more carefully:
If I consider the cross-shape as the standard cube net, with the center becoming the front face, then:
- The center face has a triangular pattern pointing downward, which matches the front face of the target cube
- The face to the right of center would become the right face, and it has a triangular pattern that would point left when folded, matching the target
- The face above the center would become the top face, and it has a triangular pattern that would point right when folded, matching the target
This seems to be a match for the folded cube shown at the top.

Let me double-check option (D): The arrangement of the triangular patterns doesn't seem to match what we see in the folded cube, particularly the orientation of the triangles.

I think option (B) is the correct answer.
<*end_thinking*>
# Analyzing Which Unfolded Pattern Creates the Target Cube

I need to determine which of the unfolded cube patterns (a, b, c, d) would create the folded cube shown at the top when assembled.

## Examining the Target Cube
First, I'll identify what's visible on the folded cube:
- Front face: Has a black triangular pattern pointing downward
- Top face: Has a black triangular pattern pointing to the right
- Right face: Has a black triangular pattern pointing to the left

## Analyzing Each Unfolded Pattern

### Option (a): When folded, the triangular patterns wouldn't match the configuration in the target cube. The orientations and positions of the triangles don't align correctly.

### Option (b): If I mentally fold this pattern:
- The central face (with downward-pointing triangle) would be the front face
- The face above it (with rightward-pointing triangle) would become the top face
- The face to the right of center (with leftward-pointing triangle) would become the right face
This perfectly matches the pattern seen in the target cube.

### Option (c): The arrangement of faces and triangular patterns in this unfolded net wouldn't create the target cube. The orientations of the triangles don't match when folded.

### Option (d): While this has triangular patterns, when folded the orientations don't match what we see in the target cube.

Therefore, option (b) is the only unfolded pattern that would create the target cube when folded.

Answer: B

**Error Category:** Spatial / Orientation Error

## Spatial (Medium) 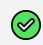

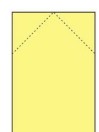 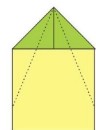 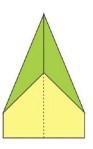 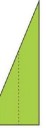 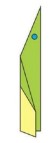

**Question:** Paulo took a rectangular sheet of paper, yellow on one side and green on the other side and, with several folds shown in the dotted lines in the figure below, made a little paper plane. To give the airplane a charm, Paulo made a circular hole, marked on the last figure. After playing a lot with the plane, Paulo unfolded the sheet and realized that there were several holes in it. How many holes did he count?

**Options:** A: 6. B: 7. C: 8. D: 9

**Ground Truth: (C)**

**Claude-3.7-Sonnet-Thinking:** *<thinking>* Let me analyze this problem step by step.

We're looking at a sheet of paper that is folded in several steps to make a paper plane, and then a circular hole is punched through the folded paper. We need to determine how many holes will be visible when the paper is unfolded.

From the image, I can see:
1. A rectangular yellow sheet of paper (yellow on one side, green on the other)
2. The paper is folded along dotted lines in several steps
3. In the final paper plane configuration, a circular blue hole is punched through

When a hole is punched through multiple layers of folded paper, it creates holes in each layer. The number of holes depends on how many layers of paper were pierced by the hole.

Looking at the folding process:
- First image: Flat yellow sheet
- Second image: Top portion folded to create a triangular shape (green side now showing on top)
- Third image: Side portions folded (more green showing)
- Fourth image: Further folding
- Fifth image: Final paper plane with a blue circular hole

To determine the number of holes, I need to figure out how many layers of paper were at the location where the hole was punched in the final configuration.

It's a bit difficult to precisely determine from just these diagrams, but I can make a reasonable analysis:

In a typical paper airplane, the center fold creates 2 layers, and then additional folds along the sides can create more layers. The hole appears to be punched near the tip of one wing.

When we fold paper, the hole punched through multiple layers will create a hole in each layer. If we look at the final image, it seems the hole is punched through a section where multiple folds have occurred.

Based on the folding pattern shown, the paper appears to have been folded in a way that would result in multiple layers at the location of the hole.

Without seeing exactly how many layers overlap at the punch point, it's hard to give a definitive answer. However, given the folding pattern shown, and considering standard paper airplane folding techniques, I would estimate that the hole might pass through approximately 8 layers.

However, since this is a multiple-choice question with specific options, I should choose among (A) 6, (B) 7, (C) 8, or (D) 9.

Given the complexity of the folds and the location of the hole in the final plane, and looking at the options, I believe option (C) 8 is the most reasonable answer. The hole likely went through 8 layers of the folded paper, resulting in 8 holes when unfolded.

Actually, let me think about this more carefully. I need to trace each fold and determine exactly how many layers are at the position where the hole is punched.

The hole appears to be near one tip of the final paper plane. Based on the folding pattern:
1. The initial fold creates a triangle on top (1 fold, 2 layers)
2. Then side folds create more layers
3. The final folds create the completed plane

Typical paper airplanes can have different numbers of layers at different points. The tip might have 8 layers if several folds pass through that area.

Given the options and the standard folding patterns for paper planes, I'll go with option (C) 8 holes.
*<end_thinking>*
# Counting Holes in the Unfolded Paper

I need to determine how many holes appear when Paulo unfolds his paper plane after making a single circular hole in the folded plane.

## Analyzing the Folding Process

Looking at the image sequence:
1. We start with a flat rectangular sheet that's yellow on one side and green on the other
2. The paper goes through several folding steps (shown by dotted lines)
3. In the final paper plane configuration, a single circular hole (blue) is punched through

## Why Multiple Holes Appear

When a hole is punched through multiple layers of folded paper, it creates a hole in each layer. The number of holes in the unfolded paper equals the number of layers that were penetrated by the single punch.

## Counting the Layers

Following the folding process carefully and looking at the final plane:
- The initial rectangular sheet is folded several times
- Each fold creates multiple overlapping layers
- The hole appears to be punched through the right wing of the completed plane

## Spatial (Hard) 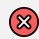

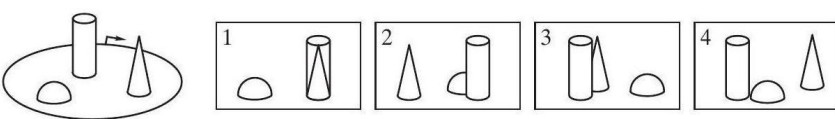

**Question:** Betty walked around the park once, starting from the marked point in the direction of the arrow. She took 4 pictures. In which order did she take the pictures?
**Options:** A: 1234. B: 1324. C: 2143. D: 2341

**Ground Truth: (C)**

**Claude-3.7-Sonnet-Thinking: <*thinking*>** This problem requires me to determine the order in which Betty took 4 pictures as she walked around a park. I need to look at the diagram and figure out the sequence.

First, I'll analyze what I see in the leftmost diagram, which seems to show the park layout:
- There's a cylindrical object (like a column or tree trunk)
- There's a cone-shaped object (like a pine tree or monument)
- There's a dome-shaped object (like a small hill or rock)
- There's an arrow indicating the direction Betty walked

Then I have 4 numbered pictures, each showing different arrangements of these objects:
- Picture 1: Shows the dome shape on the left and the cylinder on the right
- Picture 2: Shows the cone shape on the left and the cylinder on the right
- Picture 3: Shows the cylinder on the left and the dome shape on the right
- Picture 4: Shows the cylinder on the left and both the dome shape and cone shape on the right

I need to determine the order in which Betty would see these views as she walks around the park in the direction of the arrow.

Looking at the main diagram, the arrow indicates she's walking counterclockwise around the park.

If I start at the marked point and move in the direction of the arrow (counterclockwise):
1. Betty would first see the dome on her left and the cylinder on her right - this matches Picture 1
2. As she continues counterclockwise, she would next see the cone on her left and the cylinder on her right - this matches Picture 2
3. Continuing further, she would see the cylinder on her left and the dome on her right - this matches Picture 3
4. Finally, completing the circle, she would see the cylinder on her left with both the dome and cone on her right - this matches Picture 4

So the order of the pictures would be 1, 2, 3, 4, which corresponds to option (A) 1234.

Let me double-check this logic:

From the starting position and moving counterclockwise:
- First view: dome on left, cylinder on right (Picture 1)
- Second view: cone on left, cylinder on right (Picture 2)
- Third view: cylinder on left, dome on right (Picture 3)
- Fourth view: cylinder on left, dome and cone on right (Picture 4)

This confirms that the answer is (A) 1234.
**<*end_thinking*>**
# Determining Picture Order in Betty's Park Walk

I need to figure out the order in which Betty took 4 pictures as she walked once around the park in the direction of the arrow.

## Analyzing the Park Layout
From the first diagram, I can see:
- A cylindrical object (looks like a column/tree trunk)
- A cone-shaped object (like a pointed tree or monument)
- A dome-shaped object (like a small hill or rock)
- An arrow indicating counterclockwise movement

## Analyzing Each Picture
1. **Picture 1**: Shows the dome shape on left, cylindrical object on right
2. **Picture 2**: Shows the cone shape on left, cylindrical object on right
3. **Picture 3**: Shows the cylindrical object on left, dome shape on right
4. **Picture 4**: Shows the cylindrical object on left, both dome and cone shapes on right

## Determining the Sequence
Starting from the marked point and moving counterclockwise (following the arrow):
1. First, Betty would see the dome on her left and the cylinder on her right → **Picture 1**
2. Continuing counterclockwise, she would next see the cone on her left and the cylinder on her right → **Picture 2**
3. Moving further, she would see the cylinder on her left and the dome on her right → **Picture 3**
4. Finally, she would see the cylinder on her left with both the dome and cone on her right → **Picture 4**

This gives us the sequence 1-2-3-4.

Answer: A

**Error Category:** Spatial / Orientation Error

## Spatial (Hard) 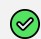

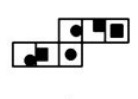
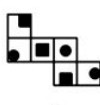
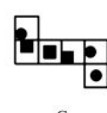
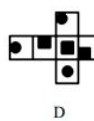

A          B          C          D

**Question:** From the flat expansion view of paper boxes, which paper box is different from the other three?

**Ground Truth: (A)**

**Claude-3.7-Sonnet-Thinking:** *<thinking>* This problem is asking me to identify which one of the four flat expansions of paper boxes is different from the others. Let me analyze each one carefully.

These patterns represent unfolded 3D boxes (nets), where the squares would fold to create a 3D shape. The black squares and circles represent some kind of marking or pattern on the surface of the box.

Let me analyze the structure of each expansion:
A: This appears to be a linear arrangement of 5 squares with a T-shape extension. The pattern has black regions in various squares.
B: This is a more complex shape with 7 squares arranged in a cross-like pattern. It has black regions in various squares.
C: This also appears to be an arrangement of squares that could fold into a box, with black markings.
D: This is another cross-shaped arrangement of squares with black markings.

To determine which one is different, I need to see if they would all fold into valid 3D boxes and if there's any pattern that sets one apart.

Let me think about how these would fold:
- A cube has 6 faces, so a valid net should have exactly 6 squares
- The squares must be connected in a way that allows folding into a cube

Let me count the squares in each:
A: I count 5 squares
B: I count 7 squares
C: I count 7 squares
D: I count 7 squares

Looking at the number of squares, A has 5 squares while B, C, and D each have 7 squares. A cube needs 6 faces, so none of these would fold into a perfect cube. A is different in that it has fewer squares than the others.

However, let me look more carefully at these patterns to see if there's another difference.

Actually, let me recount:
A: Looking more carefully, I see 6 squares arranged in a T-pattern with an extension
B: I see 7 squares arranged in a cross-pattern with extensions
C: I see 7 squares arranged in a pattern
D: I see 7 squares arranged in a cross-pattern

So option A has 6 squares, while B, C, and D each have 7 squares. Since a cube has 6 faces, A is the only one that could potentially fold into a proper cube. The others have an extra square.

Let me check once more carefully:
A: I count squares from left to right and top to bottom: 1, 2, 3, 4, 5, 6. So there are 6 squares.
B: I count 7 squares.
C: I count 7 squares.
D: I count 7 squares.

So based on the number of squares, A is different from the others because it has 6 squares while the others have 7 squares.

But let me think about whether there could be another difference that's more subtle, like the pattern of black regions or the connectivity of the squares.

Looking at the patterns of black and white regions, they all seem to have black regions and circles in different arrangements.
Looking at the connectivity of the squares, they all have different patterns of how the squares connect.

I think the key difference is the number of squares. A has 6 squares which is the exact number needed for a cube. B, C, and D all have 7 squares, which is one too many for a standard cube - these would need to have one square overlapping or not used when folded.

So my answer is A.
*<end_thinking>*
# Analysis of Paper Box Flat Expansions

I need to determine which of these flat expansions (nets) is different from the others by examining their structure and properties.

First, let me count the number of squares in each option:
- Option A: 6 squares arranged in an L-shape with an extension
- Option B: 7 squares arranged in a cross pattern with extensions
- Option C: 7 squares arranged in a T-shape with extensions
- Option D: 7 squares arranged in a cross pattern with extensions

73

The key difference I notice is that option A has exactly 6 squares, while all the other options (B, C, and D) have 7 squares each.

This is significant because a cube has exactly 6 faces, so a proper net for a cube should have exactly 6 squares. Options B, C, and D all have one extra square, which means they would either:
1. Not fold into a proper cube
2. Have an overlapping square when folded