# OpenReview forum: "VisualPuzzles: Decoupling Multimodal Reasoning Evaluation from Domain Knowledge"
_ICML.cc/2026/Conference — ICML 2026 regular_

### Official Review · Reviewer_vaqX · 2026-03-11

**Soundness:** 3
**Presentation:** 3
**Significance:** 3
**Originality:** 3
**Overall Recommendation:** 5
**Confidence:** 4

**Summary:**

The paper introduces a new visual reasoning benchmark, "VisualPuzzles", to test the domain-specific knowledge of existing LMMs. It creates high-quality data that does not contain intensive knowledge, ensuring the evaluation reflects the exact reasoning ability of tested models. The motivation and the novelty are good. I highly appreciate the correlation analysis of Figure 3, which makes the experimental results clearly match the motivation and claims.

**Compliance With Llm Reviewing Policy:**

Affirmed.

**Final Justification:**

I insist it is a good paper, even with minor evaluation limitations. The main contributions are sufficient, and I would like to see such a visual logic type mm benchmark being accepted, even though the Visulogic is already accepted. I will keep my original positive score.

**Key Questions For Authors:**

How do you ensure the discrepancy between knowledge-intensive and reasoning-intensive samples?

**Limitations:**

The main paper should have a clear section to discuss the potential limitations.

**Strengths And Weaknesses:**

**Strengths:**

1. This paper proposes a new reasoning-intensive benchmark. Although VisuLogic is the contemporary work with similar contributions, it is still a good benchmark on generic multimodal reasoning domains, which is highly different from the existing agentic benchmarks (e.g., GUI, UI 2code, HTML2Code). Such a benchmark shows significant differences from previous Seed-Bench, MMBench, MMMU-Pro.
2. Such a type of reasoning data is closer to the visual understanding rather than multimodal math reasoning, as these math benchmarks rely heavily on text-format latent reasoning.
3. The data annotation process and sources are reliable, thus the quality of curated data is acceptable.
4. The evaluation dimensions are sufficient and bring several interesting summaries.

**Weaknesses:**
1. The main concerns are the quantity of tested models. I recommend that authors test the new Qwen3-VL families and InternVL3.5 families models for comprehensive evaluations. Additionally, we would like to see the latest proprietary models on this benchmark. Even though the accuracy is probably high, the evaluation results are still valuable for subsequent researchers to reference.

2. There is a minor citation error in Lines 1648 and 1712.

3. The paper should include more case analysis rather than the quantitative results. The exact case studies would help to understand the model's behaviors.

Overall, though the evaluation models are not the latest, I still consider the contribution of the benchmark is the data itself, and such a paper meets the standard to publish.

---

> ### Author Rebuttal · Authors · 2026-03-31
>
> Thank you for recognizing the value and quality of VisualPuzzles. We hope to address your concerns as follows:
>
> > The main concerns are the quantity of tested models. I recommend that authors test the new Qwen3-VL families and InternVL3.5 families models for comprehensive evaluations. Additionally, we would like to see the latest proprietary models on this benchmark. Even though the accuracy is probably high, the evaluation results are still valuable for subsequent researchers to reference.
>
> Thank you for the suggestion! We agree that including results from more recent models would strengthen the comprehensiveness of our evaluation. As an initial step during the rebuttal period, we additionally evaluated **Qwen3-VL-30B-A3B (50.7% accuracy), Kimi-K2.5 (51.4%), Claude-4.5-Sonnet (49.4%), and Gemini-3-Pro (52.7%)** [1][2][3][4]. We will add these results to the final version, and we will continue adding more recent models thereafter.
>
> [1] Shuai Bai, Yuxuan Cai, Ruizhe Chen, Keqin Chen, Xionghui Chen, Zesen Cheng, Lianghao Deng et al. "Qwen3-vl technical report." arXiv preprint arXiv:2511.21631 (2025).
>
> [2] Kimi Team, Tongtong Bai, Yifan Bai, Yiping Bao, S. H. Cai, Yuan Cao, Y. Charles et al. "Kimi K2. 5: Visual Agentic Intelligence." arXiv preprint arXiv:2602.02276 (2026).
>
> [3] Anthropic. "Introducing Claude Sonnet 4.5." anthropic.com/news/claude-sonnet-4-5 (2025).
>
> [4] Gemini Team. "Gemini 3: A new era of intelligence with Gemini 3." https://blog.google/products-and-platforms/products/gemini/gemini-3/ (2025)
>
> > There is a minor citation error in Lines 1648 and 1712.
>
> Thank you for pointing this out. We will correct these citation errors in the final version.
>
> > The paper should include more case analysis rather than the quantitative results. The exact case studies would help to understand the model's behaviors.
>
> Thank you for the helpful suggestion! We agree that concrete case studies are very valuable for understanding model behaviours beyond quantitative results. In fact, the current paper already includes a substantial set of case studies in **Appendix M**, where we provided sample correct and error cases for each reasoning category and each difficulty level, aggregating to **30 case studies** in total. We agree that these case studies are not discussed in the main paper. Thus, in the final version, we will make these qualitative case studies more visible by explicitly referencing them in Section 5 (Analysis). We hope this will make the observed model behaviours and failure modes easier to interpret.
>
> > How do you ensure the discrepancy between knowledge-intensive and reasoning-intensive samples?
>
> Thank you for the question. We didn't treat the discrepancy between knowledge-intensive and reasoning-intensive samples as an assumption. Instead, we quantify knowledge-intensive and reasoning-intensive, as discussed in Section 4.1 and Section 4.2 respectively.
>
> To quantify knowledge intensity, we constructed knowledge checklists for VisualPuzzles and MMMU instances. As shown in Table 3, we found that under this protocol, MMMU requires an average of 3.9 knowledge questions per instance, while VisualPuzzles requires only 1.1 per instance, indicating substantially lower knowledge intensity of VisualPuzzles. Additionally, as shown in Figure 3, we also find that many models achieve over 90% knowledge accuracy on VisualPuzzles but often fall below 60% on MMMU, showing that the knowledge required by VisualPuzzles is already mostly possessed by models, unlike MMMU.
>
> To quantify reasoning intensity, we analyze step-by-step solutions and measure the proportion of steps that are genuine logical reasoning steps rather than knowledge retrieval or other operations. As shown in Table 4, logical reasoning accounts for an average of 82.1% of solution steps on VisualPuzzles, compared with an average of 71.5% on MMMU, indicating that VisualPuzzles demands more intensive reasoning.

---

> > ### Author Rebuttal · Reviewer_vaqX · 2026-04-02
> >
> > Thank you for your rebuttal. I insist it is a good paper, even with minor evaluation limitations. The main contributions are sufficient, and I would like to see such a visual logic type mm benchmark being accepted, even though the Visulogic is already accepted. I will keep my original positive score.

---

### Official Review · Reviewer_27SA · 2026-03-11

**Soundness:** 4
**Presentation:** 4
**Significance:** 4
**Originality:** 3
**Overall Recommendation:** 6
**Confidence:** 5

**Summary:**

This paper introduces VisualPuzzles, a new benchmark designed to evaluate multimodal reasoning independently of domain-specific knowledge. The authors argue that existing multimodal benchmarks confound reasoning ability with domain knowledge, making it difficult to assess genuine reasoning capabilities of MLLMs. The authors compare several proprietary and open multimodal models (e.g., GPT-4o, Gemini, Claude, Qwen-VL, LLaVA) with human participants. They report several key findings, including 1) All evaluated models perform below the 5th percentile of human performance on their benchmark. 2) Strong performance on knowledge-heavy benchmarks (e.g., MMMU) does not transfer well to VisualPuzzles. 3) Increasing model size does not correlate strongly with performance. 4) Chain-of-thought or thinking modes produce inconsistent improvements. The authors further analyze reasoning patterns (branching, re-validation) and error types, concluding that reasoning errors dominate model failures. The benchmark is proposed as a more reliable way to evaluate reasoning ability in multimodal models without confounding knowledge retrieval.

**Compliance With Llm Reviewing Policy:**

Affirmed.

**Final Justification:**

The paper addresses an important evaluation gap with a well-designed benchmark and strong empirical analysis, and the authors have satisfactorily addressed my concerns in the rebuttal, so I support acceptance.

**Key Questions For Authors:**

1. Since puzzle questions and civil service exam problems are widely available online, how do you ensure that models have not seen similar problems during training?

2. Error analysis suggests a large fraction of errors are perceptual or spatial. Can you isolate reasoning errors from perception errors more rigorously? Or if you ignore questions that model made perceptual and spatial errors, would you observe a similar performance?

**Limitations:**

Yes

**Strengths And Weaknesses:**

Strengths
1. The paper identifies a real evaluation gap in multimodal reasoning research that current benchmarks often require domain knowledge, making it unclear whether models are reasoning or retrieving facts. This is an important issue in the era of reasoning models and aligns with ongoing debates in the LLM evaluation literature.
2. The dataset design is carefully designed and well structured, with 5 canonical reasoning categories, balanced dataset, and different difficulty levels. The effort to filter out domain knowledge dependencies is thoughtful and aligns with the benchmark’s goal.
3. The authors evaluate a broad set of models including proprietary models and open models, as well as their reasoning-enabled counterparts. This provides useful comparative insights about reasoning vs knowledge benchmarks.
4. The results and observations presented in the paper are insightful and contribute useful empirical evidence about the current capabilities and limitations of MLLMs. The analysis highlights important considerations in how reasoning ability should be evaluated in AI and raises awareness about potential pitfalls in existing evaluation practices. These insights help motivate the need for more careful evaluation methodologies and could influence how future benchmarks and evaluation protocols for MLLMs are constructed.
5. Including human performance with percentiles is valuable and strengthens the benchmark analysis.

Weaknesses
1. I have some concerns about potential data contamination. If the data sources are widely available online, the models may have encountered similar problems during training. This would not necessarily weaken the paper’s conclusions—in fact, it could even make the results more compelling—but it is a general concern I have with datasets curated from internet sources.

2. For vision datasets, a large portion of the errors arises from perception failures, which is also confirmed in the paper’s error analysis. Given this significant share of visual errors, it is difficult to draw conclusions about the models’ reasoning abilities solely from the gap between model and human performance. If the models had perceptual abilities comparable to those of humans, they might achieve similar performance on this dataset. A similar issue has been observed with the ARC-AGI dataset, where researchers often evaluate textual versions of the tasks to partially bypass visual perception errors. It would therefore be interesting if the authors could adopt a similar approach to disentangle visual perception ability from reasoning ability.

---

> ### Author Rebuttal · Authors · 2026-03-31
>
> Thank you for your thoughtful feedback, and for recognizing the usefulness of VisualPuzzles and the comprehensiveness of our experiments and analysis! We hope to address your concerns as follows:
>
> > W.1 and Q.1: I have some concerns about potential data contamination. If the data sources are widely available online, the models may have encountered similar problems during training. This would not necessarily weaken the paper’s conclusions—in fact, it could even make the results more compelling—but it is a general concern I have with datasets curated from internet sources.
>
> Thank you for raising this point! We want to clarify that most puzzles were not sourced from easily accessible online resources. As described in Appendix B.2, VisualPuzzles was assembled from multiple sources, where many of the puzzles were manually gathered, translated, adapted or generated, rather than simply copied from easily accessible web sources.
>
> > W.2 and Q.2: Error analysis suggests a large fraction of errors are perceptual or spatial. Can you isolate reasoning errors from perception errors more rigorously? Or if you ignore questions that model made perceptual and spatial errors, would you observe a similar performance?
>
> Thank you for the important question. We agree that perception and reasoning should be separated as clearly as possible to provide more interesting analysis. In our current error analysis of Claude 3.7 Sonnet on VisualPuzzles, the model’s overall accuracy is 47.2%, corresponding to an error rate of 52.8%. Among the errors, we identify 21% perceptual error, 17% spatial / orientation error, as shown in Figure 7. Under a conditional estimate, these two categories together account for about 20.1% of all examples (i.e., 52.8% x (21%+17%)). **If we exclude these perceptual and spatial failure cases and consider only the remaining samples, the model’s conditional accuracy is approximately 59.1%** (i.e., 47.2% / (100%-20.1%)). While this is only a coarse estimate, it still suggests that even after discounting perceptual and spatial failure, performance remains far from solved, consistent with our conclusion that reasoning is a major bottleneck for models. We will add this discussion to the final version.

---

> > ### Author Rebuttal · Reviewer_27SA · 2026-03-31
> >
> > I appreciate the rebuttal. It has fully resolved my concerns.

---

### Official Review · Reviewer_qQMp · 2026-03-11

**Soundness:** 3
**Presentation:** 3
**Significance:** 3
**Originality:** 3
**Overall Recommendation:** 3
**Confidence:** 4

**Summary:**

This paper introduces VISUALPUZZLES, a multimodal benchmark designed to measure visual reasoning while deliberately minimizing domain-specific knowledge. The dataset comprises 1,168 multiple-choice puzzles across five reasoning categories (algorithmic, analogical, deductive, inductive, spatial) with balanced difficulty and option modalities. The authors evaluate a wide range of MLLMs and conduct a human study, finding that models perform below the human 5th percentile, that performance on knowledge-heavy benchmarks like MMMU does not transfer, and that scaling model size or using “thinking/CoT” modes yields inconsistent benefits. They further provide analyses supporting the claims that VISUALPUZZLES is knowledge-light and demands more logical reasoning steps than MMMU.

**Compliance With Llm Reviewing Policy:**

Affirmed.

**Final Justification:**

While most of my concerns have been addressed, I prefer to maintain my original score, since the paper would benefit from an additional revision incorporating the clarifications and changes discussed in the rebuttal.

**Key Questions For Authors:**

1) How were the five reasoning categories and difficulty levels annotated? Please report inter-annotator agreement (e.g., Cohen’s kappa) and provide concrete decision guidelines to reduce category ambiguity?

2) Could you provide standardized prompting templates, sampling parameters, and temperature/seed settings for all models, and report sensitivity analyses showing robustness of rankings to prompt variants?

3) Are there known ambiguous or multi-answer items? If so, how many, and how were they handled?
Please provide an adjudication protocol and a public errata process for future corrections?

**Limitations:**

Yes

**Strengths And Weaknesses:**

**Strengths:**
The benchmark explicitly targets the disentanglement of multimodal reasoning from domain-specific knowledge, a timely and important evaluation desideratum.

The coverage of five core reasoning types with balanced modality (text vs image options) and difficulty offers a well-scoped, comprehensive probe of reasoning skills.

The motivation is clear, and the framing around minimizing knowledge requirements is consistent throughout.

Provides a platform that can steer research toward models with genuine visual reasoning ability rather than improved factual recall.

**Weakness:**

Reliance on an LLM (GPT-4o) to generate “knowledge concept checklists” and to help classify solution steps as “logical reasoning” introduces potential circularity and bias; limited human auditing is mentioned but not quantified.

Multiple-choice format with four options can enable guessing and shortcut strategies; no adversarial or bias-control analysis of option formats/order is reported.

The “best of direct vs CoT” per model selection can advantage certain models; a uniform prompting protocol with fixed temperatures and seeds should be more explicitly controlled and reported.
Potential data contamination is not discussed; popular puzzles (e.g., “count squares”) may exist online and in pretraining corpora, which complicates the claim that failures are purely reasoning-related.

Limited discussion of classical reasoning-with-minimal-knowledge benchmarks: Raven/I-RAVEN(Małkiński & Mańdziuk, 2024), PGM(Shang et al., 2025), Bongard-LOGO(Nie et al., 2020), SVRT(Ren & Yi, 2004), ARC/ARC-AGI(Liao & Gu, 2025), and more recent multi-figure analogical/spatial suites; positioning relative to these would clarify incremental vs novel aspects.

*References *

Liao, I., & Gu, A. (2025). ARC-AGI Without Pretraining. https://doi.org/10.48550/arxiv.2512.06104

Małkiński, M., & Mańdziuk, J. (2024). A-I-RAVEN and I-RAVEN-Mesh: Two New Benchmarks for Abstract Visual Reasoning. https://doi.org/10.48550/arxiv.2406.11061

Nie, W., Yu, Z., Mao, L., Patel, A., Zhu, Y., & Anandkumar, A. (2020). Bongard-LOGO: A New Benchmark for Human-Level Concept Learning and Reasoning. https://doi.org/10.48550/arxiv.2010.00763

Ren, Z. F., & Yi, Z. S. (2004). Applied clipping based on multibounding types in SVRT. 5444. https://doi.org/10.1117/12.561229

Shang, J., Kreiman, G., & Sompolinsky, H. (2025). Unraveling the geometry of visual relational reasoning. https://doi.org/10.48550/arxiv.2502.17382

---

> ### Author Rebuttal · Authors · 2026-03-31
>
> Thank you for recognizing the importance of VisualPuzzles and the comprehensiveness of evaluation! We hope to address your concerns as follows:
>
> > Reliance on an LLM (GPT-4o) to generate “knowledge concept checklists” and to help classify solution steps as “logical reasoning” introduces potential circularity and bias; limited human auditing is mentioned but not quantified.
>
> We agree that reliance on LLM might introduce potential circularity and bias. Thus, we conducted human validation where we asked two human annotators to manually verify and correct the knowledge checklist questions, and the exact instructions are demonstrated in Appendix F.5. The inter-annotator agreement rate is 87.8%.
>
> > Multiple-choice format with four options can enable guessing and shortcut strategies
>
> In this work, we intentionally chose a multiple-choice format to ensure consistent, automatic grading across models, and isolate whether models can arrive at correct solutions without conflating this with abilities to produce long outputs. Importantly, this format is not trivial: a random guess yields 25% accuracy, while many models remain only modestly above this level, far below humans, suggesting that "gaming" the format is not sufficient to do well. We will add this clarification to the final version.
>
> > The “best of direct vs CoT” per model selection can advantage certain models; a uniform prompting protocol with fixed temperatures and seeds should be more explicitly controlled and reported.
>
> We appreciate and understand this concern. Our intention was to reflect the best outcome for each model, rather than to present a strictly unified prompting protocol. We report the full results for both direct and CoT prompting in Appendix E.
>
> > Potential data contamination is not discussed; popular puzzles (e.g., “count squares”) may exist online and in pretraining corpora, which complicates the claim that failures are purely reasoning-related.
>
> We want to clarify that most puzzles were not sourced from easily accessible online resources. As described in Appendix B.2, VisualPuzzles was assembled from multiple sources, where many of the puzzles were manually gathered, translated, adapted or generated, rather than simply copied from easily accessible web sources.
>
> > Limited discussion of classical reasoning-with-minimal-knowledge benchmarks: Raven/I-RAVEN(Małkiński & Mańdziuk, 2024), PGM(Shang et al., 2025), Bongard-LOGO(Nie et al., 2020), SVRT(Ren & Yi, 2004), ARC/ARC-AGI(Liao & Gu, 2025).
>
> Thank you for reminding us of these important related benchmarks! In the final version, we will add these references to the Related Work section. We will also clarify that the main contribution of VisualPuzzles in comparison with prior benchmarks is to provide a multimodal benchmark tailored to evaluating current MLLMs with an explicit focus on disentangling reasoning from domain knowledge.
>
> > How were the five reasoning categories and difficulty levels annotated? Please report inter-annotator agreement (e.g., Cohen’s kappa) and provide concrete decision guidelines to reduce category ambiguity?
>
> Appendix D provides concrete annotation instructions for labeling reasoning and difficulty categories. Around 4% of the questions, annotators disagree on the labeling of the reasoning category, and either the annotators reach consensus through discussion or the puzzles are removed, as discussed in Line 1269-1270.
>
> > Could you provide standardized prompting templates, sampling parameters, and temperature/seed settings for all models, and report sensitivity analyses showing robustness of rankings to prompt variants?
>
> Thanks for the suggestion! We provide the standardized prompting templates in Appendix C (Model Evaluation Setup). We will add the sampling parameters, temperature/seed settings for all models to Appendix C in the final version. Regarding sensitivity analyses showing robustness of rankings to prompt variants, we already reported model results on two prompt variants (CoT prompting and non-CoT prompting) in Appendix E.1 and E.2 respectively. Following your suggestion, we further measured prompt sensitivity. Across models, the average performance difference between CoT and non-CoT prompting is 0.03%, with a standard deviation of 2.89%, suggesting that **overall model performances and rankings are relatively stable under these common prompting variants**. We will add this sensitivity analysis to the final version.
>
> > Are there known ambiguous or multi-answer items?
>
> To the best of our knowledge, there are no known ambiguous or multi-answer items in VisualPuzzles. As discussed in Section 2.2 (Data Validation), during construction of VisualPuzzles, puzzles flagged by annotators as ambiguous, low-quality, or requiring domain knowledge were either adjudicated through discussion or removed from the benchmark. We will add the instruction for annotators on data validation in the final version.

---

> > ### Author Rebuttal · Reviewer_qQMp · 2026-04-03
> >
> > While most of my concerns have been addressed, I prefer to maintain my original score, since the paper would benefit from an additional revision incorporating the clarifications and changes discussed in the rebuttal.

---

### Official Review · Reviewer_JRXn · 2026-03-15

**Soundness:** 3
**Presentation:** 3
**Significance:** 3
**Originality:** 3
**Overall Recommendation:** 4
**Confidence:** 4

**Summary:**

This paper introduces VISUALPUZZLES, a new multimodal benchmark whose goal is to measure “genuine reasoning ability” while minimizing the influence of how much domain knowledge a model has memorized.
This benchmark contains approximately 1.2K visual puzzle questions, covering five reasoning categories: algorithmic, analogical, deductive, inductive, and spatial, with an emphasis on low knowledge dependence but high reasoning complexity.
The experiments show that current multimodal models still lag significantly behind humans on such knowledge-light, reasoning-heavy tasks, and that strong performance on knowledge-intensive benchmarks such as MMMU does not necessarily translate into strong performance on VISUALPUZZLES.
The authors further find that simply increasing model scale, knowledge capacity, or inference-time compute does not consistently improve performance on this type of visual reasoning task, suggesting that more targeted modeling of multimodal reasoning ability is needed in the future.

**Compliance With Llm Reviewing Policy:**

Affirmed.

**Final Justification:**

Overall, this is a relatively comprehensive multimodal evaluation benchmark, and my concerns were also addressed during the rebuttal period. For detailed comments, please refer to my review comments

**Key Questions For Authors:**

1. Regarding the analysis in Section 5.1, I would like to know whether the fact that models choose different strategies implies that these are the optimal solution strategies under the current benchmark, or whether such differences merely arise from different model reactions to latent patterns in the questions. Since the difference in strategy does not fully determine whether a model can answer correctly, it would provide further insight if the authors could additionally use explicit prompts to enforce different strategies and then compare the resulting performance.

**Limitations:**

yes

**Strengths And Weaknesses:**

# Strengths
1. The paper deliberately positions the benchmark in the “knowledge-light, reasoning-heavy” regime, allowing models to more directly expose their reasoning shortcomings. This design aligns well with the field’s current focus on model intelligence level rather than knowledge coverage.
2. The paper goes beyond reporting accuracy alone and also analyzes reasoning patterns and answering strategies in model outputs, which makes the authors’ evaluation perspective more convincing.
3. The evaluation includes both open-source and closed-source models, human percentile performance, thinking/non-thinking model variants, and horizontal comparisons with other benchmarks in this area, providing comprehensive evidence in support of the paper’s claims.

# Weaknesses
1. The paper provides a brief motivation and some literature support for its five-category reasoning taxonomy, but it does not sufficiently justify the completeness of this taxonomy, the clarity of category boundaries, or the reliability of the annotations. Therefore, the adequacy of this taxonomy as a coverage framework for “general reasoning” still requires further validation.
2. The benchmark uses only multiple-choice questions, making the task format overly simple.
3. Some of the comparative results in the tables are somewhat outdated, lacking evaluation results for more recent open-source models (e.g., Qwen3-VL) and closed-source models (e.g., Kimi-K2 and Seed1.8). If the authors could further include results from multimodal RLVR methods from the past year, this would further improve the comprehensiveness of the evaluation.
4. The authors argue that “simply increasing inference compute or extending CoT does not guarantee gains on this puzzle-style benchmark.” However, the model comparisons used to support this claim are based on models from more than a year ago. Due to the absence of a new generation of multimodal reasoning models from the most recent six months, the timeliness and external validity of this conclusion still require further verification.
5. Typo: At the end of Section 2.3, the paper states, “We demonstrate these traits of VISUALPUZZLES in sec 5.” However, this discussion should in fact correspond to Section 4.

---

> ### Author Rebuttal · Authors · 2026-03-31
>
> Thank you for recognizing the usefulness of VisualPuzzles and the comprehensiveness of our experiments! We hope to address your concerns as follows:
>
> > W.1: The paper does not sufficiently justify the completeness of this taxonomy, the clarity of category boundaries, or the reliability of the annotations.
>
> Thank you for pointing this out! We do not intend to claim that these five categories form an exhaustive taxonomy of all reasoning. Instead, we aim to provide a broad yet manageable coverage framework for reasoning abilities relevant to MLLMs. We will add this clarification to the paper. Regarding clarity of category boundaries, we agree that these reasoning categories are not strictly mutually exclusive. These five categories were selected as they represent fundamental forms of reasoning widely discussed in literature. During annotations, annotators first decided all potential labels for each puzzle, then selected a single **primary** label using a fixed rubric: (i) choose the category whose absence makes the puzzle unsolvable; (ii) if two or more remain, pick the more specific one (priority order: Spatial = Algorithmic = Analogical > Inductive = Deductive). Each question is labeled by three annotators for the reasoning category. Around 4% of the questions, annotators disagree on the labeling of the reasoning category, and either the annotators reach consensus through discussion or the puzzles are removed, as discussed in Line 1269-1270.
>
> > W.2: The benchmark uses only multiple-choice questions, making the task format overly simple.
>
> We agree that multiple-choice and open-ended generation probe different aspects of model behavior. In this work, we intentionally chose a multiple-choice format to ensure consistent, automatic grading across models, and isolate whether models can arrive at correct solutions without conflating this with abilities to produce long outputs. Importantly, this format is not trivial: a random guess yields 25% accuracy, while many models remain only modestly above this level, far below humans, suggesting that "gaming" the format is not sufficient to do well. Evaluation of open-ended argument generation is out of the scope of this paper, but it is a valuable future direction. We will add this clarification to the final version.
>
> > W.3: Lack evaluation results for more recent open-source models (e.g., Qwen3-VL) and closed-source models (e.g., Kimi-K2 and Seed1.8).
>
> Thank you for the suggestion! As an initial step during the rebuttal period, we additionally evaluated **Qwen3-VL-30B-A3B (50.7% accuracy), Kimi-K2.5 (51.4%), Claude 4.5 Sonnet (49.4%), and Gemini-3-Pro (52.7%)**. We will add these results to the final version, and we will continue adding more recent models thereafter.
>
> > W.4: The authors argue that “simply increasing inference compute or extending CoT does not guarantee gains on this puzzle-style benchmark.” However, the model comparisons used to support this claim are based on models from more than a year ago.
>
> Thank you for the suggestion. To address this concern, we additionally evaluated a more recent model Claude 4.5 Sonnet on VisualPuzzles with and without thinking enabled, showing only a marginal gain of 1% with extended thinking. This new result is consistent with our original observation that simply increasing inference-time compute via extended thinking does not reliably produce substantial improvements on VisualPuzzles. We will add this new result in the final version.
>
> |Model|Accuracy|
> |-|-|
> |Claude Sonnet 4.5 (thinking)|49.4%|
> |Claude Sonnet 4.5 (no thinking)|48.4%|
>
> > W.5: Typo: At the end of Section 2.3.
>
> Thanks for pointing this out! We will correct this in the final version.
>
> > Q.1: Regarding the analysis in Section 5.1, do the observed strategy differences indicate optimal strategies? Could explicit prompting verify this?
>
> Thank you for the interesting suggestion! Following your suggestion, we conducted an additional  analysis, where we used explicit prompts to enforce the Answer-First strategy and the Option-First strategy on Kimi-K2.5:
>
> |Benchmark|Answer-First|Option-First|
> |-|-|-|
> |VisualPuzzles|51.1%|50.3%|
> |MMMU|36.1%|38.1%|
>
> These results suggest an interesting finding:
> **when we explicitly enforce the two strategies through prompts, the relative performance shifts in the same direction as the naturally observed preferences.** In Section 5.1, our original observation was that models tend to adopt an Answer-First strategy more often on VisualPuzzles, but adopt an Option-First strategy more often on MMMU. Now the additional experiment shows that, though the difference is relatively modest, on VisualPuzzles, Answer-First performs better than Option First; whereas on MMMU, Option-First performs better than Answer-First. This might indicate the fact that models choose different strategies implies that these are the optimal solution strategies under the current benchmark. We will incorporate these analyses into the final version.

---

> > ### Author Rebuttal · Reviewer_JRXn · 2026-04-07
> >
> > Thanks for the rebuttal. My concerns have been addressed.

---

### Decision · Program_Chairs · 2026-04-30

**Decision:**

Accept (regular)

**Comment:**

The reviewers are overwhelmingly positive about the introduction of **VISUAL PUZZLES**, a new benchmark specifically designed to disentangle genuine multimodal reasoning from domain-specific knowledge memorization. The consensus is that this benchmark addresses a critical gap in the field, as current multimodal benchmarks (like MMMU) often conflate factual recall with reasoning ability.


During the rebuttal period, the authors successfully addressed several initial concerns. Over all, the paper is technically solid and highly relevant to the evolving landscape of "reasoning" models. By providing a curated set of 1.2K puzzles that models cannot solve through factual retrieval, it forces a focus on internal logic and visual grounding. While some reviewers noted the inherent difficulty of completely isolating perception from reasoning, they agreed that the benchmark is a significant step forward. The authors' commitment to updating the leaderboard with new models further ensures the paper's long-term utility.